# Anillin mediates unilateral furrowing during cytokinesis by limiting RhoA binding to its effectors

Mikhail Lebedev[1], Fung-Yi Chan[3,4], Elisabeth Rackles[2], Jennifer Bellessem[2], Tamara Mikeladze-Dvali[2], Ana Xavier Carvalho[3,4], and Esther Zanin[1,2]

During unilateral furrow ingression, one side of the cytokinetic ring (leading edge) ingresses before the opposite side (lagging edge). Anillin mediates unilateral furrowing during cytokinesis in the one-cell *C. elegans* zygote by limiting myosin II accumulation in the ring. Here, we address the role of anillin in this process and show that anillin inhibits not only the accumulation of myosin II but also of other RhoA effectors by binding and blocking the RhoA effector site. The interaction between the anillin's RhoA-binding domain (RBD) and active RhoA is enhanced by the disordered linker region and differentially regulated at the leading and lagging edge, which together results in asymmetric RhoA signaling and accumulation of myosin II. In summary, we discover a RhoA GEF- and GAP-independent mechanism, where RhoA activity is limited by anillin binding to the RhoA effector site. Spatial fine-tuning of anillin's inhibitory role on RhoA signaling enables unilateral furrow ingression and contributes to animal development.

## Introduction

During cell division, a contractile ring, consisting of filamentous actin (F-actin) and the motor protein non-muscle myosin II, assembles and constricts to form the cleavage furrow (Mishima, 2016; Pollard and O'Shaughnessy, 2019). F-actin polymerization is mediated by formin, and non-muscle myosin II is activated by Rho kinase (ROK) (Amano et al., 1996; Li and Higgs, 2003; Osório et al., 2019). The autoinhibitory conformation of formin and ROK is released upon binding to active GTP-bound RhoA (Amano et al., 2010; Kühn and Geyer, 2014; Matsui et al., 1996; Otomo et al., 2005; Rose et al., 2005). RhoA cycles between the active GTP- and inactive GDP-bound form, and during cytokinesis, activation is mediated by the guanine-nucleotide exchange factor (GEF) ECT2 and inactivation by the GTPase-activating protein (GAP) RGA-3/4 (MP-GAP in humans) (Prokopenko et al., 1999; Tatsumoto et al., 1999; Yüce et al., 2005; Zanin et al., 2013). After anaphase onset, spindle-derived signals promote the activation of ECT2 at the cell equator and inactivation of the GAP at the poles of the cell, which together generate a narrow zone of active RhoA at the cell equator (Bement et al., 2005; Burkard et al., 2007; Gómez-Cavazos et al., 2020; Mangal et al., 2018; Petronczki et al., 2007; Schneid et al., 2021; Wolff et al., 2023, *Preprint*; Zanin et al., 2013). In addition to those initial cues, several biochemical feedback mechanisms fine-tune RhoA activity by altering ECT2 and RGA-3/4 activity or localization. Positive feedback mechanisms facilitate rapid

RhoA activation by enhancing ECT2 activity via active RhoA itself (Chen et al., 2019) or centralspindlin (Zhang and Glotzer, 2015). Negative feedback mechanisms limit RhoA activity by targeting the RhoA GAPs RGA-3/4 to the cortex via GCK-1/CCM-3 (Bell et al., 2020) and F-actin (Michaud et al., 2022; Michaux et al., 2018).

In addition to those biochemical feedbacks, cortical actin flows deliver new material to the ring from the cell poles towards the furrow region to maintain a constant ring constriction rate and contribute to the alignment of the actin filaments (Khaliullin et al., 2018; Leite et al., 2020; Reymann et al., 2016; Singh et al., 2019). Ring-directed polar flows cause a compression of the actin cortex at the cell equator during ring assembly (Khaliullin et al., 2018; Reymann et al., 2016). Compression continues during ring constriction since the cortical surface area flowing into the furrow region is larger than the surface area generated by ring constriction (Khaliullin et al., 2018). In addition to the ring-directed polar flows, rotational flows around the cell equator are induced when embryos are mechanically compressed by a coverslip during image acquisition (Leite et al., 2020; Singh et al., 2019). Even in the absence of mechanical confinement of the embryos, ring constriction induces a circumferential flow from the leading edge toward the lagging edge, resulting in a compression of the actin cortex at the lagging edge (Hsu et al., 2023).

[1]Department Biologie, Friedrich-Alexander-Universität Erlangen-Nürnberg, Erlangen, Germany; [2]Department Biology II, Ludwig-Maximilians University Munich, Munich, Germany; [3]i3S - Instituto de Investigação e Inovação em Saúde (i3S), Universidade do Porto, Porto, Portugal; [4]IBMC - Instituto de Biologia Molecular e Celular, Universidade do Porto, Porto, Portugal.

Correspondence to Esther Zanin: esther.zanin@fau.de.

**Rockefeller University Press**
J. Cell Biol. 2025 Vol. 224 No. 6   e202405182

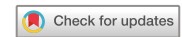

The amount of active RhoA is not only controlled by the GEF ECT2 and the GAP RGA-3/4 but also by the multidomain protein anillin. In human and *Drosophila* cells, anillin promotes RhoA signaling as its depletion reduces RhoA activity and causes the furrow to be unstable (Budnar et al., 2019; Hickson and O'Farrell, 2008; Oegema et al., 2000; Piekny and Glotzer, 2008; Straight et al., 2005; Zhao and Fang, 2005). Anillin harbors a conserved RhoA binding domain (RBD), a lipid binding region (C2), and a pleckstrin homology (PH) domain at the C-terminus (Sun et al., 2015). Via its C2 region, anillin is targeted to phosphatidylinositol 4,5-bisphosphate (PIP2)-rich membrane regions (Budnar et al., 2019) and the PH domain binds to septins (Kinoshita et al., 2002; Liu et al., 2012; Oegema et al., 2000). In addition, human anillin interacts with the effector binding site of active RhoA via the RBD domain, and subsequently, the C2 region enriches RhoA in PIP2 clusters where RhoA binds and activates its effectors. After effector release, RhoA binds to anillin again, and this cycle of effector activation and anillin binding occurs reiteratively, thereby increasing the residence time of active RhoA on the membrane and facilitating RhoA signaling (Budnar et al., 2019). Thus, in human cells, anillin stabilizes and increases the residence time of active RhoA on the plasma membrane. Consistent with the idea that anillin stabilizes RhoA in *Caenorhabditis elegans*, the anillin homolog (ANI-1) enriches myosin II (NMY-2) in cortical patches during polarization of the one-cell zygote and stabilizes NMY-2 in the ring in four-cell embryos lacking a spindle-midzone (Maddox et al., 2005; Santos et al., 2023). Interestingly, multiple observations do not align with the model that anillin promotes RhoA activity. Anillin depletion increases cortical F-actin and myosin II levels in the one-cell *C. elegans* zygote or *Drosophila* S2 cells during division and elevates RhoA activity at cell–cell junctions in *Xenopus laevis* embryos, suggesting that in certain settings, anillin rather inhibits RhoA signaling (Connors et al., 2024; Craig et al., 2025; Hsu et al., 2023; Jordan et al., 2016; Kechad et al., 2012; Reyes et al., 2014). In addition to the membrane association motifs, anillin also interacts with components of the actin cortex such as F-actin, myosin II, and formin via the N-terminus (Dorn et al., 2016; Field and Alberts, 1995; Jananji et al., 2017; Straight et al., 2005; Tian et al., 2015; Watanabe et al., 2010). The actin–myosin cortex-binding N-terminus and membrane-binding C-terminus of anillin are connected by a highly disordered linker region (Chatterjee and Pollard, 2019; Lebedev et al., 2023). Thus, in addition to regulating RhoA signaling, anillin is considered to be an important connection between the actin–myosin cortex and the membrane.

In case the spindle is positioned in the center along the transverse axis of the cell, RhoA is activated and the contractile ring is initially built all around the cell equator. In numerous instances, constriction of the ring is unilateral (also called asymmetric) with one edge of the ring (the leading edge) ingressing before the other one (lagging edge) (Sugioka, 2022). Unilateral furrow ingression contributes to the robustness of embryonic cytokinesis in *C. elegans* and the morphogenesis of tissues in developing animals (Das et al., 2003; Dorn et al., 2016; Fleming et al., 2007; Kosodo et al., 2008; Maddox et al., 2007; Reinsch and Karsenti, 1994; Singh et al., 2019). Unilateral closure

of the ring is controlled by spindle-independent signals that are incompletely understood. The current model proposes that the leading edge is initially defined by multiple positive feedback mechanisms including F-actin alignment along the curved furrow membrane (Dorn et al., 2016), F-actin guided actin filament assembly (Li and Munro, 2021), and the delivery of ring material by ring-directed flows (Khaliullin et al., 2018). Since ring-directed cortical flows start earlier and are higher at the leading than the lagging edge, cortical ring components enrich at the leading edge and thus are asymmetrically distributed around the ring (Khaliullin et al., 2018). Previous work showed that the myosin II enrichment rate positively correlates with the ring-directed flow velocities, suggesting that a mechanosensitive positive feedback mechanism defines the rate of myosin II accumulation. Interestingly, in anillin-depleted embryos, the myosin II enrichment rate is strongly increased relative to the ring-directed flow velocities, suggesting that in these embryos the mechanosensitive feedback loop is disrupted (Hsu et al., 2023). Without anillin, myosin II asymmetry is lost, its total levels are increased, and unilateral furrowing fails (Hsu et al., 2023; Maddox et al., 2007). The molecular mechanisms behind the anillin-mediated myosin II accumulation and unilateral furrowing are unknown. Here, we address this question by performing a systematic structure–function analysis of anillin and quantifying the dynamics of contractile ring components during cytokinesis in the one-cell *C. elegans* zygote.

## Results

### Anillin limits the cortical accumulation of the RhoA effectors Formin^CYK-1 and ROK^LET-502 during ring assembly

ANI-1 mediates unilateral furrow ingression by attenuating NMY-2 accumulation in the ring (Hsu et al., 2023) (Fig. 1 A). We started our analysis of the *ani-1(RNAi)* phenotype by imaging endogenously mRFP-tagged NMY-2 at the cell cortex from around anaphase onset (105 s after nuclear envelope breakdown [NEBD]) until the start of furrow ingression (~180 s) (Fig. 1, B–D and Video 1). The equatorial levels of NMY-2::mRFP were elevated after *ani-1(RNAi)* during ring assembly (165 s) and around the onset of furrowing (180 s, Fig. 1, C and D; and Fig. S1, A and B). As published, NMY-2::mRFP was no longer concentrated in cortical patches after *ani-1(RNAi)* (Maddox et al., 2005) (Fig. 1 C).

ANI-1 is expected to bind NMY-2 via its N-terminus (Fig. 1 E) (Lebedev et al., 2023; Maddox et al., 2005); however, since NMY-2 levels were increased and not decreased after ANI-1 depletion, an interaction-dependent regulation of NMY-2 levels seemed unlikely. Alternatively, ANI-1 could bind RHO-1 (RhoA in humans) through its conserved RBD and block RHO-1 signaling by occupying RHO-1's effector binding site. In such a case, the cortical levels of not only NMY-2 but also of RHO-1 effectors are expected to increase after ANI-1 depletion. To test this, we analyzed endogenously GFP-tagged ROK^LET-502 (Fig. S1 C) (Piekny and Mains, 2002), which represents a good readout for active RhoA (Bell et al., 2020). GFP::ROK^LET-502 localized to patches of RHO-1 activity and accumulated at the cell equator during anaphase (Fig. 1 F). Again, we observed that the equatorial GFP::ROK^LET-502 levels were increased in *ani-1(RNAi)*

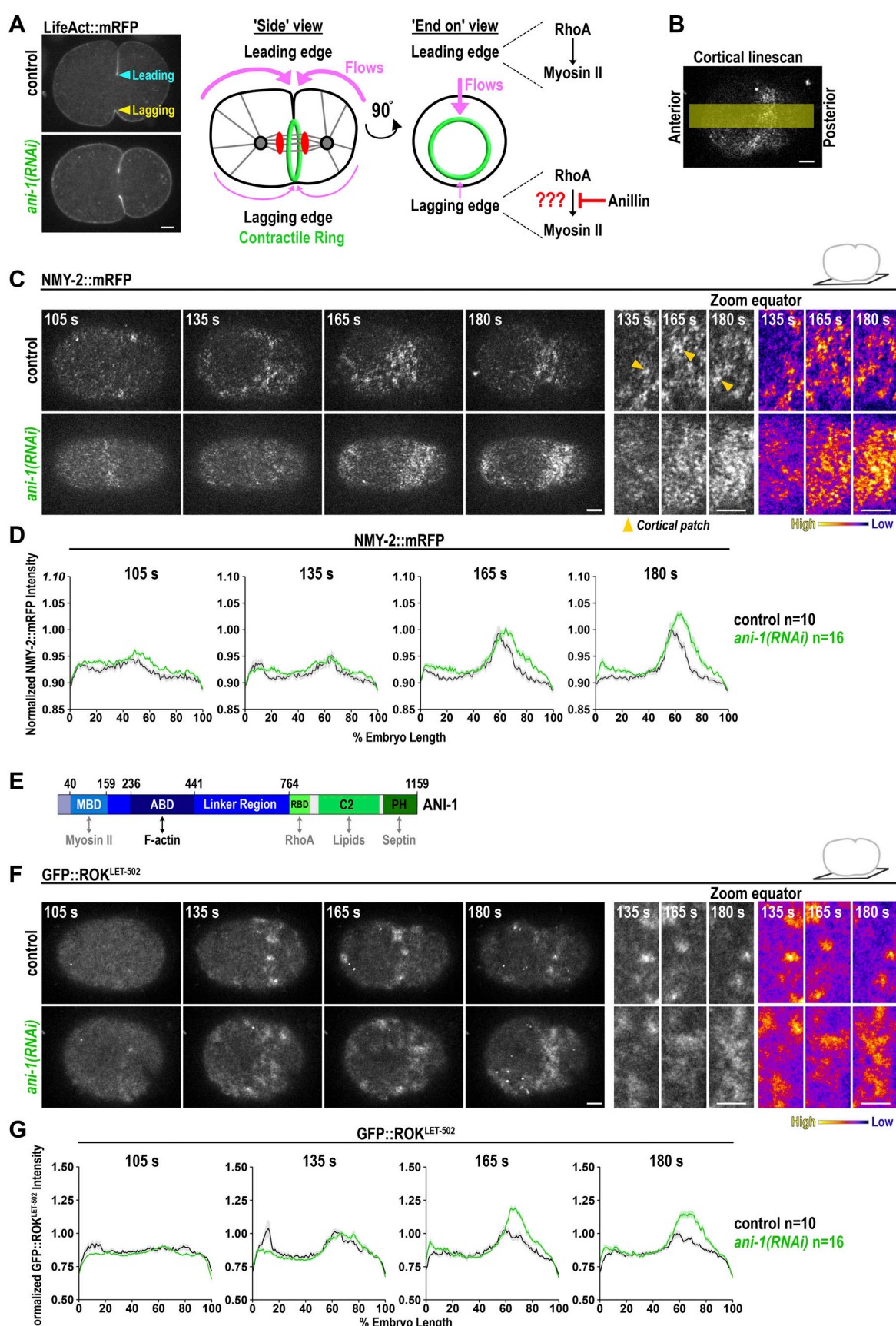

Figure 1.   **ANI-1 limits the accumulation of NMY-2::mRFP and GFP::ROK^LET-502 to the cortex during ring assembly. (A)** In the one-cell *C. elegans* zygote, the cleavage furrow ingresses unilateral with the leading edge ingressing before the lagging edge. The cyan and yellow arrowheads mark the leading edge and

lagging edge, respectively. After ANI-1 depletion by RNAi, the contractile ring ingresses symmetrically (left). Furrow ingression triggers a cortical actin flow from the poles toward the equator at the leading edge. Anillin limits myosin II accumulation in the ring by an unknown mechanism. **(B)** In one-cell *C. elegans* embryos, cortical NMY-2::mRFP fluorescence intensity was measured on single-z-plane images with a line scan from the anterior to the posterior pole. **(C)** Single z-plane images of the cell cortex of one-cell *C. elegans* embryos expressing NMY-2::mRFP treated with or without *ani-1(RNAi)*. A zoom-in of the equatorial region for the indicated time points is shown on the right with gray and fire scaling. Cortical patches of NMY-2::mRFP are highlighted by yellow arrowheads. **(D)** Normalized cortical fluorescence intensity of NMY-2::mRFP from the anterior (0%) to the posterior (100%) pole for control and *ani-1(RNAi)* embryos. **(E)** Domain organization of *C. elegans* ANI-1 highlighting the predicted (grey) and confirmed (black) interaction partners of each domain (MBD—myosin binding domain, ABD—actin-binding domain, RBD—RhoA binding domain, PH—pleckstrin homology). **(F)** Single z-plane images of the cell cortex of one-cell *C. elegans* embryos expressing GFP::ROK$^{LET-502}$ treated with or without *ani-1(RNAi)*. A zoom-in of the equatorial region for the indicated time points is shown with gray and fire scaling. **(G)** Normalized cortical fluorescence intensity of GFP::ROK$^{LET-502}$ from the anterior (0%) to the posterior (100%) pole for control and *ani-1(RNAi)* embryos. For all, time in seconds (s) after NEBD is indicated, scale bars are 5 μm, error bars are standard error of the mean (SEM), and *n* = number of embryos analyzed.

embryos at 165 s and 180 s after NEBD (Fig. 1, F and G; Fig. S1 B; and Video 2). Since ROK$^{LET-502}$ activates NMY-2 (Piekny and Mains, 2002), we tested whether ANI-1 depletion only increases the levels of the ROK signaling cascade or whether another RHO-1 effector, the formin CYK-1 (Fig. S1 C) (Severson et al., 2002), is also elevated. We had previously observed that ANI-1 depletion increased Formin$^{CYK-1}$::GFP levels at the equatorial cortex at 180 s after NEBD (Lebedev et al., 2023). Indeed our multi-timepoint analysis confirmed that the cortical levels of Formin$^{CYK-1}$::GFP increased in *ani-1(RNAi)* embryos during ring assembly and at the onset of furrowing (Fig. S1, D–F).

We conclude that the cortical levels of the two RhoA effectors Formin$^{CYK-1}$ and ROK$^{LET-502}$ as well as the RHO-1 downstream target, NMY-2, were all increased during ring assembly in *ani-1(RNAi)* embryos, suggesting that ANI-1 directly inhibits RhoA signaling.

### The linker region of ANI-1 promotes binding of the RBD to active RHO-1

Human anillin interacts with active RhoA via the RBD (Piekny and Glotzer, 2008; Sun et al., 2015) and *C. elegans* ANI-1 also harbors a conserved RBD at the C-terminus (Fig. 2 A). Thus, we tested whether the ANI-1 C-terminus binds active RHO-1 using pull-down assays. We incubated bacterial purified His-tagged RHO-1 with the GST-tagged C-terminus of ANI-1 (ANI-1$^{C-term}$) comprising the RBD, C2 region, and PH domain (Fig. 2 A). We observed that constitutively active RHO-1 (RHO-1$^{Q63L}$) and, to a much lesser extent, wild type RHO-1 (RHO-1$^{WT}$) bound to the ANI-1$^{C-term}$ (Fig. 2 B). The crystal structure of human and yeast anillin identified a conserved hydrophobic interaction surface between anillin and RhoA (Sun et al., 2015). Mutating two key residues in the binding surface of the *C. elegans* RBD (A789D and E807K), resulted in the loss of interaction between the ANI-1$^{C-term}$ and RHO-1$^{Q63L}$ (Fig. 2 B). Adjacent to the RBD is the intrinsically disordered linker region. Thus, we tested whether the linker region influences the interaction between the RBD and constitutively active RHO-1. Indeed, we observed that a protein consisting of the linker region and C-terminus showed stronger interaction with constitutively active RHO-1 than the C-terminus alone (Fig. 2 C). To test whether there is a second active RHO-1 binding site in the linker region of ANI-1, we repeated the pull-down and incubated only the linker region with RHO-1$^{Q63L}$. We did not observe an interaction between the linker region and RHO-1$^{Q63L}$ (Fig. 2 C). In sum, *C. elegans* ANI-1 binds

preferentially to active RHO-1 via its RBD, as in other species (Piekny and Glotzer, 2008; Sun et al., 2015), and the linker region facilitates this interaction but does not bind active RHO-1 itself.

### ANI-1 tethers RHO-1 to linear structures after Formin$^{CYK-1}$ depletion but does not affect IT-RHO-1 levels or dynamics

Human anillin stabilizes RhoA on the membrane (Budnar et al., 2019; Piekny and Glotzer, 2008), and therefore, we tested whether ANI-1 depletion has any influence on RHO-1 membrane levels or dynamics. We employed an internal RHO-1 (IT-RHO-1) tagging strategy, which was developed in yeast and successfully applied in vertebrate embryos (Fig. S2 A) (Bendezú et al., 2015; Golding et al., 2019). Since homozygous *it-rho-1* worms were highly sterile (Fig. S2 B), we used embryos from heterozygous *rho-1/it-rho-1* hermaphrodites for all experiments. Direct comparison of the expression levels of endogenous RHO-1 and IT-RHO-1 was hindered by the presence of a non-specific band with similar molecular weight. Nevertheless, we found that endogenous RHO-1 levels in heterozygous animals (*rho-1/it-rho-1*) were only about 20% of those in control animals (*rho-1/rho-1*) (Fig. S2 C). To verify that IT-RHO-1 did not affect cytokinesis progression, we partially depleted endogenous RHO-1 by RNAi to 25% control levels (Fig. S2 D). This resulted in 80% cytokinesis failure, whereas no cytokinesis failure was observed in one-cell embryos derived from control (*rho-1/rho-1*) or heterozygous *it-rho-1* hermaphrodites (Fig. S2 E). Comparison of cytokinesis timing revealed no difference between controls and embryos expressing IT-RHO-1 (Fig. S2 F). During anaphase, IT-RHO-1 became enriched in a zone at the equatorial membrane as expected (Fig. 3 A and Video 3). Next, we tested whether its equatorial enrichment is controlled by the GEF ECT-2 (Canevascini et al., 2005) and GAPs RGA-3 and RGA-4 (RGA-3/4) (Schmutz et al., 2007). IT-RHO-1 was no longer localized at the cell equator after ECT-2 depletion, and in *rga-3/4(RNAi)* embryos, IT-RHO-1 levels were strongly increased throughout the membrane (Fig. S2, G–I). We conclude that IT-RHO-1 exhibits the expected localization dynamics: IT-RHO-1 is enriched at the cell equator during anaphase, and its membrane targeting is controlled by the GEF ECT-2 and GAPs RGA-3/4.

To test the role of ANI-1 in RHO-1 localization, we depleted ANI-1 in the IT-RHO-1 expressing strain and quantified its membrane accumulation during ring assembly. The equatorial levels of IT-RHO-1 were not altered after *ani-1(RNAi)* (Fig. 3, A–C

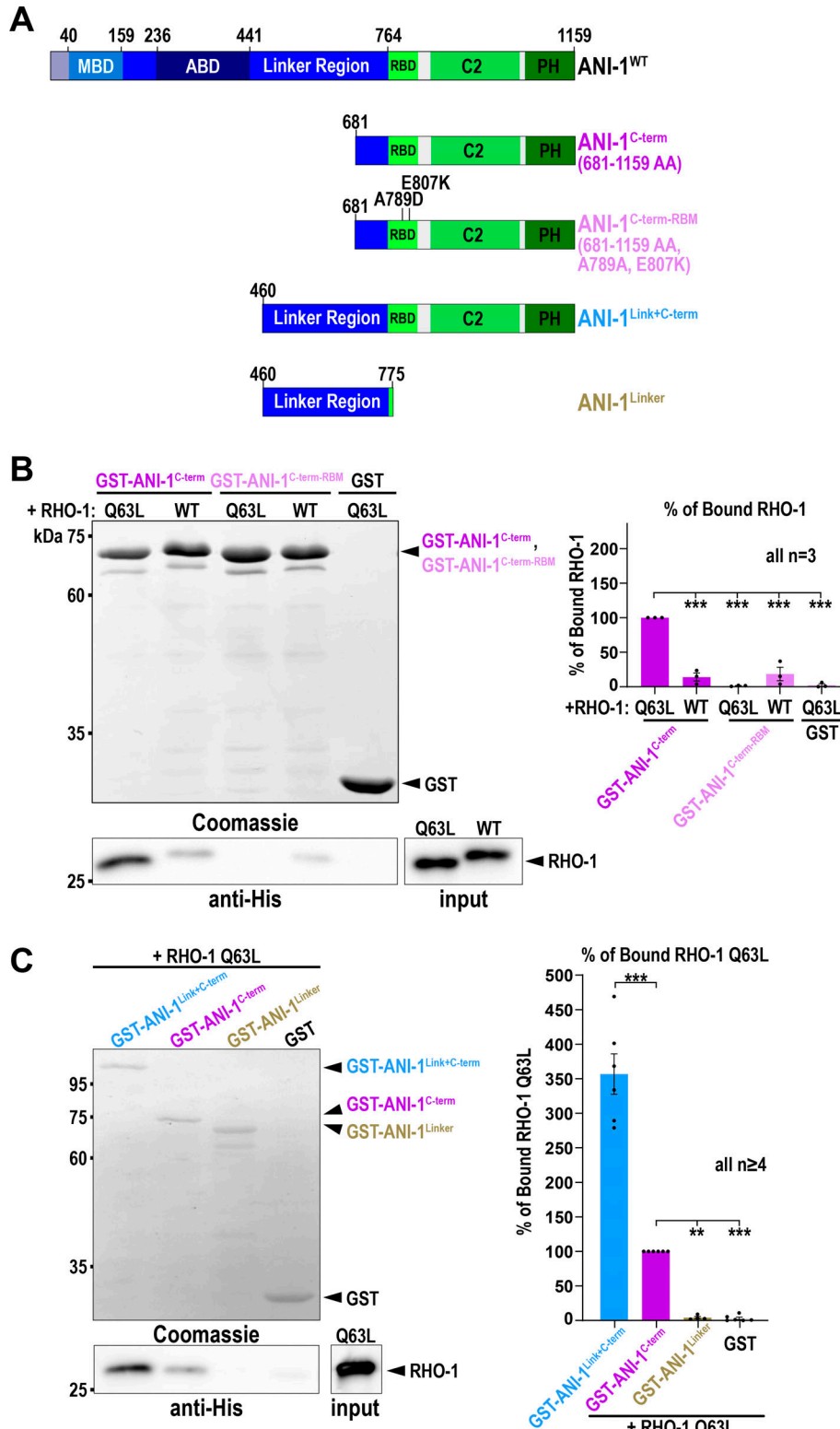

Figure 2. **The linker region of ANI-1 facilitates the binding of the RBD to active RHO-1. (A)** Domain organization of ANI-1 with predicted MBD, ABD, linker region, RBD, C2, and PH domain. The various GST-tagged ANI-1 C-terminal fragments used in the pull-down assay are shown. RBM - RHO-1 binding mutant (A789D, E807K). **(B)** His-tagged constitutive active RHO-1 (RHO-1$^{Q63L}$) or wild type RHO-1 (RHO-1$^{WT}$) were incubated with GST-tagged ANI-1$^{C-term}$, the ANI-1$^{C-term-RBM}$, or GST alone. Coomassie-stained gel of the pull-down assay shows the GST-tagged proteins and the immunoblot against the His-tag reveals the bound RHO-1 proteins (left). The mean percentages of bound RHO-1$^{WT}$ or RHO-1$^{Q63L}$ to the different GST-tagged fragments are shown. The amount of RHO-1$^{Q63L}$ pulled down with GST-tagged ANI-1$^{C-term}$ was set to 100% (right). **(C)** RHO-1$^{Q63L}$ was incubated with GST-tagged ANI-1$^{C-term}$, the ANI-1$^{Link+C-term}$, ANI-1$^{Linker}$, or GST alone. Coomassie-stained gel of the pull-down assay shows the GST-tagged proteins and the immunoblot against the His-tag reveals the bound RHO-1$^{Q63L}$ protein (left). The mean percentages of bound RHO-1$^{Q63L}$ to the different GST-tagged fragments are shown. The amount of RHO-1$^{Q63L}$ pulled down with GST-tagged ANI-1$^{C-term}$ was set to 100% (right). For all, P values were calculated using the two-tailed Student's $t$ test or Mann–Whitney U test and are **P < 0.01 and ***P < 0.001, error bars are SEM, and $n$ = number of experiments. Source data are available for this figure: SourceData F2.

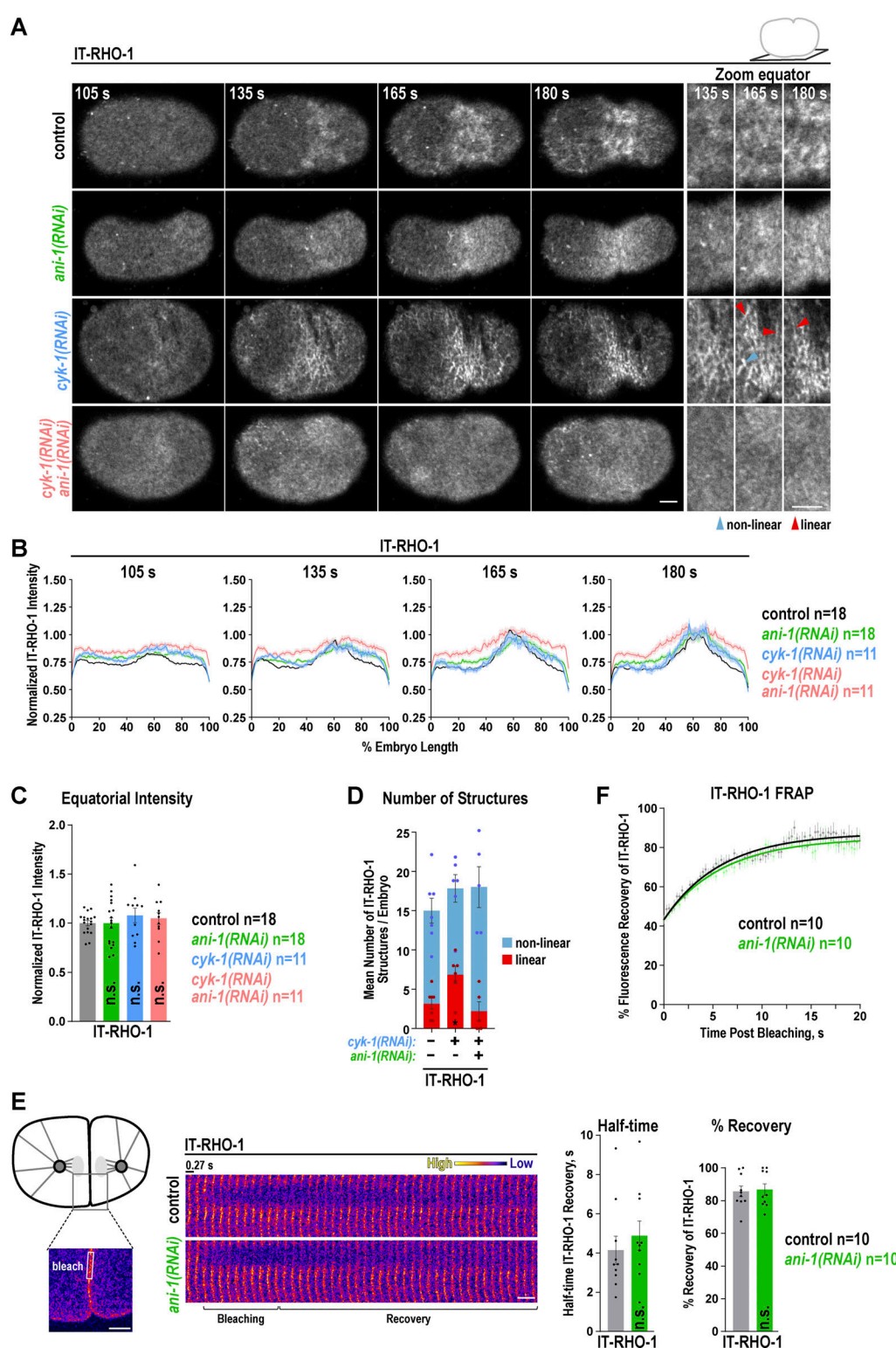

Figure 3. **ANI-1 tethers IT-RHO-1 to linear structures after Formin^CYK-1 depletion but does not affect IT-RHO-1 levels and dynamics. (A)** Single z-plane images of the cell cortex of one-cell *C. elegans* embryos expressing IT-RHO-1 treated with the indicated RNAi conditions. A zoom-in of the equatorial region for the indicated time points is shown on the right. Selected images are reproduced in Fig. S3 A to illustrate the quantification of IT-RHO-1 structures. Scale bars are 5 μm. **(B)** Normalized cortical fluorescence intensity of IT-RHO-1 from the anterior (0%) to the posterior (100%) pole for indicated RNAi conditions. **(C)** Mean normalized fluorescence intensity of IT-RHO-1 at the cell equator 180 s after NEBD for indicated RNAi conditions and dots represent data points of individual embryos. **(D)** Mean number of linear and nonlinear structures at the cell equator for the indicated RNAi conditions 180 s after NEBD. **(E and F)** IT-RHO-1 was bleached in a small region immediately after furrow ingression in control and *ani-1*(RNAi) treated embryos (E, left). Plotted are the mean fluorescence recovery in the bleached region (F) and recovery half time and the percentage of recovery for each embryo (E). For all, time in seconds (s) after NEBD is indicated, error bars are SEM, and *n* = number of embryos analyzed. P values were calculated using the two-tailed Student's *t* test or Mann–Whitney U test and are n.s. P > 0.05 and *P < 0.05.

and Video 3), suggesting that ANI-1 does not limit membrane accumulation of RHO-1. We had shown prior that the reduction of F-actin by Formin$^{CYK-1}$ depletion or Latrunculin A treatment caused ANI-1 to localize in linear structures (Lebedev et al., 2023). If ANI-1 binds RHO-1 in vivo, we expected RHO-1 to enrich in similar linear structures after reducing F-actin levels. Indeed, we observed that IT-RHO-1 formed linear structures after cyk-1(RNAi), and the number of those linear structures was reduced after co-depleting ANI-1 (Fig. 3, A and D; Fig. S3 A; and Video 3). Again, the overall equatorial IT-RHO-1 levels were not altered in cyk-1(RNAi) and cyk-1(RNAi) ani-1(RNAi) embryos in comparison with controls (Fig. 3, A–C). Similarly, the addition of Latrunculin A caused the appearance of circular and linear IT-RHO-1 structures on the membrane (Fig. S3, B and C). The depletion of ANI-1 by RNAi and subsequent Latrunculin A treatment caused a strong reduction in the number of IT-RHO-1 structures, but equatorial IT-RHO-1 levels were not reduced (Fig. S3, B–E).

Since we did not observe a change in the cortical levels of IT-RHO-1, we analyzed whether its dynamics are affected by ANI-1. For this, we performed fluorescence recovery after photobleaching (FRAP) of IT-RHO-1 just after contractile ring closure on the newly ingressed membranes separating the two daughter cells. Consistent with previous observations (Budnar et al., 2019), IT-RHO-1 was highly dynamic with a half-time recovery ($t_{1/2}$) of ~4.1 s and a mobile fraction of ~80% (Fig. 3, E and F). Depletion of ANI-1 did not change the recovery rate or the mobile fraction of IT-RHO-1 (Fig. 3, E and F).

Together, ANI-1 tethers IT-RHO-1 to cortical structures after the reduction of F-actin in C. elegans embryos in vivo. However, ANI-1 does not influence RHO-1 levels or dynamics on the membrane, and therefore, we hypothesize that ANI-1 binding to RHO-1 limits RHO-1 activity directly.

### Binding of the ANI-1$^{C-term}$ to RHO-1 reduces cortical NMY-2 levels during ring assembly

Since ANI-1 binds RHO-1 via the RBD and this interaction surface corresponds to the site where effectors bind (Dvorsky and Ahmadian, 2004; Sun et al., 2015), we hypothesized that ANI-1 might block the RHO-1-effector binding site. To test this, we checked whether NMY-2 levels increased or not when we replaced full-length ANI-1 with the RBD. Since the human RBD requires the PH and C2 domains to efficiently localize to the membrane (Sun et al., 2015), we expressed the entire C-terminus of ANI-1 using a previously established genetic replacement system (Fig. 4 A, [Lebedev et al., 2023]). In this system, the ANI-1 transgenes are GFP-tagged and resistant against ani-1(RNAi), which enables us to analyze their phenotypes and localization after depleting endogenous ANI-1 by RNAi. As a control, we expressed wild type ANI-1 (GFP::ANI-1$^{WT}$), which fully rescued embryonic lethality induced by ani-1(RNAi) (Fig. S4 B). In addition to GFP::ANI-1$^{C-term}$, we used an ANI-1 construct that included the MBD, ABD, and linker region (GFP::ANI-1$^{N-term+Link}$) and did not bind RHO-1. The cortical levels of the GFP::ANI-1$^{C-term}$ and GFP::ANI-1$^{N-term+Link}$ were similar to each other but reduced in comparison with GFP::ANI-1$^{WT}$ (Fig. S4, C and D). Embryos depleted of endogenous ANI-1 and expressing

GFP::ANI-1$^{WT}$ or GFP::ANI-1$^{C-term}$ presented NMY-2::mKate levels close to those observed in control embryos expressing no transgene (Fig. 4, B–D and Video 4). In contrast, embryos depleted of endogenous ANI-1 and expressing GFP::ANI-1$^{N-term+Link}$ had higher levels of equatorial NMY-2::mKate. This indicates that binding of the ANI-1 RBD to RHO-1 prevents the increase in NMY-2 levels. To confirm this, we abolished RHO-1 binding by introducing the A789D and E807K mutation in the C-terminal transgene (GFP::ANI-1$^{C-term-RBM}$). Although the expression levels of GFP::ANI-1$^{C-term-RBM}$ were even higher than those of GFP::ANI-1$^{WT}$ (Fig. S4 E), no cortical accumulation of GFP::ANI-1$^{C-term-RBM}$ was observed (Fig. S4, A, C, and D), suggesting that RHO-1 binding was successfully abolished. This observation confirms previous studies which found that membrane localization of the anillin C-terminus was lost when the RhoA binding site was mutated (Carim and Hickson, 2023; Sun et al., 2015). Depletion of endogenous ANI-1 by RNAi caused high embryonic lethality, which could be partially rescued by expression of GFP::ANI-1$^{C-term}$ (Lebedev et al., 2023). Consistent with the fact that GFP::ANI-1$^{C-term-RBM}$ did not enrich at the cell equator, embryonic lethality and cortical NMY-2::mKate levels were not rescued by GFP::ANI-1$^{C-term-RBM}$ expression (Fig. 4, B–D and Fig. S4 B). Since the linker region enhanced RHO-1 binding of the C-terminus in vitro, we wondered whether extending the C-terminus and including the linker region (GFP::ANI-1$^{Link+C-term}$, Fig. 4 A) had any effect on NMY-2::mKate levels. We observed that NMY-2::mKate levels were reduced to a similar extent in GFP::ANI-1$^{Link+C-term}$ and GFP::ANI-1$^{C-term}$ expressing embryos (Fig. 4, B–D and Video 4).

Together, these data suggest that the ANI-1 C-terminus binds to RHO-1 via the RBD and that this blocks the effector binding site of RHO-1, thereby potentially limiting the activation of RHO-1 effectors and RHO-1 signaling.

### ANI-1 restricts the accumulation of RhoA effectors but not IT-RHO-1 in the constricting ring

Anillin could facilitate an asymmetry of RHO-1 activity in the ring by inhibiting RHO-1 signaling at the lagging edge or promoting it at the leading edge. If ANI-1 inhibits RHO-1 signaling at the lagging edge, RHO-1 effector levels should increase after ani-1(RNAi). If ANI-1 promotes RHO-1 signaling at the leading edge, depletion of ANI-1 should cause a decrease in RHO-1 effector levels. To check whether the levels of RHO-1 downstream targets decrease or increase after ani-1(RNAi) in the constricting ring, we analyzed Formin$^{CYK-1}$, ROK$^{LET-502}$, and NMY-2. We filmed embryos during furrow ingression and generated an "end-on" view of the division plane (Fig. S5 A). GFP::ROK$^{LET-502}$ levels in the ring were very low (Fig. S5 B) and therefore it was not included in the analysis. As previously published, after ani-1(RNAi), the total levels of NMY-2::mKate in the constricting ring were elevated (Fig. 5, A and B; and Video 5) (Hsu et al., 2023; Maddox et al., 2007). The Formin$^{CYK-1}$ was weakly visible in the ring and after ANI-1 depletion the total levels of the Formin$^{CYK-1}$ were also increased (Fig. 5, B and C; and Video 6). Consistent with increased formin intensity and activity, F-actin levels in the ring were also elevated after ani-1(RNAi) (Jordan et al., 2016). ANI-1 could inhibit RHO-1 signaling by destabilizing

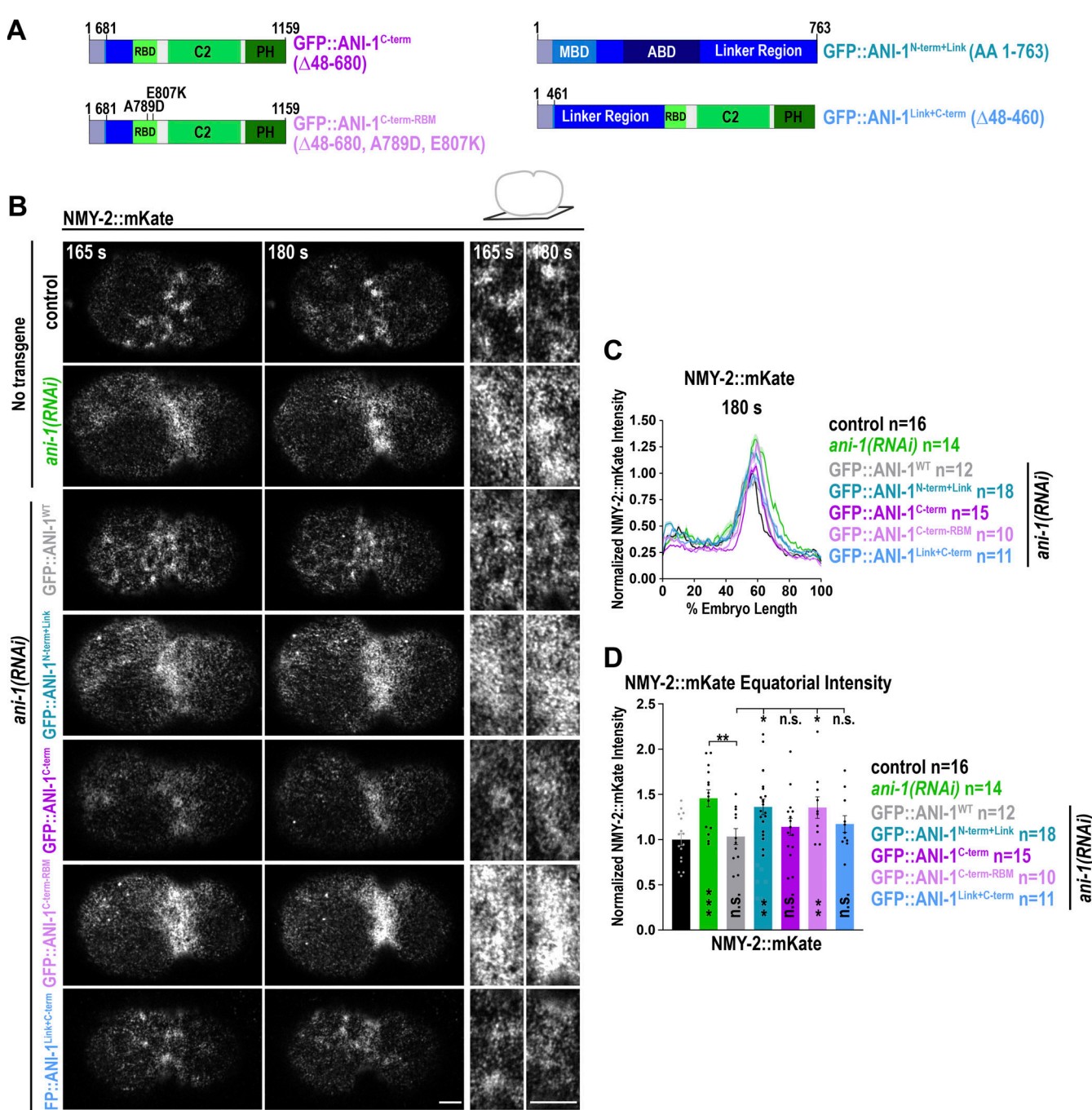

Figure 4. **ANI-1 binding to RHO-1 via the RBD limits cortical NMY-2::mKate accumulation during ring assembly. (A)** Schematic representation of the used GFP-tagged ANI-1 mutant variants. **(B)** Single z-plane NMY-2::mKate images of the cell cortex of embryos expressing indicated GFP::ANI-1 transgenes and treated with or without *ani-1(RNAi)*. A zoom-in of the equatorial region for the indicated time points is shown on the right. Scale bars are 5 μm. **(C)** Normalized cortical fluorescence intensity of NMY-2::mKate from the anterior (0%) to the posterior (100%) pole for control and *ani-1(RNAi)* embryos expressing indicated GFP-tagged ANI-1 variants at 180 s after NEBD. **(D)** Mean normalized fluorescence intensity of NMY-2::mKate at the cell equator for indicated conditions at 180 s after NEBD. Error bars are SEM and P values were calculated using two-tailed Student's *t* test and are n.s. P > 0.05, *P < 0.05, **P < 0.01, and ***P < 0.001. For all, time in seconds (s) after NEBD is indicated.

RHO-1 on the membrane or by blocking the effector binding site of RHO-1. If ANI-1 destabilizes RHO-1, the levels of RHO-1 are expected to increase after ANI-1 depletion. IT-RHO-1 was enriched in the ring and after *ani-1(RNAi)* IT-RHO-1 levels in the ring were not changed (Fig. 5, B and D; and Video 7).

We conclude that after *ani-1(RNAi)*, the total amount of RHO-1 effectors, but not IT-RHO-1, increased in the ring. These observations are consistent with a model where ANI-1 inhibits RHO-1 activity not only during ring assembly but also during its constriction.

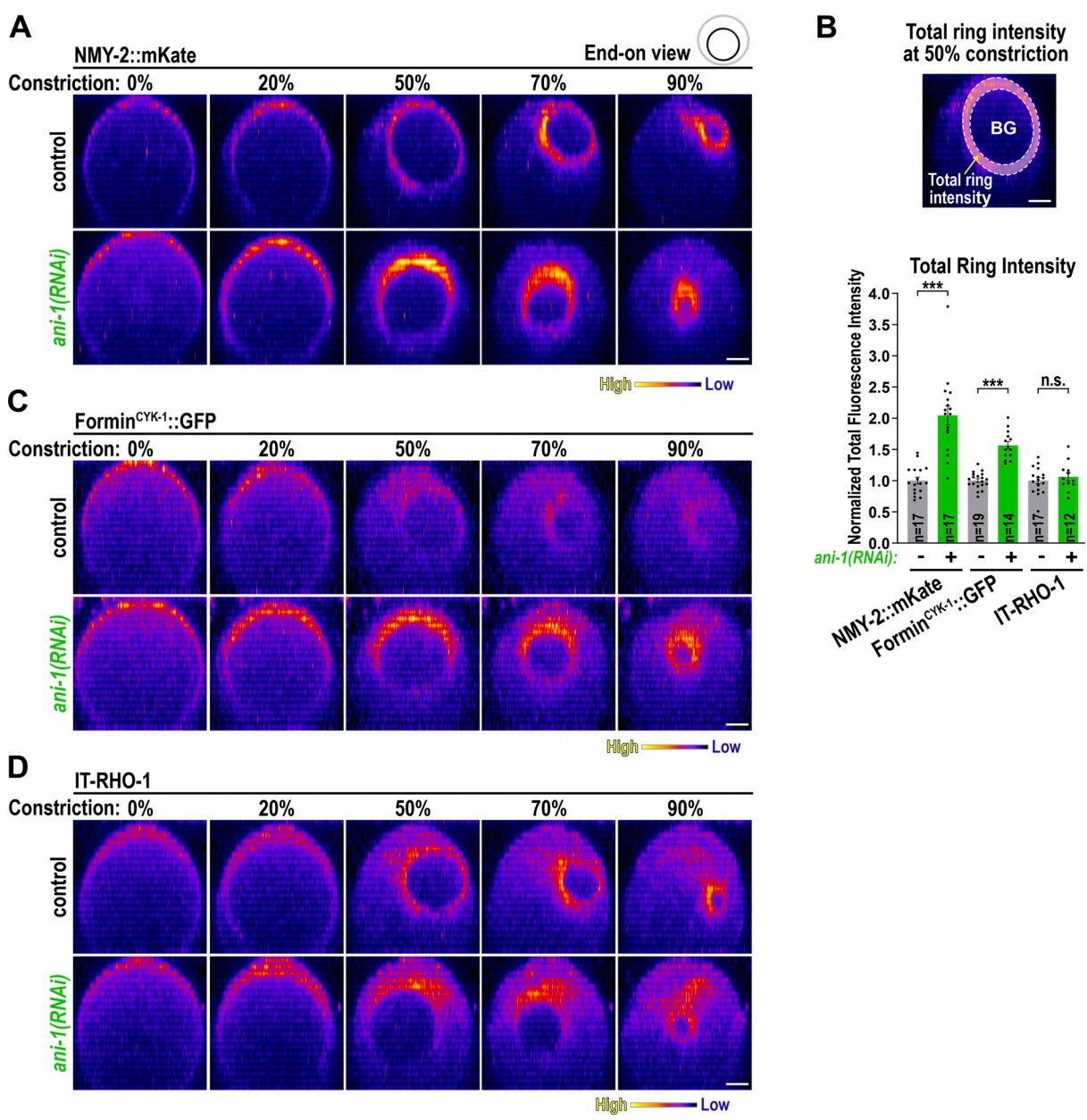

Figure 5. **ANI-1 limits the accumulation of NMY-2::mKate and Formin^CYK-1::GFP but not IT-RHO-1 in the ring. (A)** End-on reconstruction of NMY-2::mKate at the cleavage plane for indicated % of ring constriction with and without *ani-1(RNAi)*. **(B)** Total ring intensity was measured at 50% constriction after the background (BG) intensity was subtracted (*top*). Total ring intensity of NMY-2::mKate, Formin^CYK-1::GFP, and IT-RHO-1 expressing embryos treated with or without *ani-1(RNAi)* (*bottom*). P values were calculated using two-tailed Student's *t* test and are n.s. P > 0.05 and ***P < 0.001, *n* = number of embryos, and error bars are SEM. **(C)** End-on reconstruction of Formin^CYK-1::GFP at the cleavage plane for indicated % of ring constriction of embryos with or without *ani-1(RNAi)*. **(D)** End-on reconstruction of IT-RHO-1 at the cleavage plane for indicated % of ring constriction of embryos with or without *ani-1(RNAi)*. All scale bars are 5 μm.

## Asymmetric NMY-2 localization in the constricting ring requires the linker region and the C-terminus of ANI-1

Since the ANI-1 C-terminus suppressed NMY-2 hyper-accumulation during ring assembly, we analyzed whether it also restored NMY-2 levels and its asymmetric distribution in the ring. After *ani-1(RNAi)*, the total NMY-2::mKate levels in the ring of GFP::ANI-1^WT and GFP::ANI-1^C-term expressing embryos were similar to one other and reduced in comparison to *ani-1(RNAi)*-treated control embryos without transgene (Fig. S5, C and D). However, total NMY-2::mKate levels in the GFP::ANI-1^WT

embryos were still higher in comparison to non-depleted control embryos without transgene (Fig. S5 D). This suggests that the GFP-tag might partially compromise ANI-1 function during ring constriction although other characteristic *ani-1(RNAi)* phenotypes were fully rescued (Fig. S4 B, [Lebedev et al., 2023]). Consistent with that, embryos expressing ANI-1 tagged with similar-sized mNeonGreen (NG::ANI-1) also exhibited increased total levels of NMY-2::mKate during ring constriction in comparison with control embryos (Fig. S5 D). To overcome this problem, we tagged full-length ANI-1 with a smaller FLAG-tag

(FLAG::ANI-1$^{WT}$). Immunoblotting of animals revealed that expression levels of FLAG::ANI-1$^{WT}$ were comparable with those of in situ–tagged NG::ANI-1, which also harbors a FLAG tag (Fig. S4 E). Measurements in the ring of embryos expressing FLAG::ANI-1$^{WT}$ revealed that NMY-2::mKate levels were indistinguishable from embryos without transgene, suggesting that FLAG-tagged ANI-1$^{WT}$ was fully functional (Fig. 6, A and B; and Video 8). Therefore, we also generated a FLAG::ANI-1$^{C-term}$ expressing strain. Immunoblotting of these worms showed that the FLAG::ANI-1$^{C-term}$ was well expressed and largely supported embryonic development (Fig. S4, B and E). The expression of the FLAG::ANI-1$^{C-term}$ prevented hyperaccumulation of NMY-2::mKate in the ring and exhibited similar levels to FLAG::ANI-1$^{WT}$ embryos (Fig. 6, A and B; and Video 8). This shows that the C-terminus is sufficient to prevent the increase of NMY-2 levels not only during ring assembly but also during ring constriction.

NMY-2 is enriched at the leading edge, and since ANI-1 depletion increased total NMY-2 levels in the ring, we asked whether this increase was equal on the leading and lagging edge of the ring. Plotting the fluorescence intensity of NMY-2::mKate separately at the leading and lagging edge revealed that NMY-2 levels were elevated on both sides of the ring after *ani-1(RNAi)*, but the fold increase was more pronounced at the lagging edge (∼2.4-fold increase) than the leading edge (∼1.8 fold) (Fig. 6, C and D; and Fig. S5 A). Thus, these data indicate that ANI-1 inhibits RHO-1 signaling at the leading and lagging edge but the attenuation of RHO-1 signaling by ANI-1 is stronger at the lagging edge.

Since the ANI-1$^{C-term}$ fragment was sufficient to lower total NMY-2 levels in the ring, we asked whether it also restored its levels at the leading and lagging edge. The total NMY-2::mKate levels at the leading edge in the FLAG::ANI-1$^{C-term}$ expressing embryos were indistinguishable from those in FLAG::ANI-1$^{WT}$ embryos (Fig. 6 D). At the lagging edge, however, the NMY-2::mKate levels were also reduced but still elevated in comparison with those in FLAG::ANI-1$^{WT}$ expressing embryos (Fig. 6 D). In control embryos, NMY-2::mKate was ∼1.5-fold enriched at the leading edge in comparison with the lagging edge, and this asymmetric enrichment was lost after ANI-1 depletion (Fig. 6 E) (Maddox et al., 2007). Expression of a GFP-tagged or FLAG-tagged ANI-1$^{WT}$ transgene restored the asymmetric distribution of NMY-2::mKate in the ring (Fig. 6 E and Fig. S5 E). Consistent with the observation that the expression of FLAG::ANI-1$^{C-term}$ did not fully suppress NMY-2::mKate accumulation at the lagging edge, NMY-2 asymmetry was also not restored in the ring (Fig. 6 E, Fig. S5 E, and Video 5).

This suggests that the presence of certain regions of the ANI-1 N-terminal part is required for ANI-1 function, particularly at the lagging edge. Since the linker region enhanced the binding of the RBD to active RHO-1, we asked whether including the linker region in the C-terminal construct (ANI-1$^{Link+C-term}$) would be sufficient to restore ANI-1 function. Indeed FLAG-tagged ANI-1$^{Link+C-term}$ restored NMY-2::mKate levels at the leading and lagging edge as well as NMY-2::mKate asymmetry in the ring, similar to the FLAG::ANI-1$^{WT}$ (Fig. 6, A–E, Fig. S5, C–E, and Video 8).

Since the attachment of the linker region to the C-terminus rescued NMY-2 asymmetry, we tested the influence of the linker region employing two different transgenes. First, we used a GFP-tagged (Lebedev et al., 2023) and a FLAG-tagged ANI-1 N-termini expressing strain, which included the linker region and the ABD and MBD (GFP- or FLAG-tagged ANI-1$^{N-term+Link}$). Second, we utilized a transgene where the linker region alone was tethered to the plasma membrane with a CAAX motif (Fig. S5 F, GFP::ANI-1$^{Linker-CX}$ [Lebedev et al., 2023]). Immunoblotting of FLAG::ANI-1$^{N-term+Link}$ animals exhibited slightly reduced protein levels in comparison with FLAG::ANI-1$^{WT}$ but embryonic development was better supported by the expression of FLAG::ANI-1$^{N-term+Link}$ than the FLAG::ANI-1$^{C-term}$ (Fig. S4 B). The GFP- or FLAG-tagged ANI-1$^{N-term+Link}$ and the membrane-attached GFP::ANI-1$^{Linker-CX}$ did not rescue the total NMY-2::mKate levels or its asymmetric distribution in the ring (Fig. 6, A–E and Fig. S5, C–E). Last, we tested whether the coexpression of the linker containing N-terminus with the C-terminus in the same embryo but as two different proteins (FLAG::ANI-1$^{C-term}$ + GFP::ANI-1$^{N-term+Link}$) would rescue NMY-2 levels and asymmetry in the ring. In the FLAG::ANI-1$^{C-term}$ + GFP::ANI-1$^{N-term+Link}$ coexpressing animals, embryonic lethality after *ani-1(RNAi)* was still elevated in comparison with full-length ANI-1$^{WT}$ expression animals (Fig. S4 B). Furthermore, FLAG::ANI-1$^{C-term}$ + GFP::ANI-1$^{N-term+Link}$ coexpressing embryos exhibited intermediate total NMY-2::mKate levels and no asymmetric NMY-2::mKate distribution in the ring (Fig. 6, A–E), suggesting that just the presence of the linker in the same embryos is not sufficient to restore NMY-2 asymmetry. One caveat is that the levels of GFP::ANI-1$^{N-term+Link}$ and GFP::ANI-1$^{Linker-CX}$ in the ring were reduced in comparison with those of GFP::ANI-1$^{WT}$ (see below), and therefore protein levels may be insufficient to restore NMY-2 levels and asymmetry.

In summary, although the RBD-containing ANI-1 C-terminus was sufficient to lower total NMY-2 levels at the leading edge, it did not fully restore NMY-2 levels at the lagging edge of the ring. Only when the linker region was fused to the C-terminus, NMY-2 levels at the lagging and leading edge were completely restored, and NMY-2 was asymmetrically localized in the constricting ring.

### The linker region and the C-terminus are both required for enrichment of ANI-1 at the leading edge of the ring

During furrow ingression, ANI-1 is present in the ring and enriched at the leading edge (Maddox et al., 2007). To understand which region of ANI-1 is required for the enrichment in the ring and the asymmetric localization within the ring, we analyzed the localization of the different GFP-tagged ANI-1 variants. The GFP::ANI-1$^{C-term}$ and GFP::ANI-1$^{Link+C-term}$ were present in similar total levels in the ring as GFP::ANI-1$^{WT}$ (Fig. 7, A and B). Analysis of ANI-1 accumulation at the leading and lagging edge revealed that the asymmetric distribution of GFP::ANI-1$^{C-term}$ was attenuated in comparison with GFP::ANI-1$^{WT}$ (Fig. 7 C). In contrast, GFP::ANI-1$^{Link+C-term}$ exhibited an asymmetric enrichment like GFP::ANI-1$^{WT}$. One possibility is that the linker region harbors a targeting sequence that enriches ANI-1 at the leading edge. To test this, we analyzed the localization of the linker

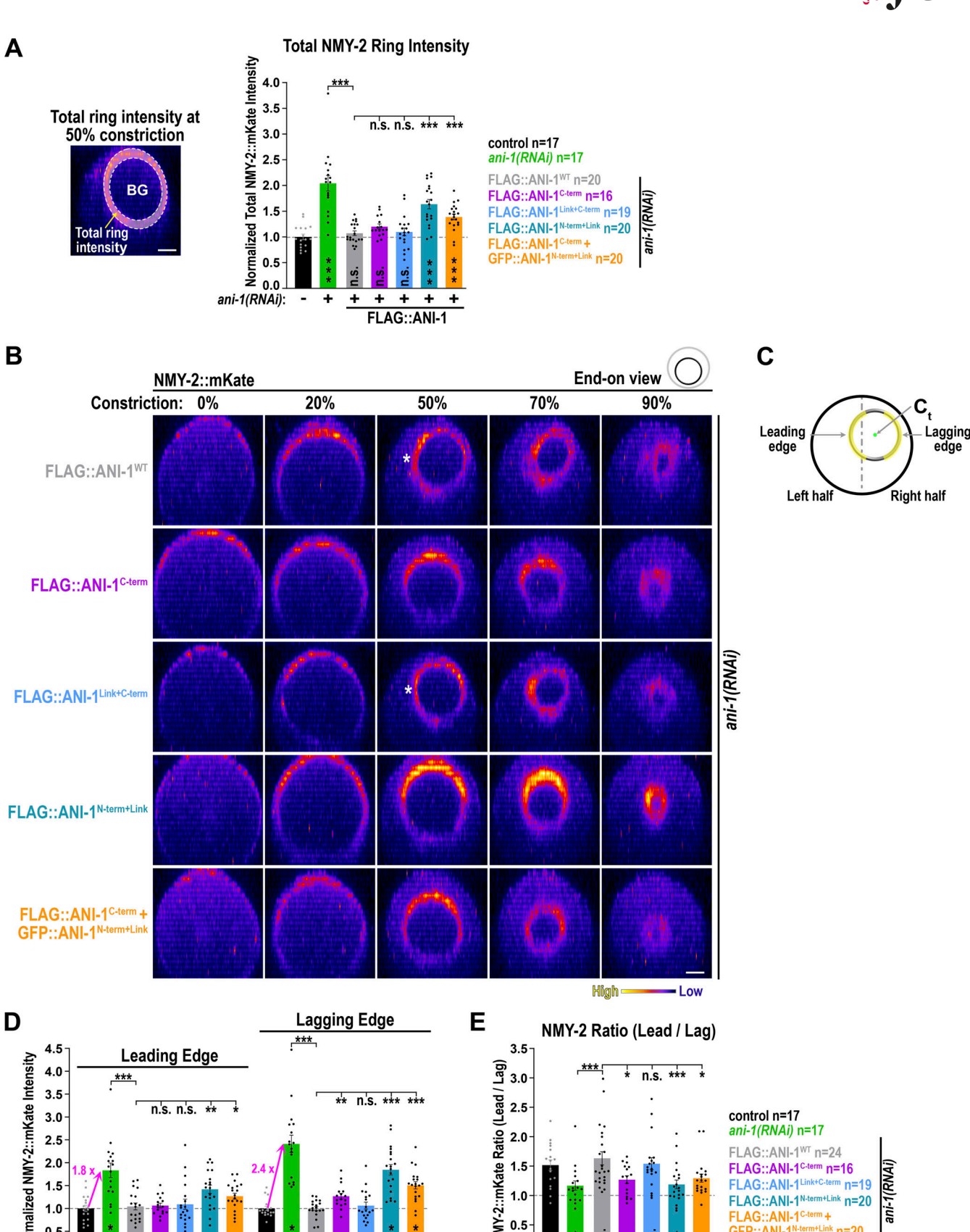

**A**

**Total NMY-2 Ring Intensity**

Total ring intensity at 50% constriction

control n=17
*ani-1(RNAi)* n=17
FLAG::ANI-1^WT n=20
FLAG::ANI-1^C-term n=16
FLAG::ANI-1^Link+C-term n=19
FLAG::ANI-1^N-term+Link n=20
FLAG::ANI-1^C-term + GFP::ANI-1^N-term+Link n=20

**B** NMY-2::mKate — End-on view

Constriction: 0% 20% 50% 70% 90%

FLAG::ANI-1^WT
FLAG::ANI-1^C-term
FLAG::ANI-1^Link+C-term
FLAG::ANI-1^N-term+Link
FLAG::ANI-1^C-term + GFP::ANI-1^N-term+Link

*ani-1(RNAi)*

High — Low

**C**

Leading edge — $C_t$ — Lagging edge

Left half — Right half

**D**

Leading Edge      Lagging Edge

1.8 x      2.4 x

**E** NMY-2 Ratio (Lead / Lag)

control n=17
*ani-1(RNAi)* n=17
FLAG::ANI-1^WT n=24
FLAG::ANI-1^C-term n=16
FLAG::ANI-1^Link+C-term n=19
FLAG::ANI-1^N-term+Link n=20
FLAG::ANI-1^C-term + GFP::ANI-1^N-term+Link n=20

**Figure 6.** **The linker region and the C-terminus of ANI-1 are both required to restore NMY-2 asymmetry in the ring. (A)** Total ring intensity of NMY-2::mKate in control and the indicated FLAG::ANI-1 expressing embryos at 50% constriction for indicated RNAi conditions. **(B)** End-on reconstruction of NMY-2::mKate at the cleavage plane for FLAG::ANI-1 expressing embryos at indicated % of ring constriction treated with *ani-1(RNAi)*. White stars indicate the position of the leading edge. Scale bars are 5 μm. **(C)** Since fluorescence intensity is attenuated with increasing distance from the objective (Fig. S5 A), we quantified the leading and lagging edge fluorescence intensities on the left and right sides of the ring. To define the leading and the lagging edge position of cytokinetic center ($C_t$) was assigned to the left or right half of the cleavage plane. The ring edge that was in the same half as the cytokinetic center was defined as the lagging edge. The fluorescence intensity was summed up at the leading or lagging edge (yellow) excluding the top and bottom regions (grey) of the ring. **(D and E)** Fluorescence intensity of NMY-2::mKate at the leading and the lagging edge (D) and the ratio of NMY-2::mKate intensities of the leading and lagging edge (E) for the indicated FLAG::ANI-1 expressing embryos at 50% ring constriction for indicated RNAi conditions. The fold increase of NMY-2::mKate at the leading and lagging edge after *ani-1(RNAi)* is highlighted with pink arrows. For all: P values were calculated using two-tailed Student's *t* test or Mann–Whitney U test and are n.s. P > 0.05, *P < 0.05, **P < 0.01 and ***P < 0.001, *n* = number of embryos, and error bars are SEM.

region alone tethered to the plasma membrane (GFP::ANI-1$^{Linker-CX}$), and the N-terminal construct, which includes the linker region (GFP::ANI-1$^{N-term+Link}$). The total levels of GFP::ANI-1$^{Linker-CX}$ and GFP::ANI-1$^{N-term+Link}$ in the ring were reduced in comparison with GFP::ANI-1$^{WT}$ (Fig. 7, A and B). This is different from our observations made during ring assembly where the levels of GFP::ANI-1$^{C-term}$ and GFP::ANI-1$^{N-term+Link}$ were equal to each other but reduced in comparison with GFP::ANI-1$^{WT}$ (Fig. S4 D). Since neither the GFP::ANI-1$^{Linker-CX}$ nor the GFP::ANI-1$^{N-term+Link}$ transgene expression restored unilateral furrow ingression (see below) after *ani-1(RNAi)*, we also analyzed their localization in non-depleted embryos with normal unilateral ingression. For both conditions, we did not observe an asymmetric enrichment of GFP::ANI-1$^{Linker-CX}$ or GFP::ANI-1$^{N-term+Link}$ in the ring (Fig. 7 C), suggesting that the linker region on its own is insufficient for enriching ANI-1 at the leading edge. Finally, we tested whether the co-expression of the FLAG::ANI-1$^{C-term}$ and GFP::ANI-1$^{N-term+Link}$ in the same embryos improved the total levels and asymmetric distribution of GFP::ANI-1$^{N-term+Link}$. We found that the total levels of GFP::ANI-1$^{N-term+Link}$ in the ring were still low and symmetrically distributed in the ring (Fig. 7, A–C).

In summary, the C-terminus is primarily responsible for targeting ANI-1 to the ring but it needs the linker region for asymmetric localization. The linker on its own is insufficient to enrich at the leading edge of the ring. Surprisingly, the predicted myosin II or F-actin binding regions of ANI-1 are largely dispensable to target ANI-1 to the contractile ring.

### Unilateral furrow ingression requires the ANI-1 linker region and C-terminus

Since NMY-2::mKate asymmetry within the ring was not restored by the ANI-1$^{C-term}$, we tested whether ring closure occurred in a symmetric manner within the division plane. To quantify the eccentricity of ring constriction, we measured the distance ($Q_t$) between the ring centroid before ($C_0$) and during ($C_t$) ring closure (Fig. 8 A). In *ani-1(RNAi)* embryos expressing NMY-2::mKate but not ANI-1 transgene, the peak eccentricity and eccentricity during ring closure were strongly reduced in comparison with controls (Fig. 8, A and B) (Hsu et al., 2023; Maddox et al., 2007). Embryos expressing GFP- or FLAG-tagged ANI-1$^{WT}$ exhibited increased eccentricity of furrow closure in comparison with *ani-1(RNAi)* embryos without the transgene (Fig. 8, A and B). In contrast, eccentricity was not restored in ANI-1$^{N-term+Link}$ or ANI-1$^{C-term}$ expressing embryos treated with

*ani-1(RNAi)* (Fig. 8, A and B). In control embryos, the time difference between the leading and lagging edge to reach 10% edge progression was ~100 s, and in *ani-1(RNAi)*, this time difference was strongly reduced (Fig. 8, C and D). Again, the lag time between the leading and lagging edge ingression initiation in ANI-1$^{N-term+Link}$ or ANI-1$^{C-term}$ embryos was lower in comparison with that in ANI-1$^{WT}$ embryos (Fig. 8 D), suggesting that the N- and C-terminal ANI-1 halves alone are insufficient to restore unilateral furrowing. Remarkably, the expression of ANI-1$^{Link+C-term}$ fully rescued all *ani-1(RNAi)* phenotypes exhibiting a high eccentricity over time and a high peak eccentricity of furrow ingression. Additionally, the lag time between the leading and lagging edge of ANI-1$^{Link+C-term}$ embryos was similar to the one observed in embryos expressing ANI-1$^{WT}$ (Fig. 8 D). Finally, we analyzed whether the linker region alone (GFP::ANI-1$^{Linker-CX}$) or the presence of the ANI-1$^{N-term+Link}$ and ANI-1$^{C-term}$ as separate proteins in the same embryo (FLAG::ANI-1$^{C-term}$ + GFP::ANI-1$^{N-term+Link}$) improved asymmetric furrowing. Neither the GFP::ANI-1$^{Linker-CX}$ nor the FLAG::ANI-1$^{C-term}$ + GFP::ANI-1$^{N-term+Link}$ coexpressing embryos exhibited unilateral furrow ingression, a caveat being that the levels of the GFP::ANI-1$^{Linker-CX}$ and the GFP::ANI-1$^{N-term+Link}$ were reduced in comparison with ANI-1$^{WT}$ in the ring.

Together, this shows that when the linker region and C-terminus are together in the same protein, they can mediate unilateral furrow ingression. Surprisingly, the predicted myosin II or F-actin binding regions of ANI-1 are not required for unilateral furrowing.

### Actomyosin flows and compression are stronger at the leading cortex than at the lagging cortex

Our data collectively indicate that the ANI-1$^{Link+C-term}$ suppresses RHO-1 activity more strongly at the lagging edge than at the leading edge. Since cortical flows have been proposed to drive unilateral ring closure (Hsu et al., 2023), we examined the differences in cortical flow speeds and compression at the cortices of the leading and lagging furrow edges. To map and measure cortical flows, we performed particle image velocimetry (PIV) analysis of cortical flows in unconfined embryos expressing in GFP::NMY-2. Polar flow velocities along the anterior–posterior (A–P) axis were quantified from the anterior to the posterior pole of the embryo (Fig. 9, A and B). At the leading cortex, we observed increasing bidirectional flows from the anterior pole (positive values) and the posterior poles (negative values) toward the furrow region over time (Fig. 9, C and D), as previously published (Khaliullin et al., 2018). In

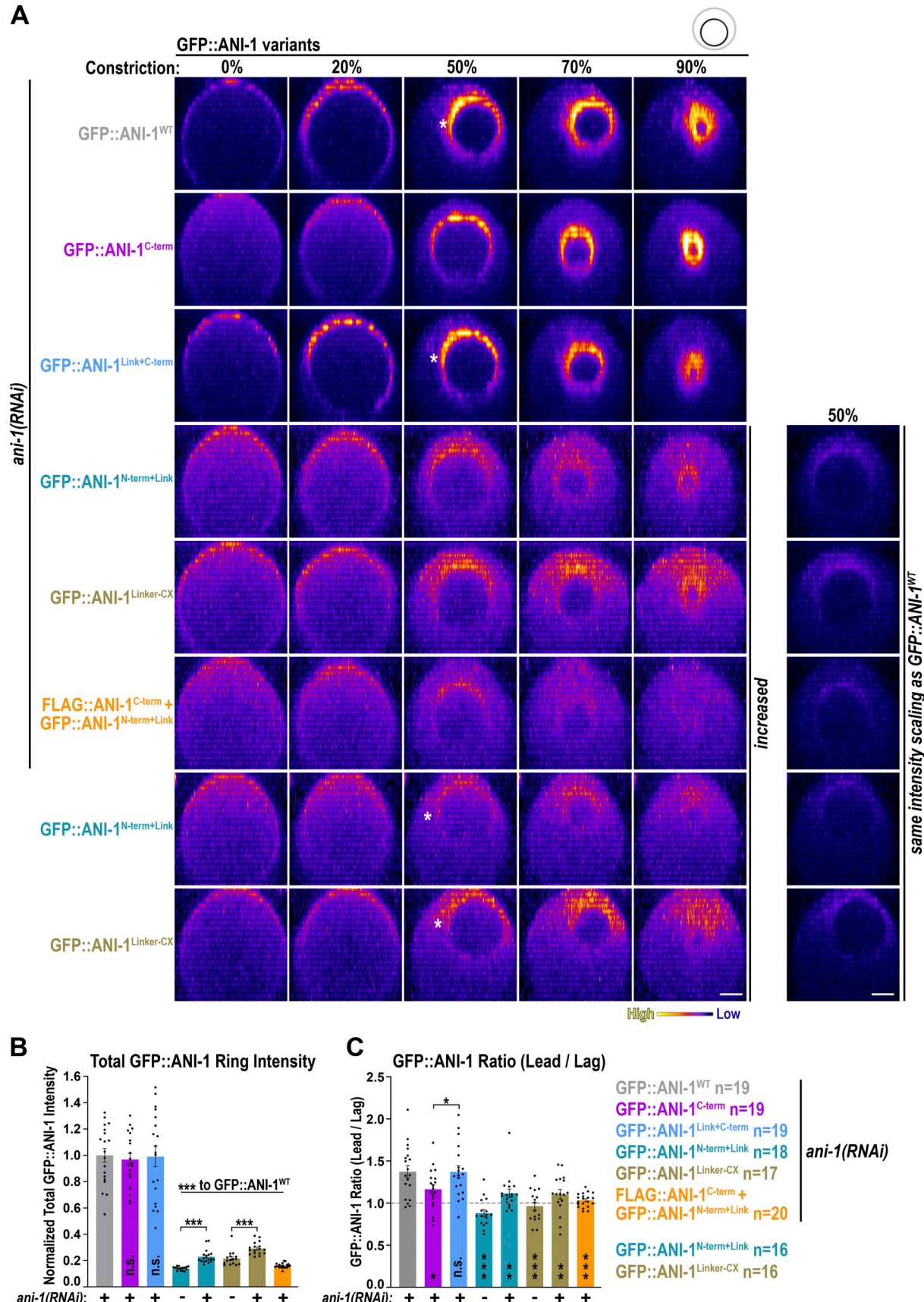

Figure 7.   **The linker region is not targeted to the leading edge but if connected with the C-terminus they localize asymmetrically in the ring. (A)** End-on reconstruction of indicated GFP::ANI-1 variants at the cleavage plane for highlighted % of ring constriction treated with or without *ani-1(RNAi)*. Scale bars is

5 µm. Since the localization of indicated GFP::ANI-1 variants to the ring is very low their intensity scaling was increased. For comparison the panels on the right have the same intensity scaling as for GFP::ANI-1$^{WT}$ at 50% constriction. White stars indicate the position of the leading edge. **(B and C)** Total GFP::ANI-1 intensity in the ring (B) and ratio of the leading and lagging edges of GFP::ANI-1 (C) for different ANI-1 variants were calculated at 50% ring constriction. P values were calculated using two-tailed Student's *t* test or Mann–Whitney U test and are n.s. P > 0.05, *P < 0.05, **P < 0.01 and ***P < 0.001, *n* = number of embryos, error bars are SEM.

comparison, the A-P flow velocities were overall much lower at the lagging cortex than the leading cortex at the same time points (Fig. 9, D and E). In particular, at the anterior region of the lagging cortex, the flow velocity values were close to zero, indicating that the bidirectional flow was very weak. This

demonstrates that strong furrow-directed bidirectional flows emerge first at the leading edge and confirms previous observations (Khaliullin et al., 2018).

The convergence of bidirectional flows is expected to cause a compression of the actin cortex and since bidirectional A-P flows

Figure 8. **Unilateral furrowing requires the linker region and the C-terminus of ANI-1. (A)** Eccentricity of ring closure was calculated as shown in schematics (*left*) and plotted for different percentages of ring constriction for the indicated conditions (*right*). **(B)** Mean peak eccentricity of the ring during constriction for the indicated control, and GFP::ANI-1 or FLAG::ANI-1 expressing embryos with and without *ani-1*(*RNAi*). **(C)** Leading and lagging edge progression over time for control and *ani-1*(*RNAi*) treated embryos. Time zero represents the onset of furrow ingression. The time difference when the leading and lagging edge reach 10 % edge progression represents the Lag time. **(D)** Mean Lag time between the leading and lagging edge for indicated conditions. For all, error bars are SEM, P values were calculated using two-tailed Student's *t* test or Mann–Whitney U test and are n.s. P > 0.05, *P < 0.05, **P < 0.01 and ***P < 0.001.

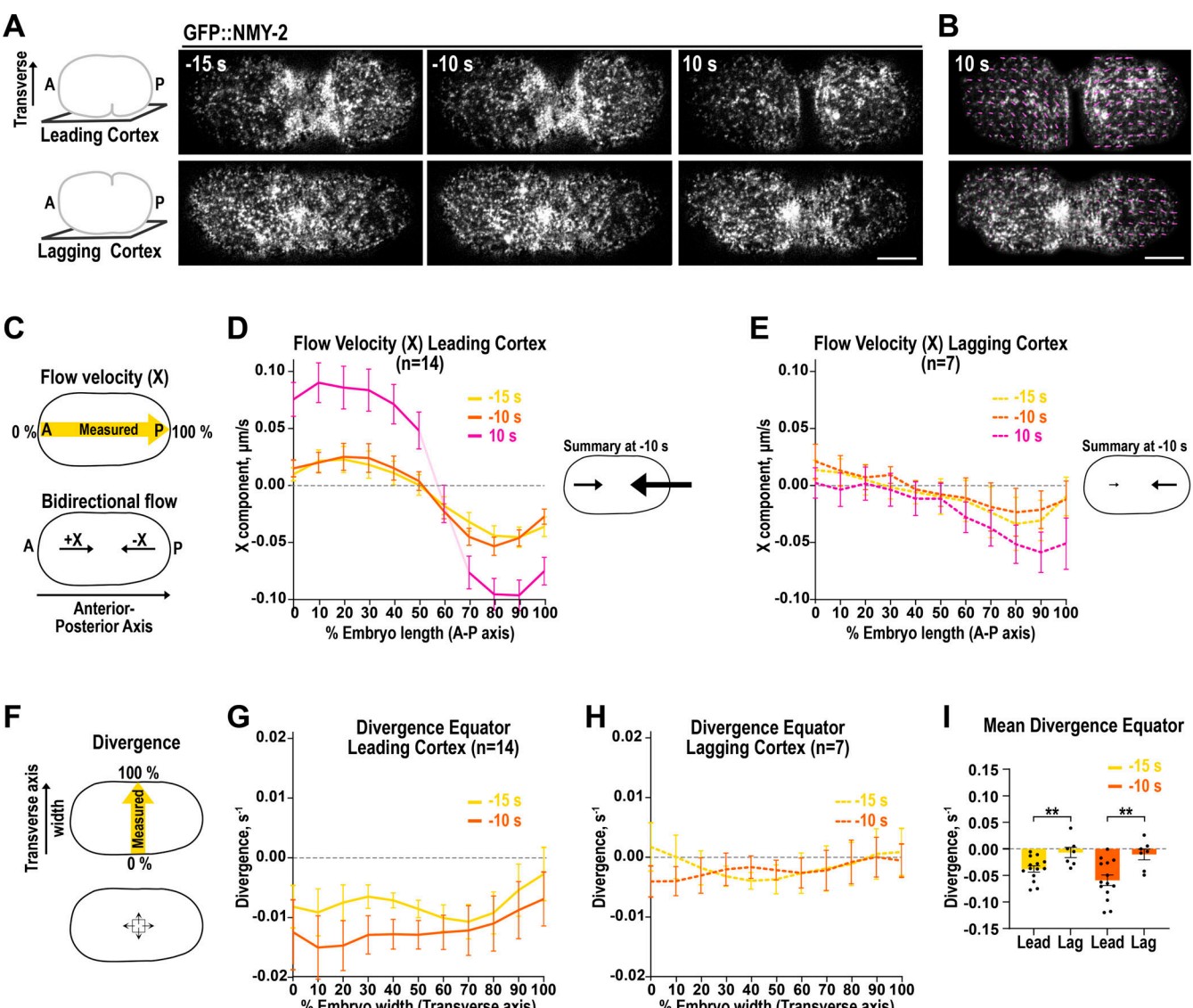

**Figure 9. Furrow directed cortical flows and compression are higher at the cortex of the leading than the lagging edge. (A)** Leading (*top*) and lagging (*bottom*) cortex of unconfined GFP::NMY-2 expressing embryos (z-projection of four planes) for indicated time points. Time zero represents the disappearance of the leading cortex from the imaging field after the leading edge ingression. The lagging cortex starts to bend inward and ingress around 20 s later. The anterior (A) pole is oriented to the left and the posterior (P) pole toward the right. The orientation of the transverse embryo axis is also shown. **(B)** Leading (*top*) and lagging (*bottom*) cortex of GFP::NMY-2 expressing embryos with measured vector field (magenta) at 10 s after the leading edge ingression. **(C)** GFP::NMY-2 flow velocities were measured from the anterior to the posterior pole of the embryo. A bidirectional A–P flow has positive X-values in the anterior region and negative X-values in the posterior region. **(D and E)** The flow velocities along the A–P axis were measured and plotted for indicated time points for the leading (D) and lagging (E) cortex. At 10 s, the leading cortex bent inwards and therefore flow velocities could not be calculated at the furrow region. For this reason, the line of the graph at the furrow region is depicted in a lighter color. A schematic summary of the measured flow velocities and their directions at −10 s are shown (*right*). The arrows reflect the flow velocities at 20% (anterior arrow) and 80% (posterior arrow) A–P embryo length and are to scale to each other. **(F–H)** The divergence of flow vectors was measured along the width of the embryo (transverse axis) at the cell equator (F) and is plotted for indicated time points for the leading (G) and lagging (H) cortex. **(I)** Mean divergence at the cell equator for the leading (Lead) and lagging (Lag) cortex at −15 s and −10 s before inward bending of the leading cortex. P values were calculated using two-tailed Student's *t* test and are **P < 0.01. Please note that the measurements at the cell equator were performed before the furrow moved inward and away from the coverslip to avoid measurement artifacts due to inward bending. For all: error bars are SEM and *n* = number of embryos analyzed, scale bars are 5 μm.

were higher and more consistently observed at the leading cortex we also expected a stronger compression at the leading cortex. To estimate the compression of the furrow region, we calculated the divergence of the flow vectors along the transverse axis across the embryo width at the cell equator (Fig. 9 F). Positive divergence values indicate that more material is flowing out than into an area, suggesting expansion of this area. Negative divergence values suggest that more material is flowing into than out of an area and thus reflect compression. At the cell equator, the divergence values were negative across the transverse axis of the embryo at the leading and lagging cortex, demonstrating that both cortices were compressed (Fig. 9, G and E).

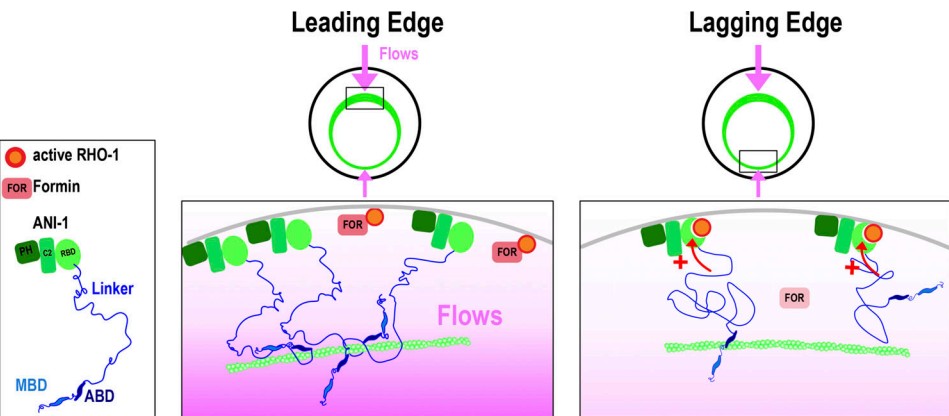

Figure 10.   **Working model.** At the lagging edge, the linker region of ANI-1 enhances the binding of the RBD to active RHO-1 (red arrows). This blocks the active side of RHO-1 for its effectors which dampens RHO-1 signaling and thereby slows down lagging edge ingression. At the leading edge, the linker region does not exert its positive effect on the RBD of ANI-1, and therefore the binding affinity of the RBD to RHO-1 is low and RHO-1 activates its effectors efficiently and drives the leading edge ingression. Since furrow-directed flows cause a compression of the actin cortex and enrichment of ANI-1 at the leading edge, we propose that the linker function is dampened at the leading edge in a flow-dependent manner.

Calculating the mean divergence at the cell equator just prior to the ingression of the leading cortex at –15 and –10 s revealed that the divergence was lower at the leading cortex than the lagging cortex, suggesting a stronger compression of the leading cortex (Fig. 9 I). We conclude that bidirectional furrow-directed cortical flows are stronger at the cortex of the leading edge than at the lagging edge. This causes a stronger compression of the actin-cortex at the leading cortex than at the lagging cortex.

## Discussion

Together, we propose the following model: an initial cue breaks the symmetry of the ring and initiates furrow-directed cortical flows which enrich ANI-1 at the leading edge (Khaliullin et al., 2018; Maddox et al., 2007; Reymann et al., 2016). The C-terminus targets ANI-1 to the ring, and its binding to active RhoA impedes the activation of RhoA effectors. The linker region of ANI-1 enhances the interaction between the C-terminus and active RhoA, particularly at the lagging edge, and thereby causes an occupation of the effector binding site and blocks the access for RhoA effectors. This limits RhoA signaling at the lagging edge, slows down its ingression, and consequently furrow-directed cortical flows (Fig. 10). At the leading edge, the function of the linker is hindered and therefore the RBD has only a low affinity for active RhoA. Consequently, the effector binding site of RhoA is not occupied and is free to bind and activate RhoA effectors. Cortical flows and compression are higher at the leading cortex than the lagging cortex, and therefore we propose that they influence the function of the linker. We speculate that the weaker the cortical flows, the more the ANI-1 linker promotes binding of the RBD to active RhoA. This reduces RhoA-dependent activation of formin and ROK, ring constriction, and furrow-induced flows at the lagging edge.

We find that ANI-1 inhibits the accumulation of multiple RhoA effectors during ring assembly and constriction. Since ANI-1 is enriched at the leading edge, previous work suggested that ANI-1 could sequester NMY-2 at the leading edge of the ring

(Maddox et al., 2007). If that is the case, we would expect after ANI-1 depletion a redistribution of NMY-2 from the leading to the lagging edge without increasing the total levels of NMY-2. However, this is not what we observed, and we found that the total levels of NMY-2 increase in the ring and that they are elevated at the leading and lagging edges. This suggests that ANI-1 inhibits NMY-2 accumulation. Similarly, anillin depletion causes an increase in myosin II levels during cytokinesis in *Drosophila* cells (Kechad et al., 2012) and elevated levels of the RhoA activity sensor during tight junction remodeling in *Xenopus* embryos (Craig et al., 2025). Interestingly, anillin depletion simultaneously abolishes myosin II recruitment to tight junctions (Craig et al., 2025). This block in myosin II accumulation could reflect an additional function of the anillin N-terminus in binding and stabilizing myosin II directly. Similarly, also in *C. elegans* one-cell embryos, although NMY-2 levels are higher in ANI-1 depleted embryos, NMY-2 no longer concentrates in cortical patches (Maddox et al., 2005) which could represent an independent function of the ANI-1 N-terminus.

During constriction the ring disassembles (Carvalho et al., 2009; Khaliullin et al., 2018), and therefore an increase in RhoA effector levels after ANI-1 depletion could also be caused by a reduction in the ring disassembly rate. After ANI-1 depletion, the constriction rate of the contractile ring is not reduced (Descovich et al., 2018), arguing against the role of ANI-1 in ring disassembly. Anillin is well-known to bind F-actin and myosin II via the N-terminus and stabilizes and crosslinks the actin networks (Dorn et al., 2016; Field and Alberts, 1995; Jananji et al., 2017; Straight et al., 2005; Tian et al., 2015; Watanabe et al., 2010). However, a function of anillin in actin–myosin network disassembly has, to our knowledge, not been reported. Further, our data show that the C-terminus of ANI-1, which lacks the ABD and MBD, is sufficient to restore total NMY-2 levels in the ring after *ani-1(RNAi)*. Together with the fact that the RBD binds active RhoA, we find a direct inhibition of RhoA activity by the RBD a more plausible explanation.

Although we find that cortical RHO-1 effector levels were increased, the levels and dynamics of IT-RHO-1 were not affected by ANI-1 depletion. We observed a rapid recovery of IT-RHO-1 after photobleaching, which aligns well with previous work on RhoA or other Rho GTPases in different organisms (Bendezú et al., 2015; Budnar et al., 2019; Wedlich-Söldner et al., 2004). However, we did not observe a change in IT-RHO-1 recovery rate or its immobile fraction after ANI-1 depletion, which contrasts with previous findings reporting a decrease in the immobile fraction and higher dynamics of RhoA in human cells depleted of anillin (Budnar et al., 2019). Consistent with the model that anillin stabilizes active RhoA multiple reports documented that anillin promotes the accumulation of RhoA effectors at cell–cell junctions and the cleavage furrow in human cells (Hickson and O'Farrell, 2008; Piekny and Glotzer, 2008; Budnar et al., 2019). How could the highly conserved RBD of anillin have RhoA stabilizing and inhibiting functions at the same time? We speculated that this switch from a RhoA inhibitor to a RhoA stabilizer could be regulated by the linker region and C2 domain, respectively. It was shown that the anillin C2 domain clusters PIP2 to which the RBD targets active RhoA. In the PIP2 cluster, the membrane residence time of active RhoA is increased and therefore RhoA signaling is enhanced (Budnar et al., 2019). This mechanism depends on a lower binding affinity of the RBD to active RhoA than other RhoA effectors (Blumenstein and Ahmadian, 2004; Budnar et al., 2019; Sun et al., 2015). Interestingly, we observed that the linker region enhances the binding of the RBD to active RhoA *in vitro* and is required for the asymmetric distribution of NMY-2 in the ring and unilateral furrowing. Thus, if the binding affinity of the RBD is increased by the linker, it could turn anillin into a negative regulator of RhoA activity. Therefore, we speculate that the RBD could inhibit or promote RhoA signaling depending on the functional state of the linker region and the C2 domain, respectively. Linker function could be controlled via phosphorylation (Kim et al., 2017) or by mechanical stimuli such as actin flows or phase separation (Chatterjee and Pollard, 2019). Interestingly, after ANI-1 depletion, cortical NMY-2 levels increased more during ring constriction (~2-fold increase, Fig. 5 B) than during ring assembly (~1.5-fold increase, Fig. 4 D), which supports our idea that the positive effect of the linker on the RBD could be modulated during different stages of cytokinesis. Similarly, the PIP2 binding affinity of the C2 domain could also be modulated by posttranslational modifications such as phosphorylation (Kim et al., 2017) and thereby alter the stabilizing effect of the RBD domain on active RhoA.

The direct binding of ANI-1 to RHO-1 via the ANI-1 RBD is, to our knowledge, the first reported RhoA inhibition mechanism that directly impacts RhoA itself, not operating via GEFs or GAPs (Bement et al., 2024). The C-terminus of human anillin also binds the GEF Ect2 (Frenette et al., 2012) and the p190RhoGAP-A (Manukyan et al., 2015), and therefore we cannot entirely exclude the possibility that *C. elegans* ANI-1 also modulates RHO-1 activity via its GEF or GAP. However, we consider this possibility unlikely since ECT-2 or RhoA GAPs have not been identified as ANI-1 binding partners (Rehain-Bell et al., 2017), and the two point mutations that prevent ANI-1-RhoA-binding

completely abolished membrane targeting of the ANI-1 C-terminus.

After ANI-1 depletion, NMY-2 levels increased on the leading and lagging edge of the ring although the fold increase was more pronounced at the lagging edge, suggesting that the inhibition of RHO-1 activity by ANI-1 is stronger at the lagging edge. Together with the observations that ANI-1 protein levels are lower at the lagging than the leading edge in control rings, these indicate that ANI-1 binds active RHO-1 more tightly at the lagging than the leading edge. Additionally, the C-terminus is sufficient to reduce NMY-2 levels at the leading edge in *ani-1(RNAi)* embryos, but at the lagging edge, the C-terminus alone was not sufficient, and for full RHO-1 inhibition, the C-terminus and the linker region had to be present (Fig. 6 D). Our data show that the linker enhances the binding of the RBD to active RhoA, and we propose that the function of the linker is differentially regulated at the leading and lagging edges of the ring. The linker is 320 amino acids long and highly disordered (Lebedev et al., 2023). Disordered protein regions have been proposed to be ideal sensors of cellular states due to their highly flexible nature and their easy transition between different conformational ensembles (Moses et al., 2023). Our measurements of cortical flows confirm that bidirectional flows are higher at the leading than the lagging cortex (Khaliullin et al., 2018), and consistent with that, the compression of the actin cortex is higher at the leading than the lagging furrow region. Based on our results, we consider the impact of the recently described circumferential flows from the leading cortex toward the lagging cortex (Hsu et al., 2023) to the compression of the actin cortex at the lagging cortex rather marginal. Since cortical flows have different strengths at the leading and lagging cortex, we speculate that they could modulate linker function. At the leading edge, cortical flows enrich ANI-1 which could, when reaching a certain concentration threshold, induce self-interaction of the linker and hinder its positive effect on the RBD-RHO-1 binding. Alternatively, since the linker region is long and flexible, its interaction with the flowing actin cortex could extend and displace it away from the membrane-localized RBD domain and thereby prevent its positive influence on the RBD. To distinguish between those possibilities, future efforts must investigate whether the linker self-interacts or binds the actin cortex. At the lagging edge, cortical flows and ANI-1 levels are low, possibly allowing the flexible linker to enhance the binding activity of the RBD to RHO-1 and strongly block RHO-1 signaling.

The ANI-1 C-terminus also harbors a PH domain, which binds and recruits septins after active RhoA binding of the RBD (Carim and Hickson, 2023). This suggests that RhoA-bound anillin blocks actin polymerization and myosin II activation but at the same time recruits septins. Anillin binding to septin contributes to anillin targeting the membrane and promotes the formation of membrane-localized circular structures of the ANI-1[C-term] in Formin[CYK-1]-depleted embryos (Lebedev et al., 2023; Piekny and Glotzer, 2008; Sun et al., 2015). Furthermore, septins are required for an asymmetric distribution of ANI-1 in the ring and unilateral furrowing (Maddox et al., 2007). Since septins typically form oligomers on the membrane (Woods and Gladfelter, 2021), they could promote the clustering of ANI-1 at the leading

edge and also contribute to hindering linker function, by an as yet unknown mechanism.

In sum, we propose that anillin has RBD-dependent positive and negative regulatory roles on RhoA signaling that are modulated by the linker (this study) and by the C2 domain (Budnar et al., 2019). How anillin switches from a positive to a negative regulator of RhoA signaling and how this contributes to different processes such as cell–cell junction repair or cytokinesis in time and space will be important avenues for future research.

# Materials and methods

## *C. elegans* maintenance, strain construction, and RNAi-mediated protein depletions

*C. elegans* strains were grown at 20°C on NGM plates seeded with *Escherichia coli* (OP50) (Stiernagle, 2006), and the strains generated and used in this study are listed in Table S1.

For RNAi-mediated protein knock-down, dsRNA was introduced by feeding or injection. The details of the dsRNA concentrations and primers are listed in Table S2. To generate dsRNA, the target locus was amplified by PCR using complementary oligonucleotides carrying T7 overhangs. PCR products were used as templates for *in vitro* transcription (MEGAscript T7 kit, AM1334; Invitrogen, Thermo Fischer Scientific). Young adults were injected with dsRNA and maintained on seeded NGM plates at 20°C for 18–22 h (*rga-3/4*), 24 h (*ect-2*), 24–32 h (*rho-1*), or 39–49 h (*cyk-1, ani-1, cyk-1 ani-1*). Mild depletion conditions were chosen for RGA-3/4 knockdown to prevent cortical hypercontractility. ANI-1 and CYK-1 depletion efficiency was verified in a previous study by fluorescence intensity measurements of endogenously tagged proteins and immunoblot analysis (Lebedev et al., 2023).

For feeding, RNAi L4440 vectors with *ani-1* or *perm-1* from the Ahringer library (Source Bioscience) were transformed into HT115 (DE3) bacteria. After growing bacterial clones overnight, they were pelleted, resuspended in LB medium containing 12.5 μg/ml tetracycline, 50 μg/ml ampicillin, and 0.23 mg/ml IPTG (isopropyl β-D-1-thiogalactopyranoside), and added to plates pretreated with 100 μl of a 1:1:1 mix of 50 μg/ml ampicillin, 12.5 μg/ml tetracycline, and 0.1 g/ml IPTG. For permeabilized embryos in Fig. S3, B–E, 20 μl of bacteria culture producing *perm-1* dsRNA was mixed either with 80 μl of LB or 80 μl of bacteria expressing *ani-1*. Plates were incubated for 5 h at 37°C in the dark. Subsequently, L4 hermaphrodites were added and incubated for 45–48 h at 20°C before imaging.

The internally tagged RHO-1 strain was generated by Suny-Biotech (https://www.sunybiotech.com). *C. elegans*-optimized superfolder GFP (Wang et al., 2017) was inserted into an external exposed loop of RHO-1 between Q136 and E137 ($Q_{136}$-SGGS-sGFP-GAPG-$E_{137}$, linker sequence), similar to Bendezú et al. (2015). Correct integration of the sfGFP was verified by sequencing, and the strain was balanced with the nT1 (IV/V) balancer.

## Latrunculin A treatment of permeabilized embryos

*perm-1*(RNAi) and *perm-1*(RNAi);*ani-1*(RNAi) embryos were imaged in meiosis medium (25 mM HEPES, pH 7.4, 0.5 mg/ml inulin, 20% heat-inactivated fetal bovine serum, and 60%

Leibowitz-15 medium) without compression in a custom-build microdevice (Carvalho et al., 2011). Latrunculin A (L5163; Sigma-Aldrich, BML-T119; Enzo Life Sciences) was added just before the onset of anaphase to obtain a final concentration of 10 μM. To confirm embryo permeability, FM 4-64 (T13320; Invitrogen; Thermo Fisher Scientific) was added at the end of filming.

## Counting embryonic lethality and mean number of progeny

To test the ability of the transgenic ANI-1 constructs to rescue embryonic lethality caused by *ani-1*(RNAi), young adult hermaphrodites were injected with *ani-1* dsRNA and maintained at 20°C. After 48 h incubation injected animals were singled onto fresh plates for 24 h. After another 24 h incubation time injected animals were removed and the number of dead embryos and hatched larvae was counted 24 h later.

To determine number of progeny control (N2) and homozygote *it-rho-1* (pharynx GFP-negative animals) gravid hermaphrodites were singled on NGM plates and incubated at 20°C for 24 h. Hermaphrodites were removed from the plates and after 24 h the total number of progeny was counted.

## Immunoblotting and RHO-1 antibody production

For immunoblot analysis, worms were collected and washed three times in MPEG (M9, 0.05% polyethylene glycol 8000 [PEG]) buffer. Excess MPEG buffer was removed, and the sample buffer was added to reach an approximate concentration of 3 worm/μl. Samples were boiled at 95°C for 5 min, centrifuged to spin down debris, and finally 20 μl supernatant was loaded on the SDS-PAGE (Zanin et al., 2011). PVDF membranes (Immobilon-P IPVH00010; Merck) were incubated with anti-Actin (1:6,000, A1978; Sigma-Aldrich), anti-RHO-1 (self-produced), anti-FLAG (1:2,500; F3165; Sigma-Aldrich) or anti-GFP (1:600) primary antibodies. HRP-conjugated anti-mouse (1706516, 1: 7,500; Bio-Rad) or HRP-conjugated anti-rabbit (170-6515; 1: 15,000; Bio-Rad) antibodies were used as secondary antibodies. Membranes were imaged on the BIO-RAD ChemiDoc XRS+ with the Image Lab software (v.5.2.1. build 11, Bio-Rad).

For RHO-1 antibody generation, the *C. elegans rho-1* cDNA sequence (AA 1-188) without the CAAX-motif was cloned into pGEX-4T-1 vector with an N-terminal glutathione-S-transferase-tag (GST-tag). GST-RHO-1 expression was induced in *E. coli* BL21 (DE3) for 18 h at 25°C with IPTG. After sonication, GST-tagged RHO-1 was purified using Glutathione Sepharose 4B protein purification resin (17075601; GE Healthcare). The purified antigen was used for immunization of rabbits to generate a polyclonal antibody (Davids Biotechnology GmbH). Affinity purification of the RHO-1 antibody was performed using the antigen coupled to SulfoLink Coupling Resin (20401; Thermo Fisher Scientific) after GST antibodies were removed.

## Cloning of GFP::ANI-1$^{C-term-RBM}$ and FLAG-tagged ANI-1 variants and their integration by Mos1-mediated single copy insertion (MosSCI)

To generate the RHO-1 binding mutation of ANI-1, two point mutations (A789D and E807K) were introduced into GFP::ANI-1$^{C-term}$ (pEZ391, [Lebedev et al., 2023]) construct by site-directed mutagenesis (Table S4). For the generation of the FLAG-tagged

ANI-1 variants, a 3xFLAG-tag was inserted at the ANI-1 N-terminus using Gibson assembly.

The GFP::ANI-1$^{C-term-RBM}$ (pEZ443), FLAG::ANI-1$^{WT}$ (pEZ450), FLAG::ANI-1$^{N-term+Link}$ (pEZ455) were integrated into the EG8079 strain on chromosome II; FLAG::ANI-1$^{C-term}$ (pEZ448) was integrated into the EG8080 strain on chromosome III; FLAG::ANI-1$^{Link+C-term}$ (pEZ456) was integrated into the EG8081 strain on chromosome IV using the MosSCI procedure (Frøkjær-Jensen et al., 2008). Young adult worms were injected with a mixture containing the vector carrying the ANI-1 transgene, the transposase (pCFJ601), and coinjection markers (pCFJ90 and pCFJ104). Animals were rescued on fresh NGM plates and incubated at 25°C for 7–10 days until starvation. Correct integration was verified by wild type moving, absence of mCherry-tagged array markers, homozygous GFP expression, and PCR genotyping.

### Cloning and protein expression of GST-ANI-1 and 10xHis-RHO-1 in *E. coli* bacteria

To clone GST-tagged ANI-1$^{Linker}$ (460-775 AA), ANI-1$^{C-term}$ (681-1159 AA), and ANI-1$^{Link+C-term}$ (460-1159 AA) for expression in *E. coli*, the corresponding regions were amplified from *C. elegans* cDNA and introduced into pGEX-4T using Gibson assembly (Table S4). The corresponding RHO-1 defective binding mutant (ANI-1$^{C-term-RBM}$, 681-1159 AA, A789D, E807K) was generated by side-directed-mutagenesis. Wild type RHO-1 (1-188 AA) was amplified from cDNA and cloned into pET-19b expression vector containing a 10xHis-tag. Constitutively active Q63L RHO-1 was generated by site-directed mutagenesis. All GST-tagged and 10xHis-tagged proteins were expressed in 1 L BL21 (DE3) *E. coli* bacterial strain. When the bacterial culture reached an OD 600 of 0.5–0.6, the bacteria were induced with 0.5 mM IPTG for 2 h at 37°C (GST only) or 16 h at 18°C (GST-ANI-1 and 10xHis-RHO-1 constructs). Subsequently, bacterial cultures were collected and washed with ice-cold 1× PBS following freezing in liquid nitrogen.

### 10xHis-RHO-1 and GST-ANI-1 protein purification

10xHis-RHO-1 and GST-ANI-1 bacterial pellets were thawed in complete lysis buffer (20 mM Tris pH 7.4, 500 mM NaCl, 0.2% Triton X-100, cOmplete protease inhibitor cocktail [11836153001; Roche], 1 mM DTT [6908.3; Roth], 1 mM PMSF [p7626-1G; Sigma-Aldrich]) for 20 min and broken by sonication. The lysates were cleared by centrifugation at 21,000 *g* for 20 min. The 10xHis-RHO-1 supernatant was collected and incubated with packed Ni-NTA Agarose (30210; Qiagen) for 1 h at 4°C by rotation. The mixture was then transferred to a polypropylene column (29922; Thermo Fisher Scientific) and left to settle beads down for 15 min. The nickel resin was washed three times with washing buffer (50 mM Tris pH 7.4, 500 mM NaCl, 50 mM Imidazole, 10% Glycerol), and proteins were eluted with 5 ml elution buffer (50 mM Tris pH 7.4, 500 mM NaCl, 500 mM imidazole, 10% glycerol). The elution buffer was exchanged to storage buffer (25 mM Tris pH 7.4, 250 mM NaCl, 10% glycerol) using Slide-A-Lyzer 10k dialysis cassettes (66380; Thermo Fisher Scientific), and then the proteins were frozen in liquid $N_2$ and stored at –80°C.

GST-ANI-1 supernatant was incubated with Glutathione Sepharose 4B (17075601; Cytiva) rotating for 1 h at 4°C. The GST-enriched beads were washed three times with 15 ml lysis buffer followed by centrifugation at 500 *g* for 5 min and used directly in the pull-down assay. The approximate concentration of purified GST-ANI-1 and 10xHis-RHO-1 proteins was analyzed by SDS-PAGE using known BSA standards.

### GST-ANI-1 and 10xHis-RHO-1 pull-down assay

For the pull-down assay, ~4 μg of purified 10xHis-RHO-1$^{WT}$ or 10xHis-RHO-1$^{Q63L}$ were mixed with ~4 μg of freshly purified GST-ANI-1$^{C-term}$, GST-ANI-1$^{C-term-RBM}$, GST-ANI-1$^{Linker}$, or GST-ANI-1$^{Link+C-term}$ containing beads in a total volume of 300 μl binding buffer (20 mM Tris, pH 7.4, 150 mM NaCl, 5 mM MgCl$_2$, 0.1% Triton X-100, 1 mM DTT) supplemented with 1 mM GTP. Samples were rotationally incubated at 4°C for 1.5 h. Beads were washed three times with binding buffer for 5 min by rotation and centrifuged at 500 *g* for 5 min. Excess supernatant was removed from the washed beads following the addition of an appropriate volume of 2x sample buffer. Pull-down samples were loaded on SDS-PAGE, and either Coomassie stained or detected by immunoblotting. PVDF membranes (Immobilon-P IPVH00010; Merck) were incubated with anti-His (1:1,000; MA1-135; Invitrogen) primary antibody, and HRP-conjugated anti-mouse (1:7,500; 1706516; Bio-Rad) was used as the secondary antibody. Membranes were imaged on the Fusion SL VILBER LOURMAT (peqlab) with the FusionCapt Advance software. The immunoblot blot (10xHis-RHO-1$^{WT}$ or 10xHis-RHO-1$^{Q63L}$) and Coomassie stained (GST fragments) band intensities were quantified using the ImageJ plugin "Gels". The intensity of the immunoblot blot bands was normalized to the corresponding bands of GST fragments in the Coomassie gel staining. The GST-ANI-1$^{C-term}$ + 10xHis-RHO-1$^{Q63L}$ pull-down was set to 100% for relative comparison.

### Image acquisition by fluorescence microscopy

For confocal microscopy, embryos were mounted in M9 buffer on an 2% agar pad (Figs. 1, 3, 4, S1, S2, and S4) or imaged without compression in open self-made imaging chambers (Figs. 5, 6, 7, 8, 9, S3 B, and S5) at 20°C. Confocal microscopy images of Fig. 1, Fig. 3 A, and Fig. S2 G were acquired using a Nikon inverted microscope (Eclipse Ti) equipped with a confocal spinning disk unit, a 100× 1.45-NA Plan-Apochromat oil immersion objective, and an Andor DU-888 X-11056 camera (1024 × 1024 pixels) controlled by NIS Elements software. GFP and red-fluorescent probes were imaged using 488 and 561 nm lasers, respectively. Confocal images of Figs. 3 E, 4 B, 5, 6, 7, 8, 9, S1 D, S4 A, and S5 were acquired on a Leica TCS SP8 DIVE-FALCON equipped with 488 nm argon and 561 nm lasers and APO CS2 63×/1.4 oil objective. Cortical images in Fig. S3 B were acquired on a spinning disk confocal system (Andor Revolution XD Confocal System; Andor Technology) with a confocal scanner unit (CSU-X1; Yokogawa Electric Corporation) mounted on an inverted microscope (Ti-E, Nikon) equipped with a 60× 1.4 NA Plan-Apochromat oil objective and solid-state lasers of 488 nm (50 mW) and 561 nm (50 mW). Z-stacks (12 × 0.5 μm) were collected every 5 s by using an electron multiplication back-thinned charge-coupled device

camera (iXon Ultra 897; Andor Technology). Acquisition parameters, shutters, and focus were controlled by Andor iQ3 software.

Imaging was started prior NEBD at the central plane with a 16-s time interval, stopped at NEBD, and continued at the cell cortex before anaphase onset with 2.5 s (Fig. 1, Fig. 3 A, and Fig. S2 G) or 5 s (Fig. 4 B, Fig. S1 D, and Fig. S4 A) time interval. Cytokinetic ring closure was imaged in uncompressed one-cell embryos using self-made chambers, which were built from double-sided tape with a hole in the center as previously described (Lebedev et al., 2023). Imaging was performed from NEBD until completion of cytokinesis with a frame rate of 15 s and 21 z-planes with 1.5-μm step size. The FRAP of IT-RHO-1 (Fig. 3 E) was conducted in a 5 × 1 μm rectangular region which was placed on the plasma membrane of the two newly formed daughter cells at about 1 min after cytokinetic ring closure. Images were acquired in a single z-plane every 0.27 s. The FRAP time series included two prebleach images with 4% of the 488 nm laser, photobleaching with 100% of the 405, 458, 476, 488, 496, and 514 nm lasers (10 frames, ~2.4 s), and recovery imaging with 4% of the 488 nm laser for 200 time points.

Imaging for PIV analysis (Fig. 9) was performed in open self-made chambers of uncompressed one-cell embryos. Imaging was started at the central plane and continued until anaphase onset was observed with the histone marker (mCherry::H2B). Imaging of GFP::NMY-2 was continued at the cell cortex with four z-planes and a step size of 0.75 μm every 5 s. After the ingression of the cleavage furrow, the time series was stopped and the entire embryo was imaged with nine z-planes and a step size of 4 μm to determine whether the imaged cortex was the leading cortex or the lagging cortex.

The noise of the cortical plane Leica SP8 images was reduced by a Gaussian Blur filter with a radius of 0.8. For images generated on the Nikon microscope, the mean fluorescence intensity was measured over time in the posterior region of the cortex for each embryo and multiplied to specific timepoint intensity values of Nikon images to correct bleaching.

Image analysis and quantification were performed in Fiji, data analysis was performed in Excel, Prism (GraphPad), or KNIME analytics (https://www.knime.org), and figures were assembled with the Affinity Designer software.

### Equatorial fluorescence intensity measurements and IT-RHO-1 structure analysis

For all experiments, NEBD was defined as the time point when the border of the nucleus was no longer visible in the transmission channel or cytoplasmic fluorescent NMY-2 entered the nucleus. Full ring constriction was reached when the furrow was completely closed. Line scan analysis of cortical plane images (Figs. 1, D and G; 3 B, 4 C, S1 E, S2 H, S3 D, and S4 C) was performed by drawing a 4.8-μm-wide line on the cortex along the A-P axis of the embryo and normalized to the mean cytoplasmic intensity measured in a box at NEBD. The equatorial peak intensity (Figs. 3 C, 4 D, S1 B and F, S2 I, S3 E, and S4 D) was determined within 30% to 70% embryonic length and the mean equatorial intensity around the peak (±5% length range) was calculated (Fig. S1 A).

To quantify the number of linear and nonlinear cortical structures (Fig. 3 D and Fig. S3 A) a Gaussian Blur filter (radius 0.5) followed by an Unsharp Mask filter (radius 2, Mask Weight 0.6) was applied to the confocal images and subsequently the mean fluorescence intensity, measured at the anterior cortex, was subtracted. Dimensions of cortical structures were counted manually in a rectangular box at the cell equator. Structures with the length/width ratio ≥4 were defined as linear and with a length/width ratio of <4 as nonlinear (Lebedev et al., 2023).

### FRAP analysis of IT-RHO-1

The mean fluorescence intensity of IT-RHO-1 in the bleached region was measured over time and the fluorescence intensity outside the embryo was subtracted as background. The fluorescence intensity of the bleached region was normalized to the intensity of an unbleached plasma membrane region of the same size throughout the entire image series. The IT-RHO-1 recovery dynamics was analyzed by fitting the fluorescence intensity measurements to a one-phase association equation in GraphPad Prism 9 and calculating the recovery half-time and percentage of recovery.

### Analysis of cytokinetic ring closure and fluorescence intensities in the ring

Cytokinetic ring closure rates and fluorescence intensities in the ring were measured in uncompressed one-cell embryos. To construct an end-on view of the ring, a 5-μm wide equatorial ingression zone was cut out using Fiji's "Reslice" function, followed by z-projection of the maximum intensity from all obtained slices (Fig. S5 A). To measure ring eccentricity and edge progression, the ring constriction rate was measured by manually placing a circular shape on the cytokinetic ring at each time point. Eccentricity values ($Q_t$) were calculated as the ratio of the distance between the initial ($C_0$) and current cytokinetic centers ($C_t$) to the initial diameter of the ring ($R_0$) before the onset of constriction (Fig. 8 A) and either plotted as peak eccentricity throughout cytokinesis (Fig. 8 B) or relative to the % of ring constriction (Fig. 8 A), similar to Hsu et al. (2023). Edge progression (Fig. 8 C) of leading and lagging edges at a particular time point were calculated according to following equations from Hsu et al. (2023):

$$\frac{R_0 + Q_t - R_t}{2R_0} * 100 = Leading\ edge, \%$$
$$\frac{R_0 - Q_t - R_t}{2R_0} * 100 = Lagging\ edge, \%$$

The lag time between the leading and lagging edge (Fig. 8 C) was calculated for each embryo as a time difference when leading and lagging edges reached 10% of the initial ring diameter (Fig. 8 C).

The total NMY-2::mKate, Formin[CYK-1]::GFP, IT-RHO-1, and GFP::ANI-1 ring fluorescence intensity (Figs. 5 B, 6 A, 7 B, and S5 E) was measured at 50% ring constriction. First, a circular shape was placed on the inner edge of the ring and the mean cytoplasmic intensity was calculated and subtracted as background from each image. Subsequently, a circular shape was drawn to encompass the outer edge of the ring (Fig. 5 B). To calculate the

total fluorescence intensity in the ring the Fiji's "Oval Profile" plugin with the "Radial Sums" mode was applied and the fluorescence intensity in the 360 radii was summed up.

To determine the NMY-2::mKate and GFP::ANI-1 fluorescence intensities at the leading or lagging edge, first, the leading and lagging edge were assigned to the left and right half of the ring. For this, the position of the cytokinetic center ($C_t$) at 50% constriction was assigned to the left and right half of the division plane and the lagging edge was defined as the site where the $C_t$ was localized (Fig. 6 C). One embryo whose leading edge moved strictly upward away from the coverslip was not included in the analysis due to the fluorescence intensity attenuation in z-planes further away from the objective. For the same reason, the fluorescence intensities at the top and bottom 40° of the ring circle were not included in the analysis (Fig. 6 C). Finally, the fluorescence intensities at the leading or lagging edge were summed up to obtain the total leading and lagging edge fluorescence intensity (Fig. 6 D) or the ratios between the total leading and lagging edge intensities were calculated (Fig. 6 E, Fig. 7 C, and Fig. S5 F).

### Particle image velocimetry (PIV) analysis

Images were analyzed with MATLAB R2024b using PIVlab 3.05 software (https://PIVlab.de) (Stamhuis and Thielicke, 2014) after the background intensity outside the embryo was subtracted from the images. In the PIV analysis time point zero is defined as the time point when the leading cortex bends inward and a "black" line appears at the cell equator. Time comparison revealed that the lagging cortex bends inward ~20 s after the leading cortex and therefore the timings of the leading edge were corrected by this value. PIV analysis employed the Multipass FFT window deformation algorithm with an interrogation area of 32 px (2.9 µm), a step size of 16 px (1.45 µm), and a Gauss 2 × 3 point sub-pixel estimator. The mean flow velocities along A–P-axis of the embryo (X component) were extracted from a four-row vector matrix with a total width of 24 px (2.2 µm) (Fig. 9, B–E). The mean divergence was extracted from a four-row vector matrix with a total width of 24 px (2.2 µm) across the embryo width along the transverse axis of the embryo (Fig. 9, F–I). Mean divergence was plotted either across the embryo width (Fig. 9, G and H) or calculated across the entire furrow region for the leading and lagging cortex (Fig. 9 I).

### Statistical analysis

Statistical analysis was performed in Prism (GraphPad). The normality of the data was tested with Shapiro–Wilk test. For parametric data, the two-tailed Student's $t$ test and for nonparametric data the Mann–Whitney U test were performed, as indicated in the figure legends.

### Online supplemental material

Fig. S1 shows that ANI-1 limits Formin[CYK-1]::GFP accumulation to the cell cortex during contractile ring assembly. Fig. S2 shows that the levels of IT-RHO-1 on the plasma membrane change after ECT-2 GEF and RGA-3/4 GAP depletion during cytokinesis. Fig. S3 shows that Latrunculin A–induced IT-RHO-1 structures

depend on ANI-1. Fig. S4 shows that abolishing RHO-1-binding of ANI-1 prevents cortical enrichment of the ANI-1 C-terminus in the one-cell *C. elegans* zygote. Fig. S5 shows NMY-2::mKate localization in the constricting ring in *C. elegans* embryos expressing different GFP::ANI-1 variants. Video 1 shows that ANI-1 limits the accumulation of NMY-2::mRFP in the one-cell *C. elegans* zygote during contractile ring assembly. Video 2 shows that ANI-1 limits the accumulation of GFP::ROK[LET-502] in the one-cell *C. elegans* zygote during contractile ring assembly. Video 3 shows that ANI-1 tethers IT-RHO-1 to linear structures after Formin[CYK-1] depletion but does not affect IT-RHO-1 levels. Video 4 shows that ANI-1 binding to RHO-1 via the RBD limits cortical NMY-2::mKate accumulation during contractile ring assembly. Video 5 shows that ANI-1 limits the accumulation of NMY-2::mKate in the ring during constriction. Video 6 shows that ANI-1 limits the accumulation of Formin[CYK-1]::GFP in the ring during constriction. Video 7 shows that ANI-1 depletion does not affect the accumulation of IT-RHO-1 in the constricting ring. Video 8 shows that the linker region and the C-terminus of ANI-1 are both required to restore NMY-2 asymmetry in the ring. Table S1 shows *C. elegans* strains used in the study. Table S2 shows information about the dsRNA used for RNAi in the study. The T7 sequence is underlined. Table S3 shows recombinant DNA used in this study. Table S4 shows DNA oligos used for cloning in this study.

### Data availability

All data supporting the findings of this study are available within the paper and its supplementary information. *C. elegans* strains and plasmids generated in this study are available upon request from the corresponding author.

## Acknowledgments

We are grateful to M. Roth and S. Lakshman for technical assistance and to A. Lochner and S. Hornych for crossing strains. Microscopy was performed at the Center for Advanced Light Microscopy (CALM) at the LMU and the optical imaging center Erlangen (OICE) at the FAU. We thank S. Müller for help with the IT-RHO-1 FRAP experiments.

The Deutsche Forschungsgemeinschaft supported E. Zanin (ZA619/3) and T. Mikeladze-Dvail (MI1867/1-1). The Leica SP8 DIVE-FALCON was supported by the DFG (INST 90/1074-1 FUGG for Benedikt Kost). Research in A.X. Carvalho lab was funded by the European Research Council under the European Union's Horizon 2020 Research and Innovation Programme (grant agreement 640553 – ACTOMYO) and by National Funds through FCT—Fundação para a Ciência e a Tecnologia, I.P., under the project UIDB/04293/2020. A.X. Carvalho was supported by national funds through the FCT by a Principal Investigator position under the Scientific Employment Stimulus (CEECIND/01967/ 2017 and 2023.14140.TENURE.0139). F.-Y. Chan was supported by an FCT junior researcher position (DL 57/2016/CP1355/ CT0013). J. Bellessem and E. Rackles were members of the Life Science Munich graduate program. For critical comments on the manuscript, we thank A. Jangir. For sharing *C. elegans* strains, we thank A. Maddox and S. Grill. Some strains were provided by the

CGC, which is funded by the National Institutes of Health Office of Research Infrastructure Programs (P40 OD010440).

Author contributions: M. Lebedev: Formal analysis, Investigation, Methodology, Visualization, Writing - review & editing, F.-Y. Chan: Funding acquisition, Investigation, Methodology, Writing - review & editing, E. Rackles: Investigation, Writing - review & editing, J. Bellessem: Investigation, T. Mikeladze-Dvali: Conceptualization, Funding acquisition, Supervision, Writing - review & editing, A. Xavier Carvalho: Conceptualization, Funding acquisition, Project administration, Resources, Supervision, Writing - review & editing, E. Zanin: Conceptualization, Funding acquisition, Methodology, Project administration, Supervision, Visualization, Writing - original draft, Writing - review & editing.

Disclosures: The authors declare no competing interests exist.

Submitted: 31 May 2024

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

# Supplemental material

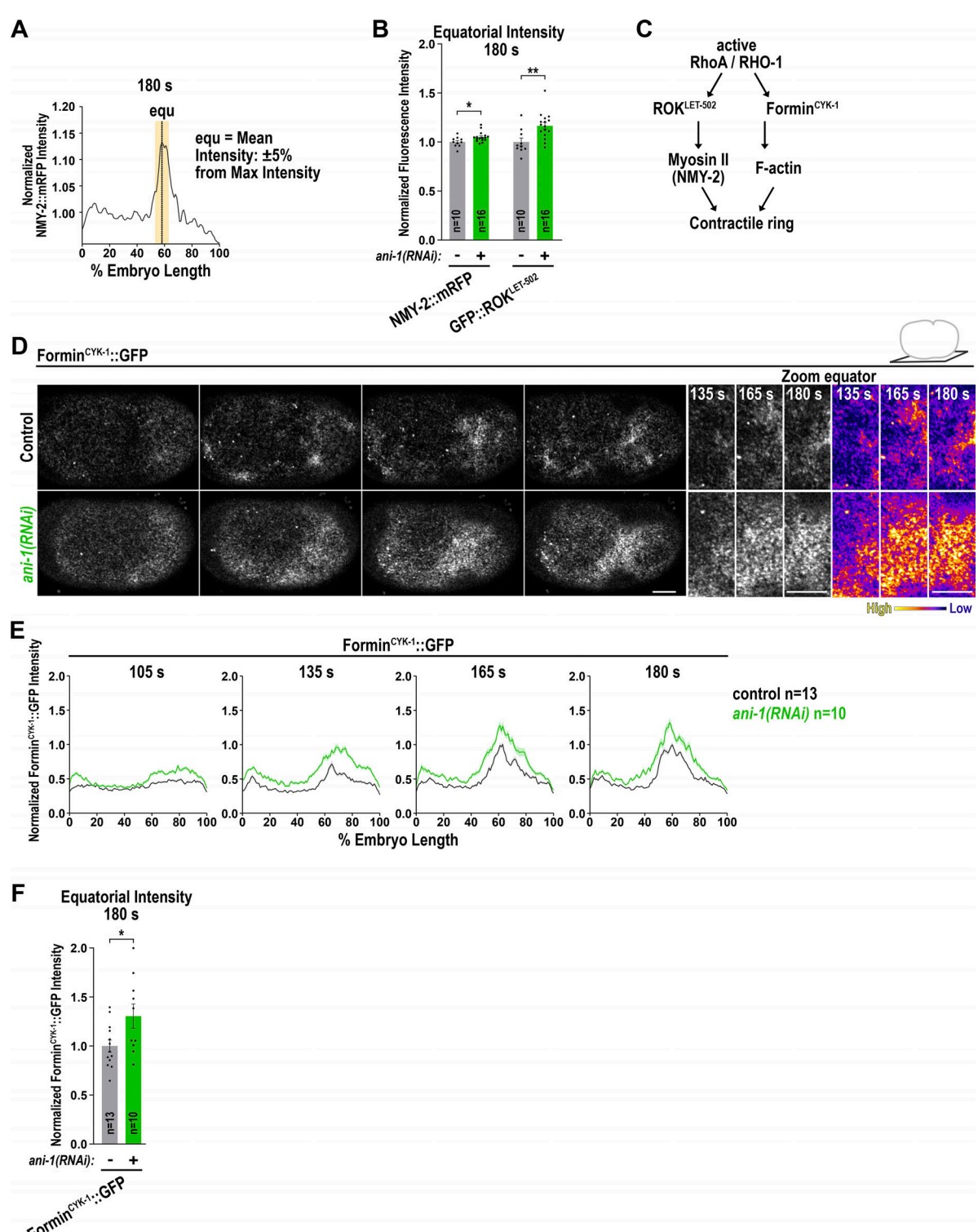

Figure S1.   **ANI-1 limits Formin^CYK-1^::GFP accumulation to the cell cortex during ring assembly. (A)** Representative intensity graph of NMY-2::mRFP line scan from anterior to the posterior pole of the embryo. For the equatorial intensity, the mean fluorescence intensity ±5% embryo length at the maximum intensity was calculated. **(B)** Mean normalized fluorescence intensity of NMY-2::mRFP and GFP::ROK^LET-502^ at the cell equator with and without *ani-1(RNAi)*. **(C)** During cytokinesis, active RhoA activates the effectors ROK^LET-502^ and Formin^CYK-1^. Activated ROK^LET-502^ activates myosin II and fomin^CYK-1^ polymerizes the long F-actin of the contractile ring. **(D)** Single z-plane images of the cell cortex of one-cell *C. elegans* embryos expressing Formin^CYK-1^::GFP treated with or without *ani-1(RNAi)*. A zoom-in of the equatorial region for the indicated time points is shown on the right in gray and fire scaling. **(E)** Normalized cortical fluorescence intensity of Formin^CYK-1^::GFP from the anterior (0%) to the posterior (100%) pole for control and *ani-1(RNAi)* embryos. **(F)** Mean normalized fluorescence intensity of Formin^CYK-1^::GFP at the cell equator with and without *ani-1(RNAi)* at 180 s after NEBD. For all, time in seconds (s) after NEBD is indicated, scale bars are 5 μm, error bars SEM, and *n* = number of embryos analyzed. P values were calculated using two-tailed Student's *t* test or Mann–Whitney U test and are *P < 0.05, **P < 0.01.

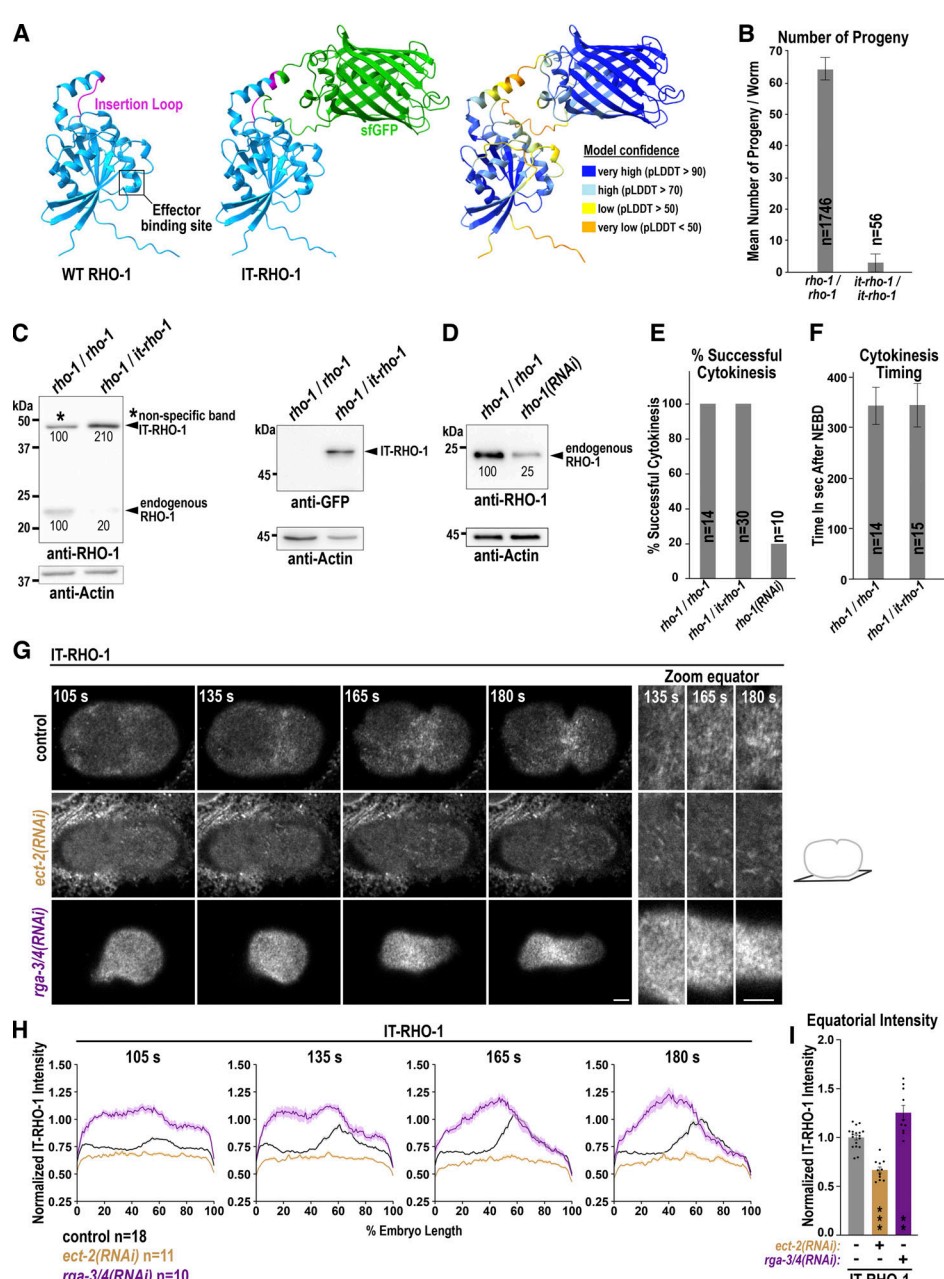

Figure S2. **IT-RHO-1 responds to ECT-2 GEF and RGA-3/4 GAP depletion. (A)** To generate IT-RHO-1 the superfolder GFP (sGFP) was inserted into a conserved external exposed loop of RHO-1. Three-dimensional structure of *C. elegans* RHO-1 (UniProt ID Q22038) and IT-RHO-1 with sfGFP (green) predicted by AlphaFold https://galaxyproject.org/citing-galaxy/ (Galaxy Community, 2022). The position of the effector binding site and the external loop are highlighted. The local confidence of the AlphaFold prediction is indicated by the scores of the predicted local distance difference test (pLDDT). **(B)** Mean number of progeny per worm of wild type (*rho-1/rho-1*) and homozygote (*it-rho-1/it-rho-1*) hermaphrodites. Error bars are standard deviation (SD) and *n* = number of progenies analyzed. **(C)** Immunoblot of wild-type (*rho-1/rho-1*) or heterozygote (*rho-1/it-rho1*) adult hermaphrodites treated with anti-RHO-1 (*left*) or anti-GFP antibodies (*right*). Please note that with the anti-RHO-1 antibody a non-specific band is present at the same height as the IT-RHO-1 (black star). After image acquisition, the membranes were washed with buffer and probed with anti-Actin antibodies to ensure similar loading. The intensity of the different bands is indicated in % (mean of three worm extracts). **(D)** Immunoblot of control and *rho-1(RNAi)* worms probed with anti-RHO-1 and anti-Actin antibodies. The intensity of the RHO-1 bands is indicated in % (mean of 3 worm extracts). The membrane was cut and the upper part was treated with the anti-RHO-1 and the lower part with the anti-Actin antibodies. **(E)** Cytokinesis success determined by live-cell imaging on central planes images for one-cell embryos derived from wild type (*rho-1/rho-1*), heterozygote (*rho-1/it-rho-1*) or *rho-1(RNAi)* treated hermaphrodite animals and *n* = number of embryos analyzed. **(F)** Time in seconds (s) from NEBD to the completion of cytokinesis in wild type (*rho-1/rho-1*) and IT-RHO-1 (*rho-1/it-rho-1*) expressing embryos quantified on central plane images from live-cell imaging. Error bars are SD and *n* = number of embryos. **(G)** Confocal single z-plane cortical images and magnifications of the equatorial region (*right*) of IT-RHO-1 expressing embryos for the indicated RNAi conditions and time points after NEBD during cytokinesis. Scale bars are 5 μm. **(H)** Normalized cortical fluorescence intensity of IT-RHO-1 from the anterior (0%) to the posterior (100%) pole for indicated RNAi conditions and *n* = number of embryos analyzed. **(I)** Mean normalized fluorescence intensity of IT-RHO-1 at the cell equator for indicated RNAi conditions at 180 s after NEBD. Error bars are SEM. P values in comparison to uninjected controls were calculated using a two-tailed Student's *t* test are **P < 0.01 and ***P < 0.001. For all, time in seconds (s) after NEBD is indicated. Source data are available for this figure: SourceData FS2.

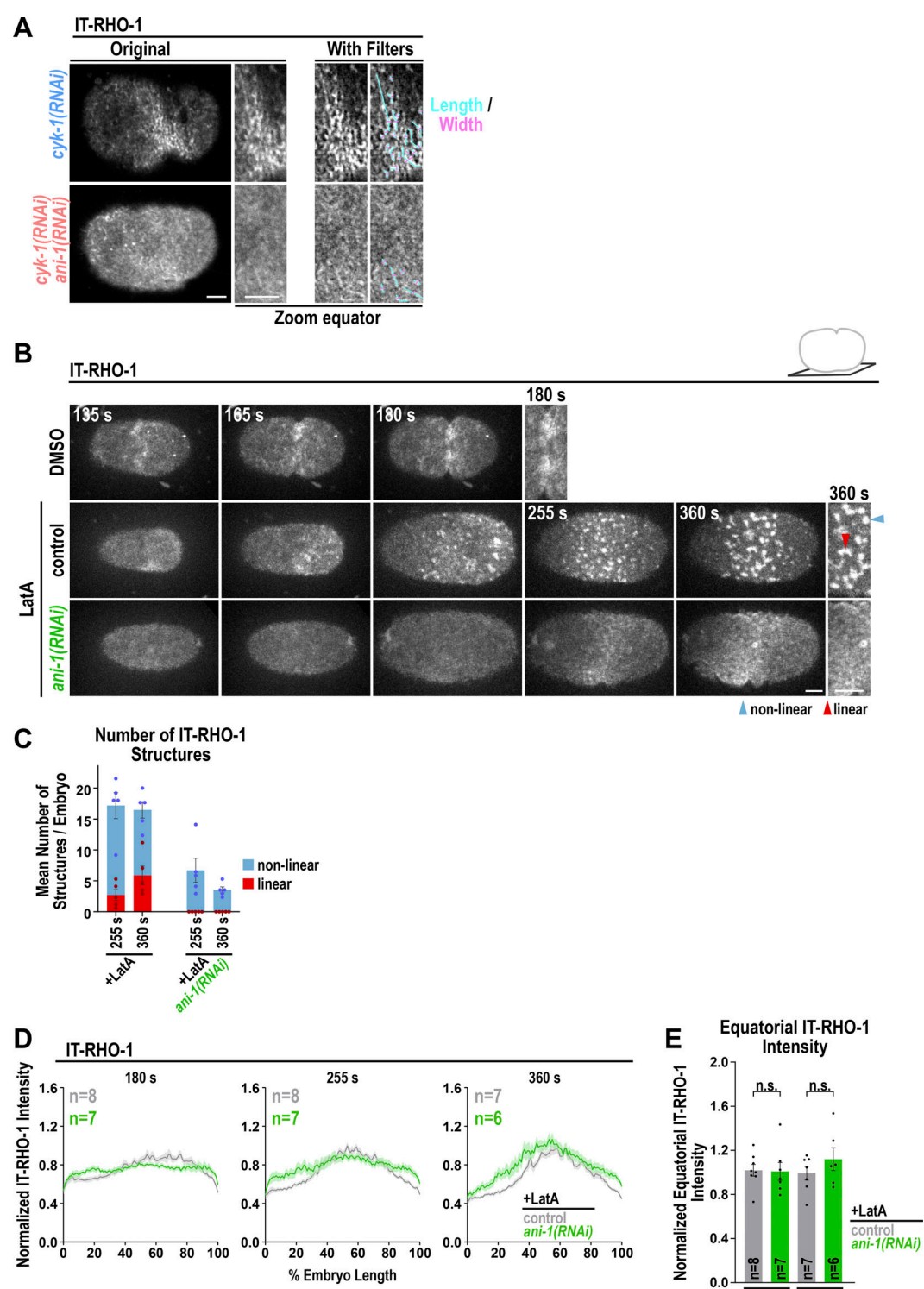

Figure S3.  **Latrunculin A–induced IT-RHO-1 structures depend on ANI-1. (A)** The length (cyan) and width (magenta) of IT-RHO-1 structures were determined at the cell equator after the application of different filters on the original IT-RHO-1 images. Images are reproduced from Fig. 3 A. IT-RHO-1 structures with a length/width ratio of ≥4 were classified as linear and structures with a length/width ratio of <4 as non-linear (Lebedev et al., 2023). **(B)** Maximum intensity projections of 10 cortical z-planes of permeabilized one-cell control and *ani-1(RNAi)* embryos treated with DMSO or Latrunculin A (LatA) as indicated. A zoom-in of the equatorial region for the indicated time points is shown on the right. Scale bars are 5 μm. **(C)** Mean number of linear and non-linear IT-RHO-1 structures at the equatorial region after LatA treatment for control and *ani-1(RNAi)* embryos for indicated time points after NEBD. **(D)** Mean IT-RHO-1 intensity from the anterior to the posterior poles for LatA-treated embryos with and without *ani-1(RNAi)* treatment at indicated time points after NEBD. **(E)** Mean normalized fluorescence intensity of IT-RHO-1 at the cell equator for LatA-treated embryos with and without *ani-1(RNAi)* treatment at indicated time points after NEBD. All error bars are SEM and dots represent data points of individual embryos. P values were calculated using two-tailed Student's *t* test and are n.s. P > 0.05.

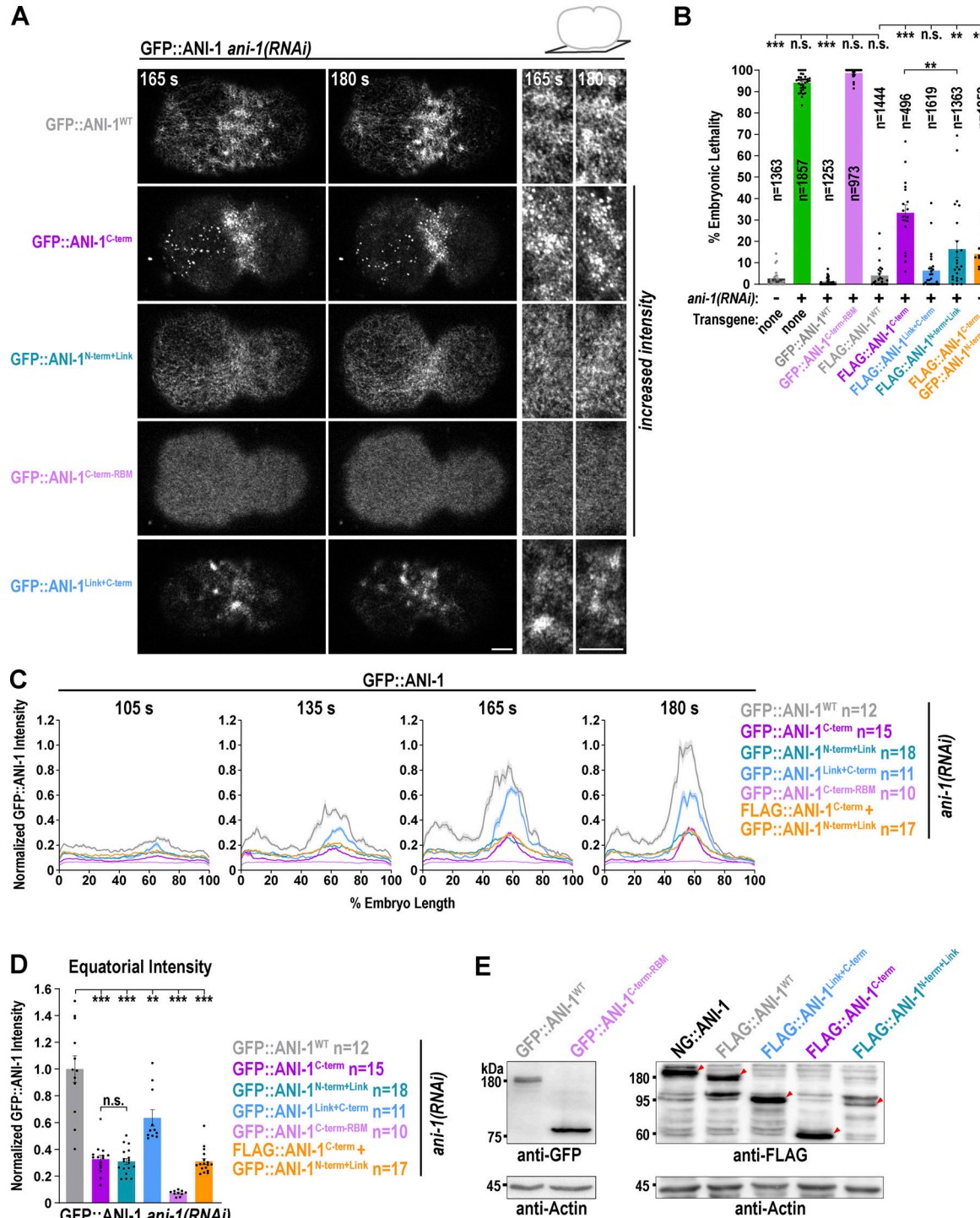

**Figure S4.** **Abolishing RHO-1 binding of ANI-1 prevents cortical enrichment of the C-terminus. (A)** Single z-plane GFP-tagged ANI-1 variants treated with *ani-1(RNAi)*. A zoom-in of the equatorial region for the indicated time points is shown on the right. Since selected ANI-1 mutants exhibited reduced cortical accumulation in comparison to GFP::ANI-1$^{WT}$, the scaling of their fluorescence intensity was increased as indicated. Scale bars are 5 µm. **(B)** Embryonic lethality in % for the indicated transgenes and RNAi conditions. The number of progenies (*n*) counted is indicated. **(C)** Normalized cortical fluorescence intensity of GFP::ANI-1 variants from the anterior (0%) to the posterior (100%) pole at indicated time points. **(D)** Mean normalized fluorescence intensity of GFP::ANI-1 variants at the cell equator for indicated conditions at 180 s after NEBD. **(E)** Immunoblot of indicated GFP::ANI-1 and FLAG::ANI-1 expressing adult hermaphrodites probed with anti-GFP (left) or anti-FLAG (right) and anti-Actin antibodies. The NG::ANI-1 is an in-situ tagged ANI-1 harboring also a FLAG tag. The membrane on the left was cut and the upper part was treated with the anti-GFP and the lower part with the anti-Actin antibodies. Since the FLAG::ANI-1$^{C-term}$ and actin bands were very close to each other, the membrane on the right was first treated with anti-FLAG antibodies and imaged. Afterward, the membrane was washed with buffer and incubated with anti-Actin antibodies. For all, time in seconds (s) after NEBD is indicated, error bars are SEM. P values were calculated using two-tailed Student's *t* test or Mann–Whitney U test and are n.s. P > 0.05, **P < 0.01, and ***P < 0.001. Source data are available for this figure: SourceData FS4.

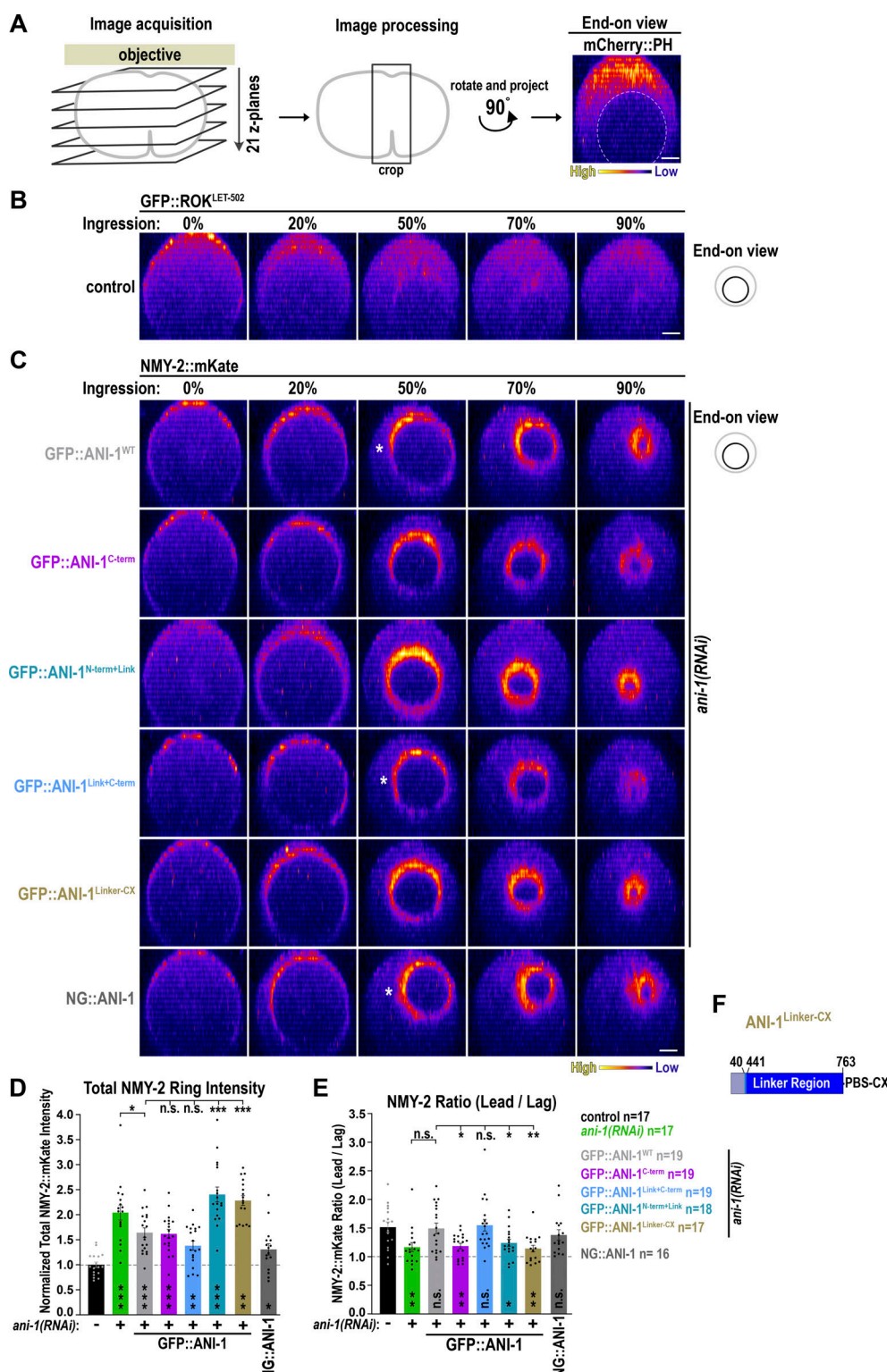

Figure S5. **NMY-2::mKate localization in the ring in embryos expressing different GFP::ANI-1 variants. (A)** To generate an end-on view of the contractile ring 21 z-planes were acquired. The equatorial region was cropped, rotated 90°, and a maximum fluorescence intensity projection was performed. End-on reconstruction of the general membrane marker mCherry::PH is shown. Please note that mCherry::PH intensity is less bright at the bottom of the image away from the objective, due to the decay of the fluorescence signal with increasing distance from the objective. **(B)** End-on reconstruction of GFP::ROK[LET-502] control embryo at the cleavage plane for indicated % of constriction. **(C)** End-on reconstruction of NMY-2::mKate in embryos expressing different GFP::ANI-1 variants treated with *ani-1(RNAi)* or NG::ANI-1 at the cleavage plane for indicated % of constriction. White star marks the leading edge. **(D and E)** Total ring intensity of NMY-2::mKate at 50% ring constriction (D) and ratio of NMY-2::mKate intensity of the leading and lagging edge (E) in embryos expressing different GFP::ANI-1 variants treated with *ani-1(RNAi)* or NG::ANI-1. Error bars are SEM. P values were calculated using two-tailed Student's *t* test or Mann–Whitney U test and are n.s. P > 0.05, *P < 0.05, **P < 0.01, and ***P < 0.001. **(F)** Schematics of the GFP-tagged ANI-1[Linker-CX] fragment of ANI-1. All scale bars are 5 μm.

Video 1.   **Confocal time-lapse movie of the cortical z-planes of one-cell *C. elegans* zygotes expressing NMY-2::mRFP treated with or without *ani-1(RNAi)*.** Images were acquired every 2.5 s on a Nikon Eclipse Ti spinning disk confocal microscope with a 100×1.45-NA Plan-Apochromat oil-immersion objective and an Andor DU-888 X11056 camera. Time in seconds after NEBD is indicated. The playback speed is 8 frames/second.

Video 2.   **Confocal time-lapse movie of the cortical z-planes of one-cell *C. elegans* zygotes expressing GFP::ROK[LET-502] treated with or without *ani-1(RNAi)*.** Images were acquired every 2.5 s on a Nikon Eclipse Ti spinning disk confocal microscope with a 100×1.45-NA Plan-Apochromat oil-immersion objective and an Andor DU-888 X11056 camera. Time in seconds after NEBD is indicated. The playback speed is 8 frames/second.

Video 3.   **Confocal time-lapse movie of the cortical z-planes of one-cell *C. elegans* zygotes expressing IT-RHO-1 treated with indicated RNAi conditions.** Images were acquired every 2.5 s on a Nikon Eclipse Ti spinning disk confocal microscope with a 100×1.45-NA Plan-Apochromat oil-immersion objective and an Andor DU-888 X11056 camera. Time in seconds after NEBD is indicated. The playback speed is 8 frames/second.

Video 4.   **Cortical confocal time-lapse movie of one-cell *C. elegans* zygotes expressing NMY-2::mKate and either no transgene or indicated GFP-tagged ANI-1 variants (not shown) treated with or without *ani-1(RNAi)*.** Images were acquired every 5 s on a Leica TCS SP8 DIVE-FALCON with the 561 nm laser and APO CS2 63×/1.4 oil objective. Time in seconds after NEBD is indicated. The playback speed is 8 frames/second.

Video 5.   **End-on reconstruction of the division plane of one-cell *C. elegans* zygotes expressing NMY-2::mKate treated with or without *ani-1(RNAi)*.** 21 z-plane images were acquired every 15 s on a Leica TCS SP8 DIVE-FALCON with the 561 nm laser and APO CS2 63×/1.4 oil objective. Time in seconds after the onset of furrow ingression is indicated. The playback speed is 8 frames/second.

Video 6.   **End-on reconstruction of the division plane of one-cell *C. elegans* zygotes expressing Formin[CYK-1]::GFP treated with or without *ani-1(RNAi)*.** 21 z-plane images were acquired every 15 s on a Leica TCS SP8 DIVE-FALCON with the 488 nm laser and APO CS2 63×/1.4 oil objective. Time in seconds after the onset of furrow ingression is indicated. The playback speed is 8 frames/second.

Video 7.   **End-on reconstruction of the division plane of one-cell *C. elegans* zygotes expressing IT-RHO-1 treated with or without *ani-1(RNAi)*.** 21 z-plane images were acquired every 15 s on a Leica TCS SP8 DIVE-FALCON with the 488 nm laser and APO CS2 63×/1.4 oil objective. Time in seconds after the onset of furrow ingression is indicated. The playback speed is 8 frames/second.

Video 8.   **End-on reconstruction of the division plane of one-cell *C. elegans* zygotes expressing NMY-2::mKate and the indicated FLAG-tagged or GFP-tagged ANI-1 variants treated with *ani-1(RNAi)*.** 21 z-plane images were acquired every 15 s on a Leica TCS SP8 DIVE-FALCON with the 561 nm laser and APO CS2 63×/1.4 oil objective. Time in seconds after the onset of furrow ingression is indicated. The playback speed is 8 frames/second.

**Provided online are Table S1, Table S2, Table S3, and Table S4. Table S1 contains *C. elegans* strains used in the study. Table S2 shows the information about the dsRNA used in the study. Table S3 contains recombinant DNA used in this study. Table S4 contains DNA oligos used for cloning in this study.**

