## [Peer Review File · The Journal of Cell Biology]

Anillin mediates unilateral furrowing during cytokinesis by limiting RhoA binding to its effectors

Mikhail Lebedev, Fung-Yi Chan, Elisabeth Rackles, Jennifer Bellessem, Tamara Mikeladze-Dvali, Ana Xavier Carvalho, and Esther Zanin

Corresponding Author(s): Esther Zanin, Friedrich-Alexander-Universität Erlangen-Nürnberg

Review Timeline:	Submission Date:	2024-05-31
	Editorial Decision:	2024-07-20
	Revision Received:	2025-02-10
	Editorial Decision:	2025-03-06
	Revision Received:	2025-03-15

Monitoring Editor: William Bement

Scientific Editor: Dan Simon

Transaction Report:

DOI: <https://doi.org/10.1083/jcb.202405182>

July 20, 2024

Re: JCB manuscript #202405182

Prof. Esther Zanin
University of Erlangen-Nuremberg
Department Biologie
Staudtstr. 5
Erlangen 91058
Germany

Dear Esther,

Thank you for submitting your manuscript entitled "Anillin mediates unilateral furrowing during cytokinesis by blocking RhoA binding to its effectors." Your manuscript has been assessed by expert reviewers, whose comments are appended below. Although the reviewers express potential interest in this work, significant concerns unfortunately preclude publication of the current version of the manuscript in JCB.

You will see that the reviewers ask for additional experiments to determine how anillin's disordered linker region senses circumferential flow and how that in turn modulates binding to RhoA resulting in different effects at the leading and lagging edges of the contractile ring. Additionally, they ask for measurements of compression at the lagging edge, confirmation that RHO-1 levels are not affected by anillin depletion, and to incorporate septins into your model and discussion.

Please let us know if you are able to address the major issues outlined above and wish to submit a revised manuscript to JCB. Note that a substantial amount of additional experimental data would be needed to satisfactorily address the concerns of the reviewers and it would likely be necessary to extend your manuscript to a full Research Article. If you decide to revise, we ask that you please first send us a revision plan outlining how you will address the reviewer comments.

The typical timeframe for revisions is three to four months. If you anticipate any difficulties in meeting this aforementioned revision time limit, please contact us and we can work with you to find an appropriate time frame for resubmission. Please note that papers are generally considered through only one revision cycle, so any revised manuscript will either be accepted or rejected.

If you choose to revise and resubmit your manuscript, please also attend to the following editorial points. Please direct any editorial questions to the journal office.

GENERAL GUIDELINES:

Text limits: Character count for an Article is < 40,000, not including spaces. Count includes title page, abstract, introduction, results, discussion, and acknowledgments. Count does not include materials and methods, figure legends, references, tables, or supplemental legends.

Figures: Your manuscript may have up to 10 main text figures. To avoid delays in production, figures must be prepared according to the policies outlined in our Instructions to Authors, under Data Presentation, <https://jcb.rupress.org/site/misc/ifora.xhtml>. All figures in accepted manuscripts will be screened prior to publication.

*****IMPORTANT:** It is JCB policy that if requested, original data images must be made available. Failure to provide original images upon request will result in unavoidable delays in publication. Please ensure that you have access to all original microscopy and blot data images before submitting your revision. ***

Supplemental information: There are strict limits on the allowable amount of supplemental data. Your manuscript may have up to 5 supplemental figures. Up to 10 supplemental videos or flash animations are allowed. A summary of all supplemental material should appear at the end of the Materials and methods section.

Please note that JCB now requires authors to submit Source Data used to generate figures containing gels and Western blots with all revised manuscripts. This Source Data consists of fully uncropped and unprocessed images for each gel/blot displayed in the main and supplemental figures. Since your paper includes cropped gel and/or blot images, please be sure to provide one Source Data file for each figure that contains gels and/or blots along with your revised manuscript files. File names for Source Data figures should be alphanumeric without any spaces or special characters (i.e., SourceDataF#, where F# refers to the associated main figure number or SourceDataFS# for those associated with Supplementary figures). The lanes of the gels/blots should be labeled as they are in the associated figure, the place where cropping was applied should be marked (with a box), and molecular weight/size standards should be labeled wherever possible.

If you choose to resubmit, please include a cover letter addressing the reviewers' comments point by point. Please also highlight all changes in the text of the manuscript.

Regardless of how you choose to proceed, we hope that the comments below will prove constructive as your work progresses. We would be happy to discuss them further once you've had a chance to consider the points raised. You can contact the journal office with any questions at cellbio@rockefeller.edu.

Thank you for thinking of JCB as an appropriate place to publish your work.

Sincerely,

William Bement, PhD
Monitoring Editor
Journal of Cell Biology

Dan Simon, PhD
Scientific Editor
Journal of Cell Biology

Reviewer #1 (Comments to the Authors (Required)):

This interesting study endeavours to provide new insight into Rho regulation by anillin during cytokinesis using the one-cell *C. elegans* embryo system. Anillin is clearly an important regulation of cytokinesis and cellular contractility, but one whose mechanisms of action remain somewhat enigmatic. By virtue of its Rho-binding domain, Anillin has often been interpreted as a conventional Rho effector. But it also supports Rho levels at the cell surface, potentially by using its Rho-binding capacity as the basis for a kinetic scaffold. Of note, the anillin RBD binds to GTP-Rho with relatively low affinity (compared with many other effects), so that its off-rate can promote recycling.

In their experiments Lebedev et al identify a potentially inhibitory effect of anillin on Rho signaling during asymmetric cytokinesis. They show that RhoA effectors (Myosin II, ROCK and formin) are recruited at lower levels at the lagging edge of asymmetric furrows. Further, anillin depletion increases levels of contractile effectors, implying an inhibitory effect on RhoA signaling. This antagonism is attributable to GTP-Rho binding, because it is restored by a C-terminal mutant bearing the RBD, but not when the Rho-binding interface is selectively disrupted. Interestingly, however, Rho-binding is not sufficient to restore the asymmetry of contractile effectors, nor asymmetric contraction of the cytokinetic furrow. The authors show that an additional region, N-terminal to the RBD which is predicted to be unstructured, is necessary to restore asymmetric closure. Together, they propose that the RBD allows anillin to serve as an inhibitor of Rho signaling, but the unstructured domain confers an additional cellular function to mediate spatial asymmetry of furrow closure. Overall, this is an interesting structure-function analysis, which has been carefully performed, and provides useful information. However, I have some reservations about the degree of advance that the information provides.

General comments

1. It is not necessarily surprising that the anillin RBD should act as a Rho antagonist. Its RBD would tend to sequester GTP-Rho and hence prevent active Rho from engaging its effectors. The low-affinity of the Anillin RBD-GTP-Rho interaction mitigates this somewhat, but investigators working with the AHPH location sensor have anecdotally found that overexpression of AHPH can antagonize Rho signaling. Therefore, a key question is how the unstructured domain may modulate the action of the RBD to confer asymmetric inhibition.

2. The authors suggest that the unstructured domain may increase affinity of the RBD for GTP-RhoA. This is plausible, but not tested. I appreciate that this is challenging and would ultimately require both structural as well as cell-based studies. However, one potential experiment might be to measure cortical turnover of Rho (e.g. by FRAP) at different locations (leading edge, lagging edge) in cells expressing different anillin transgenes. One technical issue is that it may not be sufficient to monitor XFP-Rho alone, as it is the GTP-loaded form that will be important. However, it is possible to use GTP-locked Rho mutants for these assays, although usually at low expression levels to avoid gain-of-function effects.

3. The authors' current model works on the assumption that anillin is locally inhibiting Rho - i.e. anillin at the lagging edge will locally sequester Rho or prevent its interaction with effectors. Another possibility to consider is that anillin biases a limited pool of Rho towards the leading edge. If I read it correctly (not to be guaranteed) Fig S5F shows a biased concentration of anillin at the leading edge of the furrow. If the pool of active Rho was limited, anillin at the leading edge would have the potential to deplete Rho from the trailing edge, causing an asymmetry. Moreover, this asymmetry of anillin seems to be restored with inclusion of the unstructured domain. Is it possible that contribution of anillin to asymmetric Rho signaling reflects its concentration of a limited pool of active Rho at the leading edge? In this alternative, the role of the unstructured domain would be primarily to control localization of anillin and it may not need to modulate the affinity of anillin for active Rho. However, concentration of anillin could have avidity effects on Rho sequestration. Have the authors tested the localization of the unstructured region alone?

Tiny points

The references for internal tagging of Rho (Bendezu, Golding) appear to be missing in the reference list?

Reviewer #2 (Comments to the Authors (Required)):

This manuscript examines cytokinesis of the one-cell *C. elegans* embryo and seeks to understand how Anillin/ANI-1 promotes unilateral furrowing, which occurs when the leading edge of a furrow ingresses faster than the trailing edge. Anillin depletion is shown to result in enhanced NMY-2::mRFP and GFP::ROK/LET-502 recruitment to the cell cortex as furrowing proceeds. IT-RHO-1 levels are shown under different conditions of ANI-1 perturbation, including expression of ANI-1 truncations. The effects of these ANI-1 truncations on other RhoA-dependent proteins are also documented. These experiments lead the authors to conclude that binding of the ANI-1/anillin C-terminus to RhoA blocks activation of RhoA effectors at the lagging edge, and that ANI-1 senses compression at the lagging edge through its linker region.

This is a creative attempt at a complex problem and some nice data are presented. However, it is my view that these data, as presented, do not sufficiently support the proposed model and other, more plausible models are not entertained. In particular, the data do not sufficiently support the proposal that ANI-1 inhibits RhoA signalling. Therefore, I cannot recommend publication of the manuscript in its current form.

Major points

It is assumed that increased levels of RhoA-dependent components must mean increased levels of RhoA. However, increased actin and/or myosin II levels upon Anillin depletion could also reflect decreased actin and/or myosin disassembly (which is required for ring closure). Indeed, a role for RhoA-bound Anillin in promoting local actomyosin disassembly has been proposed to occur via translocation of RhoA-bound Anillin out of the ring axis during its closure (Carim et al., 2020, PMID 33117802). It is thus possible that Anillin both promotes the maintenance of RhoA activity, via mechanisms such as those proposed by Budnar et al. (2019), and promotes local actomyosin disassembly at the ring. Therefore, the observations described here do not necessarily "contradict the model that anillin promotes RhoA activity".

The data used to support the conclusion that the linker sequence of ANI-1 senses compression at the lagging edge is weak. In fact, there is no measurement or evidence of compression (other than citing Hsu et al., 2023). Furthermore, it seems that compression ought to be highest at the leading edge, where actomyosin components are enriched? It is also hard to imagine how such a mechanism of compression of the linker region could work.

The data in Fig.S4A/F (GFP::ANI-1 constructs in end-on reconstructions) belong with the data in Fig. 4 A/B (NMY-2::mKate co-expressed in the same embryos) and it seems they should be presented together in a main Figure. However, these data show that lead/lag ratios of NMY-2 (Fig. 4B) and ANI-1 (Fig S4F) are very similar to each other across all conditions of GFP::ANI-1 construct expression. This does not seem to fit the model proposed in Fig. 4E, where there should be high ROK activity (and therefore high NMY-2) and low RhoA-bound ANI-1 at the leading edge, while there should be low ROK activation and high RhoA-bound ANI-1 at the lagging edge (blocked by ANI-1). Rather, it would seem that the high lead/lag ratios of both ANI-1 and NMY-2 in controls, GFP::ANI-1-WT and GFP::ANI-1 Δ 48-460 embryos are consistent with ANI-1 sustaining RhoA activity, leading to high NMY-2 activity at the leading edge, and not with RhoA-bound ANI-1 inhibiting NMY-2 activity at the lagging edge. This interpretation would seem more reconciliatory with prior works in other systems, such as Budnar et al., 2019.

The IT-RHO-1 tool is promising and the data are interesting. While Fig. 2C-E suggests that RHO-1 levels are not altered by ANI-1, this quantification is just from 1 z section. It would be more convincing to examine this in end-on reconstructions of z stacks as done for NMY-2 and ANI-1. On the other hand, Fig. S3, which examines IT-RHO-1 in the presence of LatA, shows robust recruitment of IT-RHO-1 to linear/non-linear structures. The levels of IT-RHO-1 ought to be quantified here too, as they appear to be greatly reduced upon ANI-1 depletion, which is consistent with ANI-1 sustaining elevated levels of RhoA-GTP, at least in this context. NMY-2 also localizes in LatA, so what happens to NMY-2 levels in the presence/absence of ANI-1? Analogous anillin-dependent structures in *Drosophila* cells recruit active myosin II (Hickson & O'Farrell, 2008), which again seems inconsistent with RhoA-bound anillin blocking ROK/myosin II activation.

Anillin-dependent structures in LatA also depend on septins as shown in (Hickson & O'Farrell, 2008 and Lebedev et al., 2023).

Discussion of septins is conspicuously absent from the manuscript and should be included. This is important since septins, like ANI-1, are required for asymmetric furrowing of the *C. elegans* zygote (Maddox et al., 2007). Indeed, the authors showed in their recent paper that the C-terminus of ANI-1 is septin-dependent (Lebedev et al., 2023). Furthermore, analogous anillin RBD mutations to those studied here were recently shown to abrogate septin recruitment to furrows in *Drosophila* and human cells, demonstrating that septin recruitment is specifically RhoA-GTP-binding-dependent (Carim et al. PMID:37378349). Thus, septins should be incorporated into any model for asymmetric furrowing.

Certain statements are written to support a singular interpretation as fact, where there are in fact multiple possible interpretations, e.g. abstract: "The mechanism by which anillin senses the mechanical flow and inhibits myosin II accumulation at the lagging edge is not known". First, it is not clear that anillin itself "senses" flow. Second, it is also not clear whether anillin inhibits myosin II accumulation at the lagging edge, stimulates myosin II accumulation at the leading edge, or both.

The end-on reconstruction data in Fig. 4 and S5 are beautiful. However, I found it difficult to understand precisely where the "leading edge" and "lagging edge" were in the images. The cartoon in Fig. 4B does not appear to be in the same orientation as the images and cartoon in 4A. It would be helpful to have them in the same orientation and to show the boundaries to the edges on the images to allow better appreciation of the quantifications of Lead/ Lag ratios.

GFP-ANI-1 Δ 48-460 in Fig. 4A looks just as asymmetric as all the other conditions except control and GFP-ANI-1-WT. Were Eccentricity measurements quantified at 50% closure only? From the examples show in Fig. 4E and Fig. S5D, at 90% eccentricity none of the mutants appear to rescue, only WT does. Please explain.

Reviewer #3 (Comments to the Authors (Required)):

In this paper, the authors investigate the role of anillin in asymmetrical cytokinesis using *C. elegans* as a model system. They build on a previous paper showing that cytokinetic ingression leads to circumferential cortical flows towards the lagging edge, which causes reductions of Myosin II levels in an anillin dependent manner. Here, they investigate the involvement of anillin. Using different variants and KD approaches, they show convincingly that anillin binds to RhoA, and that the linker region of anillin is involved in the asymmetry of the ring closure. Overall, this is a good study on an interesting topic. I have a few comments that I believe would make the study stronger.

One major limitation of this study is the lack of evidence that what the linker responds to is circumferential flow ("the linker region senses circumferential flows" (line 275)).

Could the author demonstrate this using the same MKLP1 KD that was used in the Hsu 2023? It should lead to cortical flows starting but then stopping. According to the model presented in this paper, one should expect an increase in anillin binding and a decrease in Myosin intensity, followed by the reverse.

The other major issue I have is that images do not always obviously show what is described, and I think the overall quality of the images could be better. For example, figure 1F is described in the text as showing a higher intensity of GFP::ROK compared to the control but this is not obvious from the images. Is it a bad example? (Figure 1C-D and Figure S1C-D which are showing similar things are a lot clearer)

Similarly, in figure 2, the authors comment on linear structures which I do not clearly see in the images on figure 2C. However, from the images, it looks like the intensities between the different treatments are different, but the quantification shows otherwise. Are the images not representative?

In Figure 3B, the intensity differences are interesting, but the main difference I see is the contrast between the dotted phenotype of both the control and the GFP::ANI-1 wt in siRNA context, and all the rest where the localization of Myosin is a lot more diffuse. Could the authors comment?

Others:

In figure 4, I also find it striking that when anillin is depleted the top/bottom asymmetry of the ring seems to be enhanced compared to the control. Could the authors comment?

I am finding the IT-RHO-1 tagging strategy a little unclear. How can the authors conclude that the strategy works well since the band they see in S2C as non specificity is in fact exactly where their band for IT-RHO-1 is? Why do they use such a strategy?

Figure S2E-F are lacking a control (successful cytokinesis in rho-1/rho-1 animals).

Minor:

A scheme showing the different domains of Anillin and the different actors they interact with would be helpful early on (maybe in figure 1 or s1 rather than later).

The text is sometimes difficult to follow for a non specialist. Consider adding more schemes throughout to make it easier.

Figure 1: could the authors comment on the fact that for Myosin the difference between the anillin depleted cells is only clear at

165s, vs for the Formin the difference is already visible at 135s (at least in the presented example)?

Writing could be a little smoother at times eg line 108 ANI-1 mediates [...] by [...] by [...].

Line 109: should "ANI-1 phenotype" be replaced by "phenotype of ANI-1 depletion"?

Point-by-point response to reviewers comments

We thank the reviewers for the positive evaluation and their constructive criticism. We addressed the reviewers comments by multiple new experiments and the key changes of the manuscript are the following:

1) A key question of **reviewer #1 and #2** was whether ANI-1 inhibits or promotes RHO-1 signaling during ring ingression. To address this point, we measured the total levels of the two downstream RHO-1 targets in the ring in *ani-1(RNAi)* embryos. We found that the total levels of formin^{CYK-1} and NMY-2 (myosin II) are increased in the ring in *ani-1(RNAi)* embryos (Fig. 5A-C). Together with the fact that ANI-1 depletion does not reduce the rate of furrow ingression, like RHO-1 depletions do (DOI: 10.1091/mbc.E17-06-0392), it strongly suggests that ANI-1 inhibits rather than promotes RHO-1 signaling.

2) Further **reviewer #1 and #2** asked how ANI-1 depletion affects RHO-1 levels in the ring and RHO-1 dynamics. To answer this, we measured the total levels of RHO-1 in the ring and performed photobleaching experiments with RHO-1 in ANI-1-depleted embryos. We did not observe a difference in the RHO-1 levels (Fig. 5B, D) or RHO-1 dynamics (Fig. 3E, F). Together with our findings that RHO-1 effectors are increased after ANI-1 depletion, these results support our model that ANI-1 is able to block the effector binding site of RHO-1 and inhibits RHO-1 signaling.

3) Our model predicts that cortical actin flows affect ANI-1 function differentially at the leading and lagging edge. **Reviewer #2** asked us to measure cortical flows and the resulting equatorial compression, and to compare these at the leading and lagging cortices. We performed particle image velocimetry (PIV) analysis of the actin cortex and found that bidirectional cortical flows from the cell poles towards the furrow region are very strong at the leading edge and almost absent at the lagging edge at the same time points (Fig. 9A-E), confirming published findings (DOI: 10.7554/eLife.36073). Consistent with the strong converging flows at the leading edge, we also measured higher equatorial compression at the leading than the lagging cortex (Fig. 9F-I). Based on our measurements, we now revised our model and suggest that high cortical flows at the leading edge limit ANI-1 binding to active RHO-1.

4) A point criticized by **reviewer #1, #2, and #3** was the absence of a molecular mechanism how the linker of ANI-1 could influence the interaction between the RBD and active RHO-1. We addressed this point by comparing the binding of the ANI-1 RBD with and without the linker region (ANI-1^{Link+C-term} and ANI-1^{C-term}) to active RHO-1. Interestingly we observed that the ANI-1 C-terminus with the linker region attached bound more efficiently to active RHO-1 than the C-terminus without the linker. However, the linker itself did not bind to active RHO-1 (Fig. 2C). This suggests that the linker enhances the binding of the RBD to active RHO-1.

5) One critical question asked by **reviewer #2** was whether ANI-1 acts at the leading, lagging or both edges. To answer this, we performed a detailed analysis of NMY-2 intensity at the leading and lagging edge after ANI-1 depletion. We found that NMY-2 intensity increases on both edges but stronger at the lagging than the leading edge, suggesting that inhibition of RHO-1 signaling by ANI-1 is stronger at the lagging edge (Fig. 6D). Using our genetic-replacement system we then asked which ANI-1 variant fully restored NMY-2 levels. For this we generated a new set of strains expressing FLAG-tagged ANI-1 variants. In embryos expressing the FLAG::ANI-1^{C-term}, NMY-2 levels were fully restored to control levels at the leading edge, but they were still elevated at the lagging edge (Fig. 6D). Only when the linker region was fused to the ANI-1 C-terminus (FLAG::ANI-1^{Link+C-term}), NMY-2 levels became like in control embryos at the lagging edge.

Together with our finding that the linker enhances active RHO-1 binding to the RBD *in vitro*, we propose that the linker is particularly important at the lagging edge to suppress RHO-1 signaling and enforce asymmetry of RHO-1 signaling.

Below you find a point-by-point response to the reviewer's comments:

Reviewer #1:

This interesting study endeavours to provide new insight into Rho regulation by anillin during cytokinesis using the one-cell *C. elegans* embryo system. Anillin is clearly an important regulation of cytokinesis and cellular contractility, but one whose mechanisms of action remain somewhat enigmatic. By virtue of its Rho-binding domain, Anillin has often been interpreted as a conventional Rho effector. But it also supports Rho levels at the cell surface, potentially by using its Rho-binding capacity as the basis for a kinetic scaffold. Of note, the anillin RBD binds to GTP-Rho with relatively low affinity (compared with many other effects), so that its off-rate can promote recycling.

In their experiments Lebedev et al identify a potentially inhibitory effect of anillin on Rho signaling during asymmetric cytokinesis. They show that RhoA effectors (Myosin II, ROCK and formin) are recruited at lower levels at the lagging edge of asymmetric furrows. Further, anillin depletion increases levels of contractile effectors, implying an inhibitory effect on RhoA signaling. This antagonism is attributable to GTP-Rho binding, because it is restored by a C-terminal mutant bearing the RBD, but not when the Rho-binding interface is selectively disrupted. Interestingly, however, Rho-binding is not sufficient to restore the asymmetry of contractile effectors, nor asymmetric contraction of the cytokinetic furrow. The authors show that an additional region, N-terminal to the RBD which is predicted to be unstructured, is necessary to restore asymmetric closure. Together, they propose that the RBD allows anillin to serve as an inhibitor of Rho signaling, but the unstructured domain confers an additional cellular function to mediate spatial asymmetry of furrow closure. Overall, this is an interesting structure-function analysis, which has been carefully performed, and provides useful information. However, I have some reservations about the degree of advance that the information provides.

General comments

1. It is not necessarily surprising that the anillin RBD should act as a Rho antagonist. Its RBD would tend to sequester GTP-Rho and hence prevent active Rho from engaging its effectors. The low-affinity of the Anillin RBD-GTP-Rho interaction mitigates this somewhat, but investigators working with the AHPH location sensor have anecdotally found that overexpression of AHPH can antagonize Rho signaling. Therefore, a key question is how the unstructured domain may modulate the action of the RBD to confer asymmetric inhibition.

Answer: We agree with the reviewer that this is a key question and we addressed this in multiple complementary approaches. First, we tested whether active RHO-1 binding of the ANI-1's RBD is influenced by the presence of the linker region. Using bacterial purified proteins we compared RHO-1 binding to the ANI-1 C-terminus with and without the linker. We observed that the interaction between active RHO-1 and the ANI-1 C-terminus is stronger when the linker is present (Fig. 2C). However, we did not observe an interaction between the linker and RHO-1. This suggests that the linker does not harbor a second RHO-1 binding site but rather enhances RHO-1 binding of the C-terminus.

In vivo we observed that during ring assembly (Fig. 4D) and constriction (Fig. 6A), NMY-2 levels are lowered by the ANI-1^{C-term} and ANI-1^{Link+C-term} with similar efficiencies. However, at the lagging edge only ANI-1^{Link+C-term} fully reduced NMY-2 levels back to the levels of control embryos (Fig. 6D). Together these suggest that the RBD containing C-terminus acts as a RHO-1 antagonist and inhibits RHO-1 signaling and that binding of the C-terminus to RHO-1 can be enhanced by the linker region, which is especially important at the lagging edge.

2. The authors suggest that the unstructured domain may increase affinity of the RBD for GTP-RhoA. This is plausible, but not tested. I appreciate that this is challenging and would ultimately require both structural as well as cell-based studies. However, one potential experiment might be to measure cortical turnover of Rho (e.g. by FRAP) at different locations (leading edge, lagging edge) in cells expressing different anillin transgenes. One technical issue is that it may not be sufficient to monitor XFP-Rho alone, as it is the GTP-loaded form that will be important. However, it is possible to use GTP-locked Rho mutants for these assays, although usually at low expression levels to avoid gain-of-function effects.

Answer: We agree with the reviewer that determining the dynamics of RhoA after anillin RNAi is very important. We aimed to express GTP-locked RHO-1 (RHO-1^{Q63L}) as an integrated transgene using its own regulatory 3' and 5' sequences, however we never obtained any viable *C. elegans* strains. This is most likely due to the dominant negative effect of GTP-locked RHO-1^{Q63L} on animal development. Unfortunately, there is no system that allows us to induce expression of RHO-1^{Q63L} in the gonad or early embryo and therefore we decided to use wild type IT-RHO-1 for this experiment. As the reviewer suggested it would be ideal to FRAP RHO-1 at the leading and lagging edges, however also this is technically not possible, since we image the embryos from the side and do not have an end-on view of the ring in the one-cell embryo during imaging. In previous work NMY-2 was bleached in the entire ring (DOI: 10.7554/eLife.36073) but since the position of the ring is constantly changing during ring ingression, it is very difficult to precisely bleach and measure fluorescence intensity over time. Therefore, we decided to bleach IT-RHO-1 on the membrane of the newly formed daughter cells directly after furrow ingression. Our FRAP analysis revealed that IT-RHO-1 is highly dynamic with a half-time of ~4.1 s and mobile fraction of ~80% (Fig. 3E, F) which is similar to previous observations using N-terminal GFP-tagged RhoA-WT and RhoA-Q63L (DOI: 10.1016/j.devcel.2019.04.031). After *ani-1(RNAi)* we observed no difference in the recovery half-time or the mobile fraction of IT-RHO-1 (Fig. 3E, F), suggesting that ANI-1 does not influence the dynamics of RHO-1.

3. The authors' current model works on the assumption that anillin is locally inhibiting Rho - i.e. anillin at the lagging edge will locally sequester Rho or prevent its interaction with effectors. Another possibility to consider is that anillin biases a limited pool of Rho towards the leading edge. If I read it correctly (not to be guaranteed) Fig S5F shows a biased concentration of anillin at the leading edge of the furrow. If the pool of active Rho was limited, anillin at the leading edge would have the potential to deplete Rho from the trailing edge, causing an asymmetry. Moreover, this asymmetry of anillin seems to be restored with inclusion of the unstructured domain. Is it possible that contribution of anillin to asymmetric Rho signaling reflects its concentration of a limited pool of active Rho at the leading edge? In this alternative, the role of the unstructured domain would be primarily to control localization of anillin and it may not need to modulate the affinity of anillin for active Rho. However, concentration of anillin could have avidity effects on Rho sequestration. Have the authors tested the localization of the unstructured region alone?

Answer: Previous work showed (DOI: 10.1016/j.devcel.2007.02.018) and we confirmed that ANI-1 is enriched at the leading edge (Fig. 7A, C). Therefore, it is a possibility that ANI-1 sequesters active RHO-1 at the leading edge. If that would be the case, the total levels of RHO-1 effectors in the ring should be the same after ANI-1 depletion, since the same and limited amount of active RHO-1 would distribute around the ring. To test this, we measured the total levels of NMY-2 and formin^{CYK-1} in the contractile ring after *ani-1(RNAi)*. We observed that the total levels of NMY-2 and formin^{CYK-1} in the ring increase in *ani-1(RNAi)* embryos (Fig. 5A-C). Indeed, we find that adding the linker region to the C-terminus restored asymmetric distribution of ANI-1 in the ring and therefore the linker could target ANI-1 to the leading edge. To address

whether the linker region of ANI-1 is sufficient for an asymmetric localization of ANI-1 in the ring we analyzed the localization of two ANI-1 variants harboring the linker in the ring. We chose the ANI-1 N-terminus fused to the linker region (ANI-1^{N-term+Link}) and the linker alone targeted to the plasma membrane (ANI-1^{Linker-CX}). We added a membrane tether to the linker region since the ANI-1^{N-term+Link} exhibited a weak cortical accumulation in our previous work (DOI: 10.1016/j.celrep.2023.113076). Since ANI-1^{N-term+Link} and ANI-1^{Linker-CX} did not restore unilateral furrowing (Fig. 8), we determined their localization in the ring with and without depleting endogenous ANI-1. We observed that the ANI-1^{N-term+Link} and ANI-1^{Linker-CX} were weakly localized to the constricting ring but did not show an asymmetric distribution in the ring with or without *ani-1(RNAi)* (Fig. 7).

Together our findings do not support the hypothesis that the linker region targets ANI-1 to the leading edge and helps to sequester a limited pool of active RHO-1 at this location. Instead, we propose that the linker facilitates binding of the RBD to active RHO-1 at the lagging edge. As a consequence, RHO-1 signaling, furrow ingression and actin flows are reduced at the lagging edge and therefore ANI-1 levels are lower at the lagging edge.

Tiny points

The references for internal tagging of Rho (Bendezu, Golding) appear to be missing in the reference list?

Answer: Thanks for pointing this out, we added it to the reference list.

Reviewer #2:

This manuscript examines cytokinesis of the one-cell *C. elegans* embryo and seeks to understand how Anillin/ANI-1 promotes unilateral furrowing, which occurs when the leading edge of a furrow ingresses faster than the trailing edge. Anillin depletion is shown to result in enhanced NMY-2::mRFP and GFP::ROK/LET-502 recruitment to the cell cortex as furrowing proceeds. IT-RHO-1 levels are shown under different conditions of ANI-1 perturbation, including expression of ANI-1 truncations. The effects of these ANI-1 truncations on other RhoA-dependent proteins are also documented. These experiments lead the authors to conclude that binding of the ANI-1/anillin C-terminus to RhoA blocks activation of RhoA effectors at the lagging edge, and that ANI-1 senses compression at the lagging edge through its linker region.

This is a creative attempt at a complex problem and some nice data are presented. However, it is my view that these data, as presented, do not sufficiently support the proposed model and other, more plausible models are not entertained. In particular, the data do not sufficiently support the proposal that ANI-1 inhibits RhoA signalling. Therefore, I cannot recommend publication of the manuscript in its current form.

Major points

1. It is assumed that increased levels of RhoA-dependent components must mean increased levels of RhoA. However, increased actin and/or myosin II levels upon Anillin depletion could also reflect decreased actin and/or myosin disassembly (which is required for ring closure). Indeed, a role for RhoA-bound Anillin in promoting local actomyosin disassembly has been proposed to occur via translocation of RhoA-bound Anillin out of the ring axis during its closure (Carim et al., 2020, PMID 33117802). It is thus possible that Anillin both promotes the maintenance of RhoA activity, via mechanisms such as those proposed by Budnar et al. (2019), and promotes local actomyosin disassembly at the ring. Therefore, the observations described here do not necessarily "contradict the model that anillin promotes RhoA activity".

Answer: We thank the reviewer for bringing up the idea that anillin could limit the accumulation of ring components not by inhibiting RhoA signaling but rather by promoting the disassembly of the ring. However, we think that several observations made by others and us argue against this possibility. First, although removal of ring components by membrane shedding is observed in *Drosophila* cells, this has not been noticed by us or others in the one-cell *C. elegans* zygote, although we frequently image membrane and ring markers during cytokinesis. Second, if ANI-1 would be required for ring disassembly (DOI: 10.1016/j.cell.2009.03.021), the rate of furrow ingression should be reduced in ANI-1-depleted embryos. However, in embryos with strong ANI-1 depletion furrow ingression is only weakly delayed (Fig. 1B in DOI: 10.1016/j.devcel.2007.02.018) and furrowing ingression rates are normal (Fig. 2 in DOI: 10.1091/mbc.E17-06-0392). Third, we observe an increase in the cortical levels of RHO-1 effectors not only during ring constriction but also during ring assembly (Figs. 1, S1), when ring disassembly should not have been initiated, yet. Fourth, we find that the putative MBD and ABD of ANI-1 are not required for unilateral furrowing and therefore a destabilization of the actin-myosin network seems rather unlikely. Fifth, ANI-1 was shown to stabilize and crosslinks actin networks (DOI: 10.1016/j.cub.2015.02.072) and, to our knowledge, there are no reports that anillin destabilizes actin networks. Taken together, we consider the possibility that ANI-1 regulates unilateral furrowing by inhibiting ring disassembly or shedding of membrane vesicles rather unlikely. However, we now included this possibility in the discussion (lines 518-528).

2. The data used to support the conclusion that the linker sequence of ANI-1 senses compression at the lagging edge is weak. In fact, there is no measurement or evidence of compression (other

than citing Hsu et al., 2023). Furthermore, it seems that compression ought to be highest at the leading edge, where actomyosin components are enriched? It is also hard to imagine how such a mechanism of compression of the linker region could work.

Answer: We thank the reviewer for pointing out the importance of the cortical flows and the resulting compression for our model. We included a summary of the current literature (line 51-62; 105-116) and also performed cortical flow measurements to confirm and expand previous findings. During furrowing cortical actin flows bidirectionally from the anterior and posterior poles towards the furrow region (Fig. 9A, C) (DOI: 10.7554/eLife.17807; 10.1242/jcs.231357, 10.3389/fcell.2020.573393; 10.7554/eLife.36073). A detailed study by the Oegema laboratory compared this furrow-directed flows at the cortices of the leading and lagging edge and found that flows are first detected at the leading cortex and subsequently at the lagging cortex (Figs. 1, 3 in DOI: 10.7554/eLife.36073). Since the anterior and posterior flows converge at the furrow region they are expected to cause a compression of this area. They further estimated cortex compression within the furrow plane during ring closure (Fig. 3 in DOI: 10.7554/eLife.36073). During furrow ingression actin flows cannot be measured in the furrow plane since the cortex turns inward and away from the imaging plane. Therefore, they estimated compression during ring ingression by comparing the area of the cortex flowing into the furrow with the area of the newly generated furrow area. This showed that the area flowing into the furrow region is about two times larger than the newly generated area therefore suggesting a constant compression of the actin cortex also during ring ingression.

In the study of the Oegema laboratory embryos were filmed under physical confinement between an agarose pad and a coverslip. Physical confinement of the embryos generates a unidirectional rotational flow along the transverse axis of the embryos (DOI: 10.1242/jcs.231357, 10.3389/fcell.2020.573393), which can influence the analysis. Therefore, we repeated the experiments using our experimental set up without any confinement of the embryo. We measured flow velocities of the actin cortex at the leading and lagging cortex using PIV of GFP-tagged NMY-2. Our measurements confirmed previous observations that strong furrow-directed flows are first observed at the cortex of the leading edge (Fig. 9A-E). Cumulatively this predicts that compression is stronger at the equatorial cortex of the leading edge than the lagging edge. Since our embryos were not physically confined during image acquisition, rotational flows along the transverse axis of the embryos were strongly reduced (DOI: 10.3389/fcell.2020.573393; 10.1242/jcs.231357).

A recent publication discovered a novel type of circumferential flows along the transverse axis away from the leading towards the lagging edge (Reviewer 2, Figure A a) (DOI: 10.1038/s41467-023-43996-4). To consider these flows in our analysis, we measured flow velocities along the transverse axis of the embryo at the cell equator. We found that the transverse flow velocities were much lower than the A-P flow velocities (Fig. 9D, E and Reviewer 2, Figure A b, c)). In case of a bidirectional flow away from the leading cortex, we expect a switch from negative to positive flow velocity values along the transverse axis (Reviewer 2, Figure 1A). However, the measured flow velocities did not exhibit this pattern and instead were positive along most of the transverse axis. A bidirectional flow towards the lagging cortex is expected to have the opposite order of flow directions, with positive flow velocities changing into negative flow velocities (Reviewer 2, Figure A a)). We observed a very weak bidirectional flow at -15 s but not at -10 s at the lagging cortex (Reviewer 2, Figure A c)). Thus, with our measurements we did not find strong evidence for a persistent circumferential flow from the leading towards the lagging edge. If the reviewer thinks this is important for the reader to know, we are happy to include these data in the paper. However, we feel it is not highly relevant and unnecessarily complicates the story, since with the divergence measurements we are presenting in Fig. 9F-I take into account all flows irrespective of their direction.

To estimate the compression of the actin cortex by the various flows, we measured the divergence of the cortical flows. Negative divergence values indicate equatorial compression and positive divergence values indicate equatorial expansion. Indeed, we find that the convergence is more negative at the cortex of leading than the lagging edge prior to furrow ingression of the lagging edge (Fig. 9F-I). This confirms our observation that cortical A-P flows are strong at the leading cortex and demonstrates that compression is higher at the leading than the lagging cortex.

Based on these measurements we revised our model and propose that at the leading edge RHO-1 binding by the RBD is weakened due to strong cortical actin flows. Together with our new *in vitro* data that the linker region enhances RHO-1 binding to the RBD and that the function of the linker is especially important at the lagging edge, we now discuss two possibilities for how high cortical flows could attenuate linker function. High cortical flows cause an enrichment of ANI-1 at the leading edge and it is possible that above a certain concentration threshold ANI-1 self-interacts via its linker region and this hinders the positive effect that the linker has on the RBD-RHO-1 interaction. Alternatively, if the linker is connected to the flowing actin cortex by binding actin or an actin binding protein, this could drag the linker away from the membrane bound RBD and thereby inhibit its function. It will be interesting to follow up on these possibilities in the future, however, this goes beyond the scope this study.

3. The data in Fig.S4A/F (GFP::ANI-1 constructs in end-on reconstructions) belong with the data in Fig. 4 A/B (NMY-2::mKate co-expressed in the same embryos) and it seems they should be presented together in a main Figure. However, these data show that lead/lag ratios of NMY-2 (Fig. 4B) and ANI-1 (Fig S4F) are very similar to each other across all conditions of GFP::ANI-1 construct expression. This does not seem to fit the model proposed in Fig. 4E, where there should be high ROK activity (and therefore high NMY-2) and low RhoA-bound ANI-1 at the leading edge, while there should be low ROK activation and high RhoA-bound ANI-1 at the lagging edge (blocked by ANI-1). Rather, it would seem that the high lead/lag ratios of both ANI-1 and NMY-2 in controls, GFP::ANI-1-WT and GFP::ANI-1 Δ 48-460 embryos are consistent with ANI-1 sustaining RhoA activity, leading to high NMY-2 activity at the leading edge, and not with RhoA-bound ANI-1 inhibiting NMY-2 activity at the lagging edge. This interpretation would seem more reconciliatory with prior works in other systems, such as Budnar et al., 2019.

Answer: We agree with the reviewer that based on the co-enrichment of ANI-1 and active RHO-1 at the leading edge a model where ANI-1 stabilizes RHO-1 at the leading edge seems obvious. But although we performed similar experiments as in Budnar et al. 2019, we did not find evidence that ANI-1 promotes RHO-1 signaling during cytokinesis in *C. elegans* one-cell embryo. First, Budnar et al. reported that cortical levels of RhoA effectors and RhoA itself are decreased after anillin depletion. Already in the first version of the manuscript we showed that ROK, NMY-2 and formin levels are increased during ring assembly in *ani-1(RNAi)* embryos (Fig. 1, S1). We are now including new data showing that the total levels of NMY-2 and formin^{CYK-1} are also elevated in the ring during constriction (Fig. 5A-C). We also measured the cortical levels of IT-RHO-1 during ring assembly (Fig. 3A-C) and in the revised version also during ring constriction, and for both time points we found no reduction of IT-RHO-1 levels (Fig. 5B, D). Second, Budnar et al. demonstrated, using photobleaching experiments, that after anillin depletion RhoA is much more dynamic with a shorter recovery time and decreased immobile fraction. For the revision we performed similar bleaching experiments on IT-RHO-1 and found no change in the dynamics of IT-RHO-1 after ANI-1 depletion (Fig. 3E, F).

Since our new data showed that total NMY-2 levels increased in the ring in *ani-1(RNAi)* embryos during furrowing, we asked whether they rise equally at the leading and lagging edge. Interestingly, NMY-2 levels increased at both the leading and the lagging edge but their elevation at the lagging edge was more pronounced (Fig. 6D). Taking into account the lower protein concentration of ANI-1 at the lagging edge, this suggests that ANI-1 inhibits RHO-1 signaling stronger at the lagging than the leading edge. Since our GFP-tagged ANI-1 lines did not fully restore total NMY-2 levels in the ring (Fig. S5E), we also build several FLAG-tagged ANI-1 strains during the revision to test the function of the different ANI-1 variants. The ANI-1^{C-term} fragment lowered total NMY-2 levels in the ring and at the leading and lagging edge (Fig. 6A, D). However, at the lagging edge NMY-2 levels were still higher in comparison to control or ANI-1^{WT} expressing embryos (Fig. 6D). Only with the linker region attached to the C-terminus (ANI-1^{Link+C-term}) were NMY-2 levels fully restored and indistinguishable from control or ANI-1^{WT} expressing embryos at the lagging edge. Together, this suggests that ANI-1 inhibits RHO-1 signaling at the leading and lagging edge with different strengths and that this is modulated by the linker. At the leading edge the ANI-1^{C-term} is sufficient to reduce NMY-2 levels back to control conditions suggesting that the linker function is dispensable. In contrast, at the lagging edge the linker region and the C-terminus had to be both present (ANI-1^{Link+C-term}) to completely reduce NMY-2 levels back to control levels, suggesting that the linker must enhance binding of the ANI-1^{C-term} to RHO-1 at the lagging edge. Based on all these observations we conclude that ANI-1 inhibits RHO-1 signaling especially at the lagging edge where its protein concentrations are low. Like suggested by the reviewer, we moved the localization data of ANI-1 to the main Figure 7 to highlight the difference in ANI-1

concentration. Moreover, in order to summarize our data, we depicted three ANI-1 proteins at the leading and two at the lagging edge in our model in Figure 10.

To reconcile the Budnar et al. and our data, we propose that the effect of the RBD on active RhoA is modulated in opposite ways by the C2 domain and the linker region. The C2 domain maintains RhoA after RBD binding in PIP2 clusters and this increases the residence time of active RhoA on the membrane (DOI: 10.1016/j.devcel.2019.04.031). The linker region, in turn, strengthens the binding of the RBD to RhoA, sequestering active RhoA away from its effectors. This turns anillin into a negative regulator of RhoA activity (lines 539-558). Whether both functions exist in the same anillin protein and are spatially and temporally controlled or whether in some species the positive and in others the negative effect on RhoA is dominant, needs to be determined in the future.

4. The IT-RHO-1 tool is promising and the data are interesting. While Fig. 2C-E suggests that RHO-1 levels are not altered by ANI-1, this quantification is just from 1 z section. It would be more convincing to examine this in end-on reconstructions of z stacks as done for NMY-2 and ANI-1. On the other hand, Fig. S3, which examines IT-RHO-1 in the presence of Lata, shows robust recruitment of IT-RHO-1 to linear/non-linear structures. The levels of IT-RHO-1 ought to be quantified here too, as they appear to be greatly reduced upon ANI-1 depletion, which is consistent with ANI-1 sustaining elevated levels of RhoA-GTP, at least in this context. NMY-2 also localizes in Lata, so what happens to NMY-2 levels in the presence/absence of ANI-1? Analogous anillin-dependent structures in *Drosophila* cells recruit active myosin II (Hickson & O'Farrell, 2008), which again seems inconsistent with RhoA-bound anillin blocking Rok/myosin II activation.

Answer: To address the comments of the reviewer we analyzed in more depth the levels and dynamics of IT-RHO-1. First, we measured the levels of IT-RHO-1 in the ring during constriction using end-on reconstructions and found no difference between control and *ani-1(RNAi)* treated embryos (Fig. 5B, D). Second, we also performed FRAP experiments on IT-RHO-1 and again observed that ANI-1 depletion had no effect on IT-RHO-1 dynamics (Fig. 3E, F). Third, we quantified IT-RHO-1 levels in Latrunculin A treated embryos with and without *ani-1(RNAi)* and found that IT-RHO-1 levels were not reduced after *ani-1(RNAi)* even in the absence of F-actin (Fig. S3D, E). Finally, we analyzed NMY-2 localization in Latrunculin A treated embryos depleted of ANI-1. In *ani-1(RNAi)* embryos treated with Latrunculin A NMY-2 still localized in defined structures at the membrane and at late time points (360 s after NEBD) cortical NMY-2 levels were elevated (Reviewer 2, Figure B). This is supported by our previous study where we found that NMY-2 still localized to structures on the membrane in embryos expressing the ANI-1^{C-term} (lacking the MBD). Consistent with the fact that the MBD domain is missing in the ANI-1^{C-term} truncation, the structures of NMY-2 and the ANI-1^{C-term} did not co-localize in Latrunculin A-treated embryos (Fig. 4A-C in DOI: 10.1016/j.celrep.2023.113076). We feel that adding the NMY-2 localization in Latrunculin A-treated embryos is not really necessary for our conclusions and would require extensive explanations that would disrupt the flow of the text. Therefore, in the interest of space we prefer to leave this data out. However, if the reviewer is convinced that this data is indispensable for the reader, we are happy to incorporate it into the manuscript.

5. Anillin-dependent structures in LatA also depend on septins as shown in (Hickson & O'Farrell, 2008 and Lebedev et al., 2023). Discussion of septins is conspicuously absent from the manuscript and should be included. This is important since septins, like ANI-1, are required for asymmetric furrowing of the *C. elegans* zygote (Maddox et al., 2007). Indeed, the authors showed in their recent paper that the C-terminus of ANI-1 is septin-dependent (Lebedev et al., 2023). Furthermore, analogous anillin RBD mutations to those studied here were recently shown to abrogate septin recruitment to furrows in *Drosophila* and human cells, demonstrating that septin recruitment is specifically RhoA-GTP-binding-dependent (Carim et al. PMID:37378349). Thus, septins should be incorporated into any model for asymmetric furrowing.

Answer: We agree with the reviewer that the role of septins in unilateral furrowing is important but currently not clear. Since septin depletion causes a ~30% reduction of ANI-1 during ring assembly (Fig. S3B in DOI: 10.1016/j.celrep.2023.113076) one possibility is that the main function of septins is to stabilize enough ANI-1 on the membrane. Alternatively, septins could have a more active role

in unilateral furrowing since they form higher order clusters on the membrane. Septins also promote asymmetric accumulation of ANI-1 in the ring (DOI: 10.1016/j.devcel.2007.02.018) and together with the fact that septins form oligomers they could induce clustering of ANI-1 and hinder linker function at the leading edge by an unknown mechanism. As suggested by the reviewer we added those points on septin, the RBD function, and the similarities to *Drosophila* anillin in the text (lines 257-260, 595-604).

6. Certain statements are written to support a singular interpretation as fact, where there are in fact multiple possible interpretations, e.g. abstract: "The mechanism by which anillin senses the mechanical flow and inhibits myosin II accumulation at the lagging edge is not known". First, it is not clear that anillin itself "senses" flow. Second, it is also not clear whether anillin inhibits myosin II accumulation at the lagging edge, stimulates myosin II accumulation at the leading edge, or both.

Answer: We carefully revised our wording and removed all references suggesting a sensing mechanism by anillin. For the second point, we hope that our new data convinces the reviewer that ANI-1 inhibits RHO-1 signaling in the ring. We now demonstrate that: 1) the total levels of NMY-2 and formin are increased in the ring after *ani-1(RNAi)* (Fig. 5A-C); 2) the total levels of IT-RHO-1 in the ring and IT-RHO-1 dynamics are not influenced by ANI-1 (Fig. 3E, F; Fig 5B, D); 3) the levels of NMY-2 increase on the leading and lagging edge, and that the increase is more pronounced at the lagging edge (Fig. 6D); 4) the presence of the ANI-1^{C-term} lowers NMY-2 levels at the leading and lagging edge, but that the linker region must be present for a full reduction of NMY-2 levels specifically at the lagging edge (Fig. 6D).

7. The end-on reconstruction data in Fig. 4 and S5 are beautiful. However, I found it difficult to understand precisely where the "leading edge" and "lagging edge" were in the images. The cartoon in Fig. 4B does not appear to be in the same orientation as the images and cartoon in 4A. It would be helpful to have them in the same orientation and to show the boundaries to the edges on the images to allow better appreciation of the quantifications of Lead/ Lag ratios.

Answer: We changed the left-right orientation of all unilaterally ingressing embryos with the leading edge oriented to the left side as depicted in the cartoon Fig. 6C. We also marked the leading edge with a white star in the images at 50% ingression and explained better how the leading and lagging ring intensity were quantified (Fig. 6C, lines 807-810, 1151-1161). We hope that those changes make is more accessible for the reader.

8. GFP-ANI-1 Δ 48-460 in Fig. 4A looks just as asymmetric as all the other conditions except control and GFP-ANI-1-WT. Were Eccentricity measurements quantified at 50% closure only? From the examples show in Fig. 4E and Fig. S5D, at 90% eccentricity none of the mutants appear to rescue, only WT does. Please explain.

Answer: We measured the eccentricity from 0% to 90% ring ingression and also in control and ANI-1^{WT} expressing embryos the ring center moves closer towards the initial ring center at late ingression time points (Fig. 8A). We compared the eccentricity at 90% ingression between control embryos and the FLAG::ANI-1 Δ 48-460 (now named FLAG::ANI-1^{Link+C-term}) expressing embryos and found that indeed the eccentricity of the FLAG::ANI-1^{Link+C-term} embryos was a bit a lower (meaning closer to the initial ring center) than in control embryos and embryos expressing FLAG::ANI-1^{WT} (Reviewer 2, Figure C). We speculate that this movement of the ring center towards initial ring center at the late time points is caused by the onset of ingression at the lagging edge and a ceased ingression at the leading edge. Together this will change the forces on the sides of the ring and causes a movement towards the initial ring center. However, since the

difference to control embryos is not significant and our explanation remains speculative, we decided not to include this detailed analysis in the manuscript.

Reviewer #3:

In this paper, the authors investigate the role of anillin in asymmetrical cytokinesis using *C. elegans* as a model system. They build on a previous paper showing that cytokinetic ingression leads to circumferential cortical flows towards the lagging edge, which causes reductions of Myosin II levels in an anillin dependent manner. Here, they investigate the involvement of anillin. Using different variants and KD approaches, they show convincingly that anillin binds to RhoA, and that the linker region of anillin is involved in the asymmetry of the ring closure.

Overall, this is a good study on an interesting topic. I have a few comments that I believe would make the study stronger.

1. One major limitation of this study is the lack of evidence that what the linker responds to is circumferential flow ("the linker region senses circumferential flows" (line 275)).

Could the author demonstrate this using the same MKLP1 KD that was used in the Hsu 2023? It should lead to cortical flows starting but then stopping. According to the model presented in this paper, one should expect an increase in anillin binding and a decrease in Myosin intensity, followed by the reverse.

Answer: We thank the reviewer for his/her enthusiasm for our study and for proposing the MKLP1 (*C. elegans* ZEN-4) depletion experiment. However, the *zen-4(RNAi)* phenotype has multiple issues that make the results inconclusive. ZEN-4 and its interaction partner CYK-4 (human MgcRacGAP1) promote the activation of RHO-1 by activating the RhoA GEF ECT-2 (DOI: 10.1016/j.cub.2020.05.090). Therefore, the furrow ingression rate and the levels of ANI-1 and NMY-2 in the ring are strongly reduced when ZEN-4 function is compromised (Fig. S3 in DOI: 10.1126/science.1163086, Fig. 5D in DOI: 10.1091/mbc.E09-01-0089) (Reviewer 3, Figure D). Furthermore, the cortical flows are largely reduced in *zen-4(RNAi)* embryos in comparison to control embryos (compare Fig. 4C with 4D in DOI: 10.1038/s41467-023-43996-4).

During the slow half ingression our model would predict that weak cortical flows promote the function of the linker and thereby facilitate the binding of the ANI-1's RBD to active RHO-1. This should result in lower NMY-2 levels in the ring after *zen-4(RNAi)*, which indeed is what we observed (Reviewer 3, Figure D). However, since ZEN-4 also promotes RHO-1 activation we are not able to distinguish between the role of the flows and of ZEN-4 in RHO-1 activation. Therefore, we decided to not include this experiment into the revised manuscript.

2. The other major issue I have is that images do not always obviously show what is described, and I think the overall quality of the images could be better. For example, figure 1F is described in the text as showing a higher intensity of GFP::ROK compared to the control but this is not obvious from the images. Is it a bad example? (Figure 1C-D and Figure S1C-D which are showing similar things are a lot clearer)

Similarly, in figure 2, the authors comment on linear structures which I do not clearly see in the images on figure 2C. However, from the images, it looks like the intensities between the different treatments are different, but the quantification shows otherwise. Are the images not representative?

In Figure 3B, the intensity differences are interesting, but the main difference I see is the contrast between the dotted phenotype of both the control and the GFP::ANI-1 wt in siRNA context, and all the rest where the localization of Myosin is a lot more diffuse. Could the authors comment?

Answer: To highlight the intensity differences of NMY-2::mKate, GFP::ROK^{LET-502} and formin^{CYK-1::GFP} between control and *ani-1(RNAi)* embryos we added zoom-in panels of the equator region with 'fire' coloring (Figs. 1C, F, S1D). In the 'fire' coloring regime yellow reflects high fluorescence intensity and blue/black indicates low fluorescence intensity. We hope that reviewer now finds it easier to observe the intensity differences between control and *ani-1(RNAi)* embryos.

We described the quantification of the IT-RHO-1 structures in detail in our previous publication (Fig. S3A in DOI: 10.1016/j.celrep.2023.113076). We now added Fig. S3A to the manuscript to explain in detail how the IT-RHO-1 structures were measured and classified. After applying different filters to the fluorescent IT-RHO-1 images, we measured the width and length of all bright structures manually at the equatorial region. Structures that were four times longer than wider were defined as linear and the remaining ones as non-linear. We summarized this analysis in the methods section in the current manuscript (lines 1108-1114). IT-RHO-1 fluorescence intensity is similar at the cell equator for all RNAi treatments (Fig. 3C) and to reflect this result better we exchanged the control, *ani-1(RNAi)* and *cyk-1(RNAi)* *ani-1(RNAi)* embryos in Fig. 3A.

The "dotted phenotype" the reviewer refers to in Fig. 3B (now Fig. 4B) are the 'NMY-2 patches' that are ANI-1-dependent and were first described by Maddox et al. (DOI: 10.1242/dev.01828). We mention them in the manuscript and marked them in Fig. 1C (lines 78-81, 135-136). Formation of NMY-2 patches probably requires NMY-2 binding to ANI-1. Since the exact NMY-2 binding site on ANI-1 is not known and the patches are not highly relevant for our story, we limited our analysis to measuring average NMY-2 intensity regardless of being in patches or not.

Others:

3. In figure 4, I also find it striking that when anillin is depleted the top/bottom asymmetry of the ring seems to be enhanced compared to the control. Could the authors comment?

Answer: For the end-on reconstruction of the contractile ring we acquired 21 z-planes every 15 s and cropped, rotated and projected the fluorescence intensity of the cell equator (line 325-329, Fig. S5A). Since the fluorescence intensity declines with increasing z-distance from the objective, the top part of the ring that is closer to the objective, appears typically much brighter than the bottom part (farther away from the objective). To illustrate that, we included an embryo expressing a general membrane marker (PH domain) which is not enriched any a particular side of the ring but appears also brighter at the top, which is the side next to the objective (Fig. S5A). To prevent any influence of this intensity difference between top and bottom on our fluorescence intensity quantifications, we always quantified the left and right sides of the ring and assigned them as the leading or lagging edge (Fig. 6C, lines 1133-1143). We had one embryo where the leading edge moved directly upward, away from the coverslip and therefore we excluded this embryo from the analysis as mentioned in the Materials and Methods (lines 1137-1139).

4. I am finding the IT-RHO-1 tagging strategy a little unclear. How can the authors conclude that the strategy works well since the band they see in S2C as non specificity is in fact exactly where their band for IT-RHO-1 is? Why do they use such a strategy?

Answer: If we blot control (*rho-1/rho-1*) animals with anti-RHO-1 antibody we have a band for endogenous RHO-1 and an additional unspecific band just below 50 kDa (Fig. S2C, left blot). In

the animals expressing IT-RHO-1 this unspecific band at ~50 kDa is more intense than in control animals, suggesting that IT-RHO-1 has the same molecular weight. The anti-RHO-1 antibody was self-made and, unfortunately, we have no other anti-RHO-1 antibody available. That IT-RHO-1 is expressed and runs at the expected molecular weight is confirmed by our immunoblot with the anti-GFP antibody. Here, we only detect a band in the IT-RHO-1 expressing animals but not in the control animals (Fig. S2C, right).

We pursued different fluorophore tagging strategies for RHO-1. We introduced a GFP or mCherry tag at the N-terminus with different linker regions between the fluorophores and the RHO-1 coding region. However, in all those embryos, N-terminal tagged RHO-1 was not enriched at the cell equator during cytokinesis (Reviewer 3, Figure E). Tagging at the C-terminus is not possible, since RHO-1 has a C-terminal CAAX motif that must be cleaved off. Since the N-terminal tagged RHO-1 did not localize as expected and the C-terminal tagging is not feasible, we decided to pursue the internal tagging strategy which was first successfully established for Cdc42 in yeast and later for RhoA in *Xenopus* embryos (DOI: 10.7554/eLife.50471; 10.1371/journal.pbio.1002097).

5. Figure S2E-F are lacking a control (successful cytokinesis in *rho-1/rho-1* animals).

Answer: Thanks for pointing this out, we added the control (*rho-1/rho-1*) embryos to the graph.

Minor:

6. A scheme showing the different domains of Anillin and the different actors they interact with would be helpful early on (maybe in figure 1 or s1 rather than later).

Answer: We added in Fig. 1E a scheme of *C. elegans* ANI-1 with its predicted and known interaction partners that are relevant for our manuscript.

The text is sometimes difficult to follow for a non specialist. Consider adding more schemes throughout to make it easier.

Answer: To illustrate our image processing and image analysis better we added a few more schemes and updated existing ones. We included a new overview of the components of the RhoA signaling pathway analyzed in our study in Fig. S1C, a new scheme on the image acquisition and processing to construct the end-on view of the contractile ring (Fig. S5A), and modified the existing schemes in Figs. 5B, 6C, and 8C.

Figure 1: could the authors comment on the fact that for Myosin the difference between the anillin depleted cells is only clear at 165s, vs for the Formin the difference is already visible at 135s (at least in the presented example)?

Answer: We agree with the reviewer that the difference between control and *ani-1(RNAi)* is visible earlier for the formin than for myosin in the quantifications (Fig. 1D, S1E). We speculate this difference could be caused by the different affinities of ROK and formin for active RhoA. Maybe formin binds to active RhoA more efficiently than ROK and therefore if ANI-1 is not blocking the RhoA effector side anymore, it gets recruited and activated faster.

Writing could be a little smoother at times eg line 108 ANI-1 mediates [...] by [...] by [...].
Line 109: should "ANI-1 phenotype" be replaced by "phenotype of ANI-1 depletion"?

Answer: We revised the wording of these examples and worked to improve the English of the manuscript.

March 6, 2025

RE: JCB Manuscript #202405182R

Esther Zanin
Friedrich-Alexander-Universität Erlangen-Nürnberg

Dear Esther,

Thank you for submitting your revised manuscript entitled "Anillin mediates unilateral furrowing during cytokinesis by limiting RhoA binding to its effectors" which has been evaluated by the three original reviewers. We would be happy to publish your paper in JCB pending final revisions necessary to address the final reviewer comments and to meet our formatting guidelines (see details below).

We ask that you address the request to provide quantification of the experiments in which anillin fragments were pulled down with Rho. We also ask that you carefully consider the comments made by the reviewers intended to increase the accessibility of the manuscript to the readership of JCB.

A. MANUSCRIPT ORGANIZATION AND FORMATTING:

1) Text limits: Character count for Articles is < 40,000, not including spaces. Count includes title page, abstract, introduction, results, discussion, and acknowledgments. Count does not include materials and methods, figure legends, references, tables, or supplemental legends.

2) Figure formatting: Articles may have up to 10 main text figures. Scale bars must be present on all microscopy images, including inset magnifications. Molecular weight or nucleic acid size markers must be included on all gel electrophoresis. Please add MW markers to the actin blot in Figure S2C as well as AlphaFold quality control metrics to S2A.

Also please avoid pairing red and green for images and graphs to ensure legibility for color-blind readers. If red and green are paired for images, please ensure that the particular red and green hues used in micrographs are distinctive with any of the colorblind types. If not, please modify colors accordingly or provide separate images of the individual channels.

3) Statistical analysis: Error bars on graphic representations of numerical data must be clearly described in the figure legend. The number of independent data points (n) represented in a graph must be indicated in the legend. Please, indicate whether 'n' refers to technical or biological replicates (i.e. number of analyzed cells, samples or animals, number of independent experiments). If independent experiments with multiple biological replicates have been performed, we recommend using distribution-reproducibility SuperPlots (please see Lord et al., JCB 2020) to better display the distribution of the entire dataset, and report statistics (such as means, error bars, and P values) that address the reproducibility of the findings.

Statistical methods should be explained in full in the materials and methods. For figures presenting pooled data the statistical measure should be defined in the figure legends. Please also be sure to indicate the statistical tests used in each of your experiments (both in the figure legend itself and in a separate methods section) as well as the parameters of the test (for example, if you ran a t-test, please indicate if it was one- or two-sided, etc.). Also, if you used parametric tests, please indicate if the data distribution was tested for normality (and if so, how). If not, you must state something to the effect that "Data distribution was assumed to be normal but this was not formally tested."

4) Materials and methods: Should be comprehensive and not simply reference a previous publication for details on how an experiment was performed. Please provide full descriptions (at least in brief) in the text for readers who may not have access to referenced manuscripts. The text should not refer to methods "...as previously described." Please also indicate the type of membrane used for immunoblotting as well as describe acquisition and quantification methods.

5) For all cell lines, vectors, constructs/cDNAs, etc. - all genetic material: please include database / vendor ID (e.g. Addgene, ATCC, etc.) or if unavailable, please briefly describe their basic genetic features, even if described in other published work or gifted to you by other investigators (and provide references where appropriate). Please be sure to provide the sequences for all of your oligos: primers, si/shRNA, RNAi, gRNAs, etc. in the materials and methods. You must also indicate in the methods the source, species, and catalog numbers/vendor identifiers (where appropriate) for all of your antibodies, including secondary. If

antibodies are not commercial, please add a reference citation if possible.

6) Microscope image acquisition: The following information must be provided about the acquisition and processing of images:

- a. Make and model of microscope
- b. Type, magnification, and numerical aperture of the objective lenses
- c. Temperature
- d. Imaging medium
- e. Fluorochromes
- f. Camera make and model
- g. Acquisition software
- h. Any software used for image processing subsequent to data acquisition. Please include details and types of operations involved (e.g., type of deconvolution, 3D reconstitutions, surface or volume rendering, gamma adjustments, etc.).

7) References: There is no limit to the number of references cited in a manuscript. References should be cited parenthetically in the text by author and year of publication. Abbreviate the names of journals according to PubMed.

8) Supplemental materials: Articles may have up to 5 supplemental figures and 10 videos. Please also note that tables, like figures, should be provided as individual, editable files. A summary of all supplemental material should appear at the end of the Materials and methods section. Please include one brief sentence per item.

9) Video legends: Should describe what is being shown, the cell type or tissue being viewed (including relevant cell treatments, concentration and duration, or transfection), the imaging method (e.g., time-lapse epifluorescence microscopy), what each color represents, how often frames were collected, the frames/second display rate, and the number of any figure that has related video stills or images.

10) eTOC summary: A ~40-50 word summary that describes the context and significance of the findings for a general readership should be included on the title page. The statement should be written in the present tense and refer to the work in the third person. It should begin with "First author name(s) et al..." to match our preferred style.

11) Conflict of interest statement: JCB requires inclusion of a statement in the acknowledgements regarding competing financial interests. If no competing financial interests exist, please include the following statement: "The authors declare no competing financial interests." If competing interests are declared, please follow your statement of these competing interests with the following statement: "The authors declare no further competing financial interests."

12) A separate author contribution section is required following the Acknowledgments in all research manuscripts. All authors should be mentioned and designated by their first and middle initials and full surnames. We encourage use of the CRediT nomenclature (<https://casrai.org/credit/>).

13) ORCID IDs: ORCID IDs are unique identifiers allowing researchers to create a record of their various scholarly contributions in a single place. Please note that ORCID IDs are required for all authors. At resubmission of your final files, please be sure to provide your ORCID ID and those of all co-authors.

14) JCB requires authors to submit Source Data used to generate figures containing gels and Western blots with all revised manuscripts. This Source Data consists of fully uncropped and unprocessed images for each gel/blot displayed in the main and supplemental figures. For assays performed using capillary electrophoresis and/or immunoassay-based detection, authors should instead provide the electropherogram graph(s) for each experiment, plotting fluorescence/chemiluminescence intensity vs. molecular weight/size. Since your paper includes cropped gel and/or blot images, please be sure to provide one Source Data file for each figure gels, blots, and/or capillary electrophoresis assays along with your revised manuscript files. File names for Source Data figures should be alphanumeric without any spaces or special characters (i.e., SourceDataF#, where F# refers to the associated main figure number or SourceDataFS# for those associated with Supplementary figures). For traditional gels and blots, the lanes of the gels/blots should be labeled as they are in the associated figure, the place where cropping was applied should be marked (with a box), and molecular weight/size standards should be labeled wherever possible. For capillary electrophoresis assays, each trace in the graph should be color-coded and labeled to indicate which protein, gene, or sample is being measured (please try to avoid red/green combinations to accommodate our color-blind readers).

Source Data files will be directly linked to specific figures in the published article. Source Data Figures should be provided as individual PDF files (one file per figure). Authors should endeavor to retain a minimum resolution of 300 dpi or pixels per inch. Please review our instructions for export from Photoshop, Illustrator, and PowerPoint here: <https://rupress.org/jcb/pages/submission-guidelines#revised>

15) Journal of Cell Biology now requires a data availability statement for all research article submissions. These statements will be published in the article directly above the Acknowledgments. The statement should address all data underlying the research presented in the manuscript. Please visit the JCB instructions for authors for guidelines and examples of statements at (<https://rupress.org/jcb/pages/editorial-policies#data-availability-statement>).

B. FINAL FILES:

Thank you for your attention to these final processing requirements. Please revise and format the manuscript and upload materials within 14 days. If you need an extension for whatever reason, please let us know and we can work with you to determine a suitable revision period.

Thank you for this interesting contribution, we look forward to publishing your paper in Journal of Cell Biology.

Sincerely,

William Bement, PhD
Monitoring Editor
Journal of Cell Biology

Dan Simon, PhD
Scientific Editor
Journal of Cell Biology

Reviewer #1 (Comments to the Authors (Required)):

The authors have very reasonably addressed all the questions that I raised in my earlier review. I thank them for their good-faith effort and the substantial amount of additional work that they have done. I think this is a very interesting report and will interest the readers of JCB.

I hesitate to ask this of them, given all their efforts: but it would be good to have some confirmation that the binding data in Fig 2B and C are reproducible - quantitation or some statement of reproducibility.

Reviewer #2 (Comments to the Authors (Required)):

The authors did an impressive job at exhaustively responding to my reviews and improving the manuscript with new data and important textual changes. The experiments are performed to a very high standard, the data are compelling and should be published.

However, I am still finding some of the major conclusions incomplete and misleading, namely those concerning "anillin's inhibitory role on RhoA signaling", which implies that anillin globally suppresses RhoA signaling. However, only RhoA signalling to actomyosin components are shown and considered. From multiple lines of evidence in the literature, ANI-1 does not inhibit RhoA signalling to the septin cytoskeleton, rather it mediates it. It seems that RhoA bound to ANI-1/septins is not being considered as RhoA signalling, which is what I find misleading. I fully accept that the data shown support the conclusion that anillin can act as a negative regulator of actomyosin, but the substantial evidence of its role as a positive regulator of septins should not be ignored, especially since this is also under the control of RhoA and thus constitutes an "activity" of RhoA. Rather than labelling anillin as "a negative regulator of RhoA activity" throughout the manuscript, I suggest the authors rephrase as a negative regulator of RhoA-dependent actomyosin effectors. I urge they also consider and discuss what RhoA-bound ANI-1 and septins might be doing besides "inhibiting" actomyosin.

It would seem that in binding RhoA (and subsequently recruiting septins as shown by the authors in Lebedev et al 2023)), ANI-1 sequesters some RhoA away from the actomyosin cytoskeleton to recruit/assemble the septin cytoskeleton. This supports the hypothesis by Carim and Hickson (2020 and 2023) in their discussions of the actomyosin-anilloseptin contractile ring. Indeed, a putative transfer of RhoA from the actomyosin effectors to ANI-1 would fit the data shown here that the levels of RHO-1 are unchanged upon ANI-1 depletion. Without ANI-1, a greater proportion of RHO-1 remains associated with actomyosin than usual because it cannot stimulate the competing septin assembly pathway. Although the authors try to argue against it in their rebuttal, the observed enhanced levels of actomyosin upon ANI-1 depletion could reflect reduced actomyosin disassembly (coinciding with reduced septin assembly). This does not contradict, but rather provides an explanation for the inhibitory mechanism on actomyosin uncovered by the authors. In sequestering RhoA (and assembling septins), ANI-1 could inhibit actomyosin assembly, which equates to promoting net actomyosin disassembly as ring circumference decreases.

On a related note, ANI-1-depleted contractile rings, clearly shown here to have enhanced levels of NMY-2, need those enhanced levels of NMY-2 to close, as shown by Maddox (2007) who found that ANI-1 depleted furrows are sensitive to normally-tolerated partial depletions of NMY-2. If ANI-1 was only "an inhibitor of RhoA signalling", this should not be the case. Rather it shows that other roles of ANI-1 are required, the most obvious being the sequestration of RHO-1 and recruitment of septins to the plasma membrane.

When discussing the interesting data regarding the linker region, it's not entirely clear what is meant by a RhoA stabilizer versus inhibitor. RhoA inhibition usually implies GTP-hydrolysis via GAP activity, so I think the word inhibitor should be avoided. But I appreciate the authors trying to accommodate different models, including the Budnar et al work. Perhaps the differences with and without the linker could reflect different degrees of sequestration of RHO-1, such that weaker sequestration may allow some RHO-1 to be released to re-activate actomyosin effectors (as per Budnar) while stronger sequestration does not?

Reviewer #3 (Comments to the Authors (Required)):

The authors have essentially answered my main comments. My remaining suggestion at this point would be to streamline the paper. There are many different constructs/strains/experiments and it renders the manuscript very dense and very long and difficult to read. There could be more intermediary conclusions eg line 221 and transitions between paragraphs. One could also consider reorganizing the manuscript to really focus on the main conclusions which would require changing the titles of paragraphs/figures to put more emphasis on the general conclusions and less on the technical details.

Some minor point I think should be clarified:

I did not understand the sentence "in contrast ..." line 208. Do the authors mean the levels are no more increased than controls?

Line 229: what does "after the reduction of F-actin in vivo" refer to?

Line 331: leading of lagging: should it be "or"?

I like the model in figure 10, could this be made more complete/bigger?

FRIEDRICH-ALEXANDER
UNIVERSITÄT
ERLANGEN-NÜRNBERG

NATURWISSENSCHAFTLICHE
FAKULTÄT

Prof. Dr. Esther Zanin

Friedrich-Alexander-Universität Erlangen
Experimentelle Molekulare Zelldynamik
Department Biologie, Raum 02.382
Staudtstr. 5
91058 Erlangen / Germany
Email: esther.zanin@fau.de

Tuesday, March 18, 2025

Dear Bill and Dan,

we are delighted that our manuscript entitled “**Anillin mediates unilateral furrowing during cytokinesis by limiting RhoA binding to its effectors**” is provisionally accepted as an article in *JCB*.

We addressed the final comment of reviewer 1 and added two panels in which we quantified the pull-downs of RhoA with different anillin fragments (Fig. 2B, 2C). This quantification supports our conclusion that 1) the C-terminus of ANI-1 preferentially binds active RHO-1, 2) the linker region enhances the interaction between the C-terminus and RHO-1, and 3) the linker region alone does not bind active RHO-1.

We included in our discussion a part which highlights that ANI-1 inhibits myosin II activation and F-actin polymerization but at the same time still recruits septins to the ring, as suggested by reviewer 2 (lines 559-568).

We also shortened and streamlined the manuscript to improve the accessibility of the manuscript for the readership of *JCB*.

We hope that our manuscript will now be accepted as an article in *JCB*.

Sincerely,

Esther Zanin

Professor for Experimental Molecular Cellular Dynamics